## OPEN

# Within-sibship genome-wide association analyses decrease bias in estimates of direct genetic effects

**Estimates from genome-wide association studies (GWAS) of unrelated individuals capture effects of inherited variation (direct effects), demography (population stratification, assortative mating) and relatives (indirect genetic effects). Family-based GWAS designs can control for demographic and indirect genetic effects, but large-scale family datasets have been lacking. We combined data from 178,086 siblings from 19 cohorts to generate population (between-family) and within-sibship (within-family) GWAS estimates for 25 phenotypes. Within-sibship GWAS estimates were smaller than population estimates for height, educational attainment, age at first birth, number of children, cognitive ability, depressive symptoms and smoking. Some differences were observed in downstream SNP heritability, genetic correlations and Mendelian randomization analyses. For example, the within-sibship genetic correlation between educational attainment and body mass index attenuated towards zero. In contrast, analyses of most molecular phenotypes (for example, low-density lipoprotein-cholesterol) were generally consistent. We also found within-sibship evidence of polygenic adaptation on taller height. Here, we illustrate the importance of family-based GWAS data for phenotypes influenced by demographic and indirect genetic effects.**

GWAS have identified thousands of genetic variants associated with complex phenotypes[1,2], typically using samples of non-closely related individuals[3]. GWAS associations can be interpreted as estimates of direct individual genetic effects, that is, the effect of inheriting a genetic variant (or a correlated variant) on a phenotype[4–6]. However, there is growing evidence that GWAS associations for some phenotypes estimated from samples of unrelated individuals also capture effects of demography[7,8] (assortative mating[9–11] and population stratification[12]) and indirect genetic effects of relatives[13–19] (Fig. 1). For example, Lee et al.[14] found that within-sibship GWAS estimates for educational attainment variants were around 40% lower than estimates from unrelated individuals, indicating the presence of demographic and indirect genetic effects. These nondirect sources of genetic associations are themselves of interest for estimating parental effects[13,18], understanding human mate choice[9–11] and genomic prediction[14,19]. However, they can also impact downstream analyses using GWAS summary data such as biological annotation, heritability estimation[20–22], genetic correlations[23], Mendelian randomization (MR)[7,24,25] and polygenic adaptation tests[26–29].

Within-family genetic association estimates, such as those obtained from samples of siblings, can provide less biased estimates of direct genetic effects because they are unlikely to be affected by demographic and indirect genetic effects of parents[7,17,30–34]. GWAS using siblings (within-sibship GWAS) (Fig. 2) have been previously limited by available data, but are now feasible by combining well-established family studies with recent large biobanks that incidentally or by design contain thousands of sibships[35–39].

Here, we report findings from a within-sibship GWAS of 25 phenotypes using data from 178,076 siblings from 19 studies, the largest GWAS conducted within sibships to date (Fig. 3). Our results are broadly consistent with previous studies comparing population and within-sibship genetic effect estimates in smaller sample sizes[13,14,19,40]. We found that within-sibship meta-analysis GWAS estimates are smaller than population estimates for seven phenotypes (height, educational attainment, age at first birth, number of children, cognitive ability, depressive symptoms and smoking).

We show that these differences in GWAS estimates, which are likely to partially reflect demographic and indirect genetic effects, can affect downstream analyses such as estimates of heritability, genetic correlations and MR. However, we find that genetic associations with most clinical phenotypes, such as lipids, are less strongly affected. We found strong evidence of polygenic adaption on taller human height using within-sibship data. Our study illustrates the importance of collecting genome-wide data from families to understand the effects of inherited genetic variation on phenotypes that are affected by assortative mating, population stratification and indirect genetic effects.

## Results

**Within-sibship and population-based GWAS comparison.** For GWAS analyses we used data from 178,076 individuals (with one or more genotyped siblings) from 77,832 sibships in 19 studies. Sample sizes for individual phenotypes ranged from 13,375 to 163,748 (median: 82,760, mean: 79,794). More information on sample sizes from individual cohorts and for each phenotype is contained in Supplementary Table 1. We used within-sibship models which use deviations of the individual's genotype from the mean genotype within the sibship (that is, all siblings in the family present in the study). For example, in a sibling pair where one sibling has two risk alleles and the other sibling has one risk allele, the mean sibship genotype is 1.5 risk alleles and the individual's deviations are +0.5 and −0.5, respectively. The within-sibship model includes the mean sibship genotype as a covariate to capture the between-family contribution of the SNP[14]. For comparison, we also applied a standard population GWAS model; a covariate-adjusted linear regression of the outcome on raw genotype, which does not account for the mean sibship genotype. Standard errors were clustered by sibship. Age, sex and principal components were included as covariates in both models. All GWAS analyses were performed in individual cohort studies separately using R (v.3.5.1) and meta-analyses were conducted across these using summary data. Amongst the phenotypes analyzed,

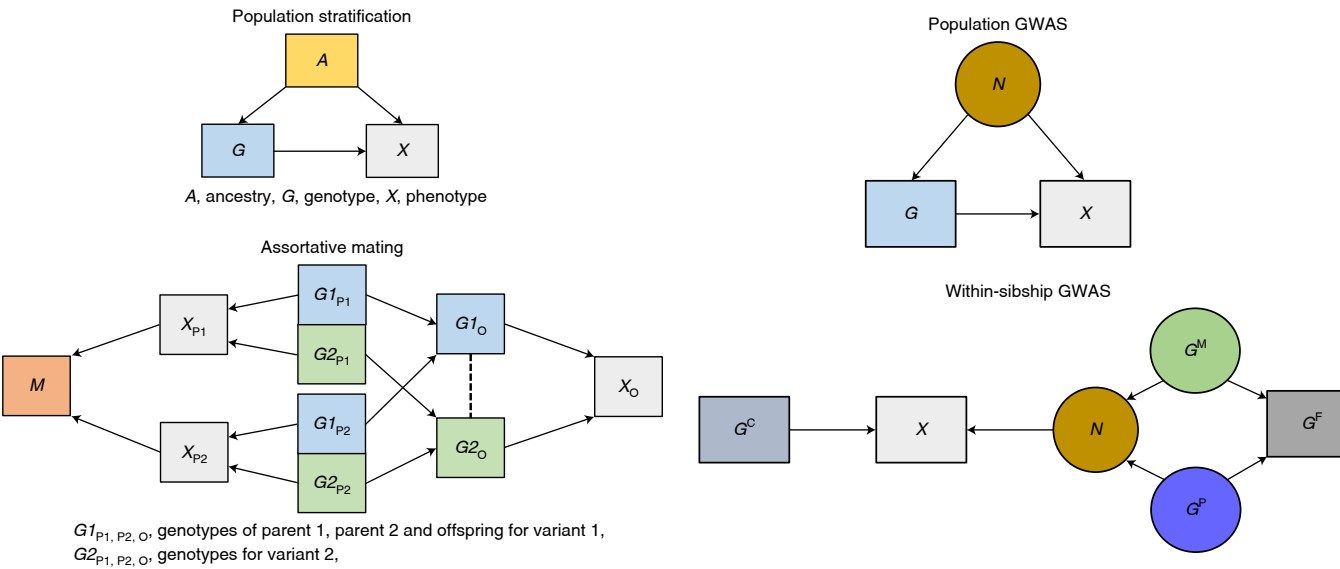

Fig. 1 | Demographic and indirect genetic effects. Population stratification: population stratification is defined as the distortion of associations between a genotype and a phenotype when ancestry $A$ influences both genotype $G$ (via differences in allele frequencies) and the phenotype $X$. Principal components and linear mixed model methods control for ancestry but they may not completely control for fine-scale population structure. Assortative mating: assortative mating is a phenomenon where individuals select a partner based on phenotypic (dis)similarities. For example, tall individuals may prefer a tall partner. Assortative mating can induce correlations between causes of an assorted phenotype in subsequent generations. If a phenotype $X$ is influenced by two independent genetic variants $G1$ and $G2$ then assortment on $X$ (represented by effects of $X$ on mate choice $M$) will induce positive correlations between $G1$ in parent 1 and $G2$ in parent 2 and vice versa. Parental transmission will then induce correlations between otherwise independent $G1$ and $G2$ in offspring. These correlations can distort genetic association estimates. Indirect genetic effects: indirect genetic effects are effects of relative genotypes (via relative phenotypes and the shared environment) on the index individual's phenotype. These indirect effects influence population GWAS estimates because relative genotypes are also associated with genotypes of the index individual. Indirect genetic effects of parents on offspring are of most interest because they are likely to be the largest. However, indirect genetic effects of siblings or more distal relatives are also possible.

the largest available sample sizes in a meta-analysis of European cohorts were for height ($N = 149,174$), body mass index (BMI) ($N = 140,883$), educational attainment ($N = 128,777$), ever smoking ($N = 124,791$) and systolic blood pressure (SBP) ($N = 109,588$) (Supplementary Table 2). We also report stratified results from non-European samples including 13,856 individuals from the China Kadoorie Biobank. Sample sizes here refer to the number of individuals across all sibships.

Fig. 2 | Population GWAS estimate the association between raw genotypes $G$ and phenotypes $X$. As outlined in Fig. 1, estimates from population GWAS may not fully control for demography (population stratification and assortative mating) and may also capture indirect genetic effects of relatives. For simplicity we use $N$ to represent all sources of associations between $G$ and $X$ that do not relate to direct effects of $G$. Circles indicate unmeasured variables and squares indicate measured variables. If parental genotypes are known, $G$ can be separated into nonrandom (determined by parental genotypes) and random (relating to segregation at meiosis) components. Within-sibship GWAS include the mean genotype across a sibship ($G^f$) (a proxy for the mean of the paternal and maternal genotypes $G^{P,M}$) as a covariate to capture associations between $G$ and $X$ relating to parents. The within-sibship estimate is defined as the effect of the random component: that is, the association between family-mean-centered genotype $G^c$ (that is, $G - G^f$) and $X$. Demography and indirect genetic effects of parents ($N$) will be captured by $G^f$. The association between $G^c$ and $X$ will not be influenced by these sources of association but could be affected by indirect effects of the siblings themselves, which are not controlled for.

Previous studies have found that association estimates of height and educational attainment genetic variants are smaller in within-family models[13,14,40]. We aimed to investigate whether similar shrinkage in association estimates is observed for other phenotypes by comparing within-sibship and population genetic association estimates for 25 phenotypes that were widely available in family-based studies. We observed the largest within-sibship shrinkage (% decrease in association estimates from population to within-sibship models) for genetic variants associated with number of children (67%; 95% confidence interval (95% CI) 4%, 130%), age at first birth (52%; 30%, 75%), depressive symptoms (50%; 18%, 82%) and educational attainment (47%; 41%, 52%). We also found evidence of shrinkage for cognitive ability (22%; 6%, 37%), ever smoking (19%; 9%, 30%) and height (10%; 8%, 12%). In contrast, within-sibship association estimates for C-reactive protein (CRP) were larger than population estimates (−9%; −15%, −2%). We found limited evidence of within-sibship differences for the remaining 17 phenotypes, including BMI and SBP (Fig. 4 and Supplementary Table 3).

We investigated possible heterogeneity in shrinkage for height and educational attainment genetic variants across variants and between cohorts. Using the meta-analysis results, we did not observe strong evidence of heterogeneity in shrinkage across variants that were strongly associated with height and educational attainment. This suggests that shrinkage may be largely uniform across these

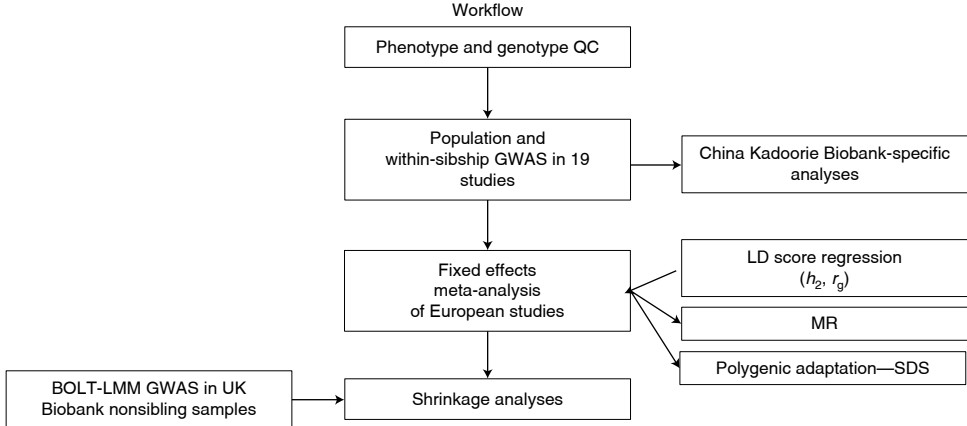

**Fig. 3 | A flowchart of analyses undertaken in this project.** We started by performing quality control and running GWAS models in 19 individual cohorts. We then meta-analyzed GWAS data from 18 of these cohorts with European-ancestry individuals. We then used the European meta-analysis data for downstream analyses including LDSC, MR and polygenic adaptation testing. We performed analyses in the China Kadoorie Biobank separately. QC, quality control.

signals for these phenotypes. We also found limited evidence of cohort heterogeneity in shrinkages for height (heterogeneity $P = 0.89$) and educational attainment ($P = 0.40$) across the European-ancestry cohorts (Extended Data Figs. 1 and 2). In contrast, there was limited evidence for shrinkage on height in the China Kadoorie Biobank (shrinkage −3%; 95% CI −13%, 7%; heterogeneity with European meta-analysis $P = 0.006$) but some evidence of shrinkage on ever smoking (shrinkage = 134%; 10%, 258%) (Extended Data Fig. 3).

**Within-sibship SNP heritability estimates.** Linkage disequilibrium (LD) score regression (LDSC) can use GWAS data to estimate SNP heritability, the proportion of phenotypic variation explained by common SNPs[20,23]. We used simulations to investigate the applicability of LDSC when using within-sibship GWAS data, finding evidence that LDSC can estimate SNP heritability using both population and within-sibship model GWAS data if effective sample sizes (based on standard errors) are used to account for differences in power between the models (Methods).

To evaluate the impact of controlling for demographic and indirect genetic effects, we compared LDSC SNP heritability estimates based on population and within-sibship effect estimates for 25 phenotypes. Theoretically, within-sibship shrinkage in GWAS estimates will also lead to attenuations in within-sibship SNP heritability estimates (Methods). The within-sibship SNP heritability point estimate for educational attainment attenuated by 76% from the population estimate (population $h^2$: 0.13; within-sibship $h^2$: 0.04; difference $P = 5.3 \times 10^{-26}$), with attenuations also observed for cognitive ability (population $h^2$: 0.24; within-sibship $h^2$: 0.14; attenuation 44%; difference $P = 0.011$), ever smoking (population $h^2$: 0.10; within-sibship $h^2$: 0.07; attenuation 25%; difference $P = 0.029$) and height (population $h^2$: 0.41; within-sibship $h^2$: 0.34; attenuation 17%; difference $P = 1.6 \times 10^{-3}$). The observed attenuations were consistent with theoretical expectation (Supplementary Table 4), suggesting that the lower within-sibship SNP heritability estimates are explained by genetic association estimate shrinkage. Across the 21 additional phenotypes, population and within-sibship SNP heritability estimates were relatively consistent (Fig. 5 and Supplementary Table 5). SNP heritability estimates using SumHer[21] with the LDAK-Thin model (expected heritability contribution of each SNP is dependent on allele frequencies and local LD) provided consistent evidence for within-sibship attenuations in SNP heritability for height, educational attainment and cognitive ability (Supplementary Table 6 and Extended Data Fig. 4).

**Within-sibship $r_g$ with educational attainment.** We used LDSC[23] to estimate cross-phenotype genome-wide genetic correlations ($r_g$) between educational attainment and 20 phenotypes with sufficient heritability (population/within-sibship $h^2$ point estimate > 0) and statistical power. To determine the effects of demographic and indirect genetic effects on $r_g$, we compared estimates of $r_g$ using population and within-sibship estimates.

There was strong evidence using population estimates that educational attainment is negatively correlated with BMI ($r_g = -0.32$; −0.37, −0.26), ever smoking ($r_g = -0.41$; −0.49, −0.34) and CRP ($r_g = -0.46$; −0.67, −0.25). However, these correlations attenuated towards zero when using within-sibship estimates: BMI ($r_g = -0.05$; −0.22, 0.12), ever smoking ($r_g = -0.14$; −0.42, 0.14) and CRP ($r_g = -0.06$; −0.43, 0.30), with some evidence at nominal significance for differences between population and within-sibship $r_g$ estimates (BMI difference $P = 5.3 \times 10^{-4}$, ever smoking difference $P = 0.040$, CRP difference $P = 0.039$). These attenuations indicate that genetic correlations between educational attainment and these phenotypes from population estimates may be inflated by demographic and indirect genetic effects (Fig. 6 and Supplementary Table 7).

**Within-sibship MR (WS-MR): effects of height and BMI.** MR uses genetic variants as instrumental variables to assess the causal effect of exposure phenotypes on outcomes[24,41]. MR was originally conceptualized in the context of parent–offspring trios where offspring inherit a random allele from each parent[24]. However, with limited family data, most MR studies have used data from unrelated individuals. WS-MR is largely robust against demographic and indirect genetic effects that could distort estimates from nonfamily designs[7,25]. Here, we used population MR and WS-MR to estimate the effects of height and BMI on 23 phenotypes. These provide a useful comparison as we find evidence of shrinkage in GWAS estimates for height but little evidence of shrinkage for BMI, and both height and BMI have large sample sizes.

WS-MR estimates for height and BMI on the 23 outcome phenotypes were largely consistent with population MR estimates for height based on the slope of a regression of the WS-MR and population MR estimates (−3%; 95% CI −16%, 10%) and BMI (−5%; 95% CI −14%, 4%). However, in agreement with the genetic correlation analyses, we observed differences between population MR and WS-MR estimates of height and BMI on educational attainment. Population MR estimates provided strong evidence that taller height and lower BMI increase educational attainment (0.06 s.d. increase in education per s.d. taller height; 95% CI 0.04, 0.07; 0.19

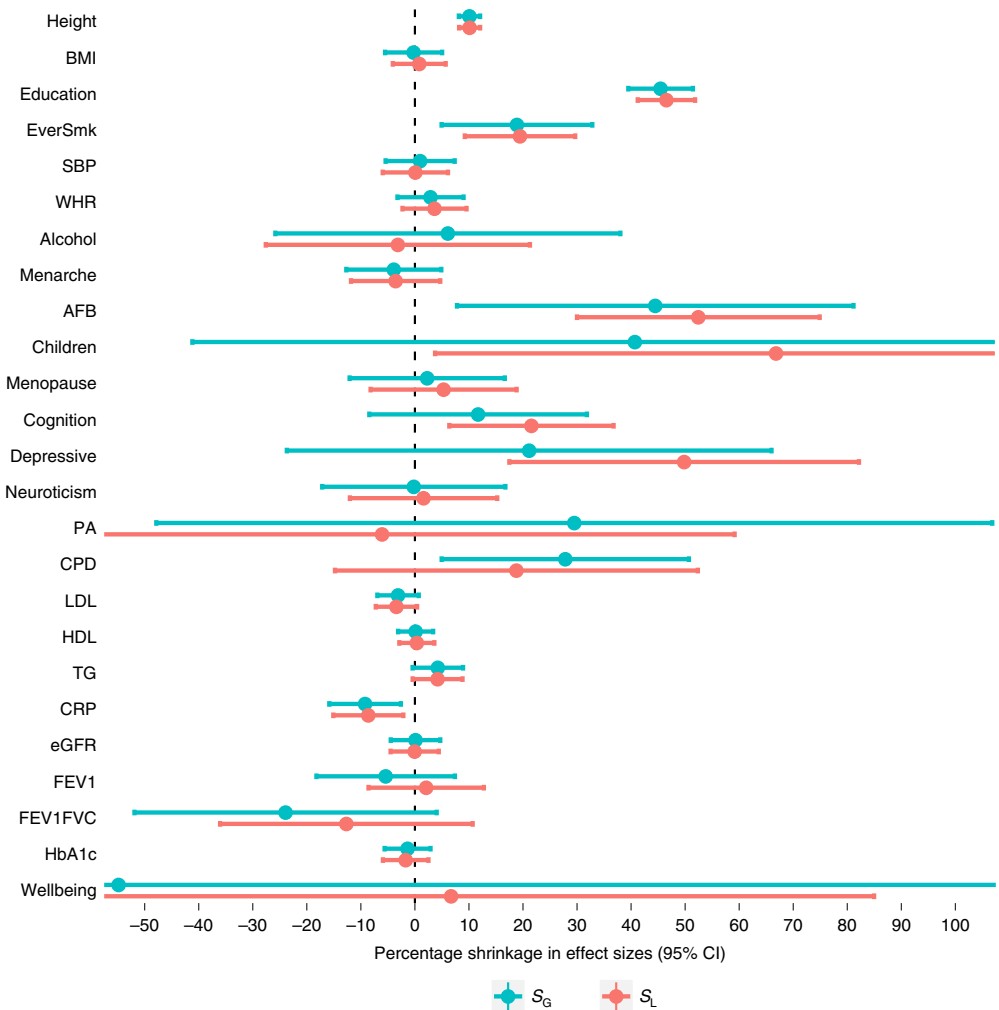

**Fig. 4 | Estimates of shrinkage between population and within-sibship models with corresponding 95% CIs.** Shrinkage is defined as the % decrease in association between the relevant weighted score and phenotype when comparing the population estimate with the within-sibship estimate. Shrinkage was computed as the ratio of two weighted score association estimates with standard errors derived using leave-one-out jackknifing. The number of individuals contributing to each phenotype ranged from $n=149,174$ for height to $n=13,375$ for age at menopause. Further information on the sample sizes of each phenotype is contained in Supplementary Table 2. $S_G$, weighted score at genome-wide significance ($P<5\times10^{-8}$); $S_L$, weighted score at more liberal threshold ($P<1\times10^{-5}$); Education, educational attainment; EverSmk, ever smoking; WHR, waist-to-hip ratio; Alcohol, weekly alcohol consumption; Menarche, age at menarche; AFB, age at first birth; Children, number of biological children; Menopause, age at menopause; Cognition, cognitive ability; Depressive, depressive symptoms; PA, physical activity; CPD, cigarettes per day; LDL, low-density lipoprotein-cholesterol; HDL, HDL-cholesterol; TG, triglycerides; eGFR, estimated glomerular filtration rate; FEV1, forced expiratory volume; FEV1FVC, ratio of FEV1/forced vital capacity; HbA1c, hemoglobin A1C.

s.d. decrease in education per s.d. higher BMI; 0.16, 0.22). In contrast, WS-MR estimates for these relationships were greatly attenuated (height: 0.02 s.d. increase; −0.01, 0.04; difference $P=1.2\times10^{-3}$; BMI: 0.05 s.d. decrease; 0.01, 0.09; difference $P=2.8\times10^{-7}$). We also observed similar attenuation from population MR and WS-MR estimates for BMI on age at first birth (difference $P=2.3\times10^{-3}$) and cognitive ability (difference $P=0.020$); phenotypes highly correlated with education. These differences illustrate instances where population-based MR estimates might be distorted by demographic and indirect genetic effects or other factors (Table 1).

**Polygenic adaptation.** Polygenic adaptation is a process via which phenotypic changes in a population over time are induced by small shifts in allele frequencies across thousands of variants. One method of testing for polygenic adaptation is to compare Singleton Density Scores (SDS), measures of natural selection over the previous 2,000 years (ref. [28]), with GWAS $P$ values. However, this approach is sensitive to population stratification as illustrated by recent work

using UK Biobank data which showed that population stratification in GWAS data likely confounded previous estimates of polygenic adaptation on height[26,27]. Within-sibship GWAS data are particularly useful in this context as they are robust against population stratification[26,27,29]. Here, we recalculated Spearman's rank correlation ($r$) between tSDS (SDS aligned with the phenotype-increasing allele) and our population/within-sibship GWAS $P$ values for 25 phenotypes, with standard errors estimated using jackknifing over blocks of genetic variants.

We found strong evidence for polygenic adaptation on taller height in the European meta-analysis GWAS using both population ($r=0.022$; 95% CI 0.014, 0.031) and within-sibship GWAS estimates ($r=0.012$; 0.003, 0.020) (Extended Data Figs. 5 and 6). These results were supported by several sensitivity analyses: (1) evidence of enrichment for positive tSDS (mean$=0.18$, s.e.$=0.06$, $P=0.003$) amongst 310 putative height loci from the within-sibship meta-analysis results (Extended Data Fig. 7); (2) positive LDSC $r_g$ between height and tSDS in the meta-analysis results

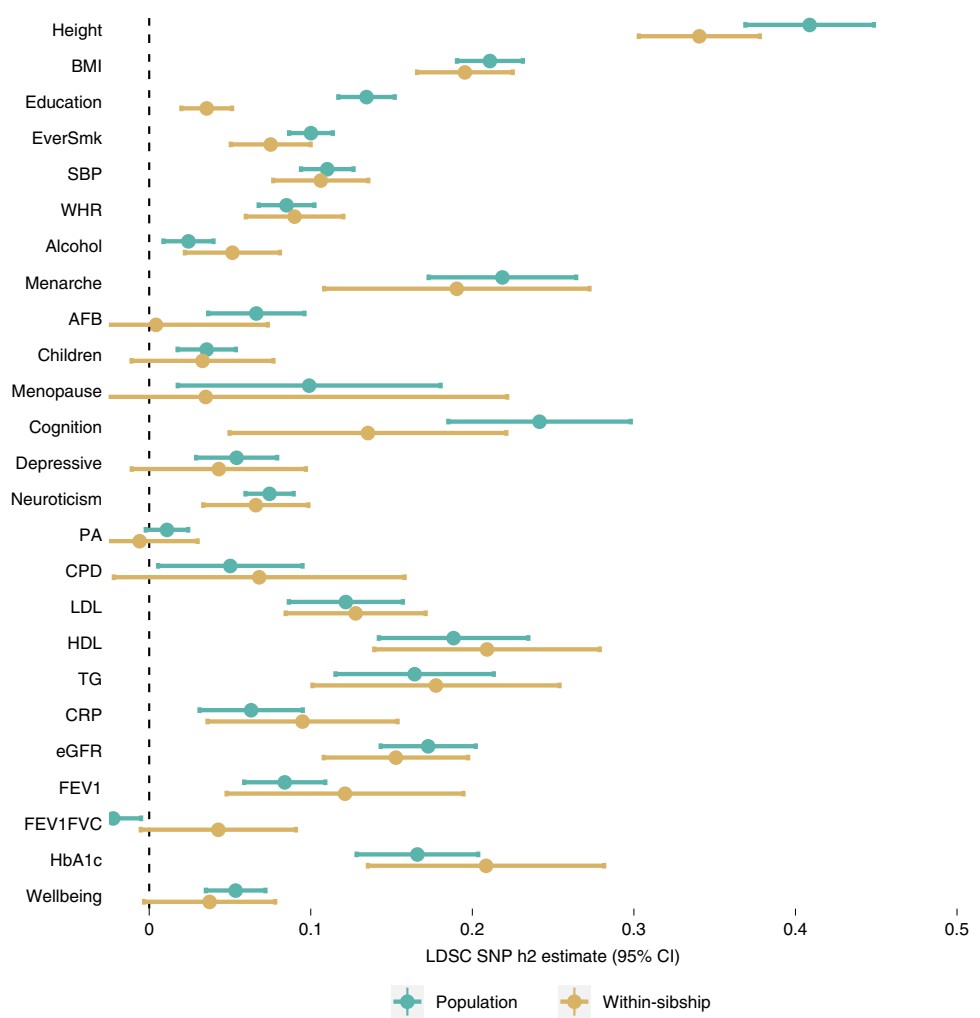

**Fig. 5 | LDSC SNP $h^2$ estimates for 25 phenotypes using population and within-sibship meta-analysis data with corresponding 95% CIs.** The number of individuals contributing to each phenotype ranged from $n = 149,174$ for height to $n = 13,375$ for age at menopause. BMI, body mass index; Education, educational attainment; EverSmk, ever smoking; SBP, systolic blood pressure; WHR, waist-hip ratio; Alcohol, weekly alcohol consumption; Menarche, age at menarche; AFB, age at first birth; Children, number of biological children; Menopause, age at menopause; Cognition, cognitive ability; Depressive, depressive symptoms; PA, physical activity; CPD, cigarettes per day; LDL, LDL cholesterol; HDL, HDL cholesterol; TG, triglycerides; CRP, C-reactive protein; eGFR, estimated glomerular filtration rate; FEV1, forced expiratory volume; FEV1FVC, ratio of FEV1/forced vital capacity; HbA1c, Haemoglobin A1C. Further information on the sample sizes of each phenotype is contained in Supplementary Table 2.

(Supplementary Table 8); and (3) evidence for polygenic adaptation on taller height when meta-analyzing correlation estimates from eight individual studies (for example, SDS using only UK Biobank GWAS summary data) for population ($r = 0.013$; 0.010, 0.015) and within-sibship ($r = 0.004$; 0.002, 0.007) estimates (Fig. 7). There was also some putative within-sibship evidence for polygenic adaptation on increased number of children ($P = 0.024$) and lower high-density lipoprotein (HDL)-cholesterol ($P = 0.024$) (Extended Data Fig. 5).

## Discussion

Here, we report results from the largest within-sibship GWAS to date which included 25 phenotypes and combined data from 178,076 siblings. Consistent with previous studies[13,14,19,40], we found that GWAS results and downstream analyses of behavioral phenotypes (for example, educational attainment, smoking behavior) as well as some anthropometric phenotypes (for example, height, BMI) are affected by demographic and indirect genetic effects. However, we found that most analyses involving more molecular phenotypes, such as lipids, were not strongly affected. This suggests that the best strategy for gene discovery and polygenic prediction for these

phenotypes remains to maximize sample sizes using unrelated individuals. For phenotypes sensitive to demographic and indirect genetic effects, such as educational attainment, family-based estimates are likely to provide less biased estimates of direct genetic effects.

A key aim of GWAS is to estimate direct genetic effects on phenotypes, but other sources of genetic associations can be extremely informative. For example, knowledge of indirect genetic effects can be used to elucidate maternal effects[15,42] or the extent to which health outcomes are mediated by family environments[13,18]. Future family-based GWAS could also provide further estimates of indirect genetic effects[6,18,43].

We found little evidence of heterogeneity in shrinkage estimates at genetic variants strongly associated ($P < 1 \times 10^{-5}$) with height and educational attainment, although power was limited by available samples. The limited detectable heterogeneity could indicate that the observed shrinkage is largely driven by assortative mating or indirect genetic effects. Both of these tend to influence associations proportional to the direct effect, whereas population stratification is likely to have larger effects on ancestrally informative markers. Notably, twin studies have indicated effects of the common environment

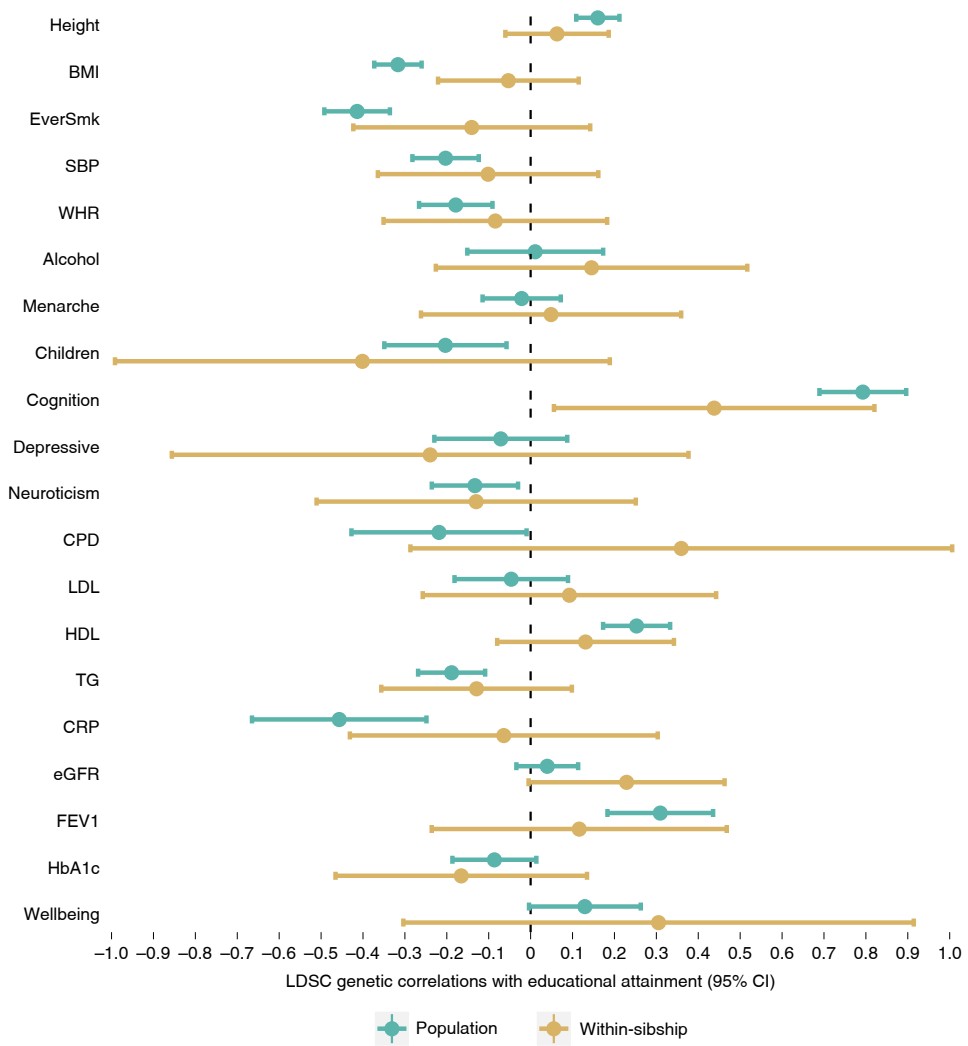

**Fig. 6 | LDSC $r_g$ estimates between educational attainment and 20 phenotypes using population and within-sibship meta-analysis data with corresponding 95% CIs.** The number of individuals contributing to the educational attainment GWAS was $n = 128,777$ with sample sizes for outcomes ranging from $n = 149,174$ for height to $n = 27,638$ for cognitive ability. BMI, body mass index; Education, educational attainment; EverSmk, ever smoking; SBP, systolic blood pressure; WHR, waist-hip ratio; Alcohol, weekly alcohol consumption; Menarche, age at menarche; AFB, age at first birth; Children, number of biological children; Menopause, age at menopause; Cognition, cognitive ability; Depressive, depressive symptoms; CPD, cigarettes per day; LDL, LDL cholesterol; HDL, HDL cholesterol; TG, triglycerides; CRP, C-reactive protein; eGFR, estimated glomerular filtration rate; FEV1, forced expiratory volume; HbA1c, Haemoglobin A1C. Further information on the sample sizes of each phenotype is contained in Supplementary Table 2.

on many of the phenotypes for which we observed shrinkage, such as educational attainment[44], cognitive ability[45] and smoking[46], potentially consistent with indirect genetic effects of parents. In contrast, twin studies do not find strong evidence for common environmental effects on height, where shrinkage is more likely to be a consequence of assortative mating[10,46,47].

The weak evidence for within-sibship shrinkage in the association between BMI genetic variants and BMI is in contrast to the strong evidence from MR analyses (and genetic correlation analyses) that the association between BMI genetic variants and educational attainment does attenuate. These results indicate cross-trait shrinkage in association estimates for BMI genetic variants even in the absence of same-trait shrinkage.

Within-sibship GWAS data can be useful for validating results from larger samples of unrelated individuals. Here, we showed that population MR and WS-MR estimates of the effects of height and BMI were generally consistent for 23 outcome phenotypes. However, we observed differences between within-sibship and population MR estimates of height (on educational attainment) and BMI (on

educational attainment, cognitive ability and age at first birth). This suggests the MR assumptions do not hold for these relationships in samples of unrelated individuals. In subsequent studies, WS-MR could be used as a sensitivity analysis when including phenotypes likely to be affected by demographic or indirect genetic effects[7,25].

We used non-European data from the China Kadoorie Biobank to evaluate whether demographic and indirect genetic effects influence GWAS analyses conducted in the Chinese population. In this sample, we found minimal evidence of shrinkage for height genetic variants but—consistent with the European meta-analysis—suggestive evidence of shrinkage for variants associated with smoking initiation. The absence of shrinkage for height suggests that demographic effects such as assortative mating may differ between populations. Larger within-family studies in non-European populations could be used to evaluate population differences in demographic and indirect effects.

We also used the within-sibship GWAS data to evaluate evidence for recent selection. A previous study reporting polygenic adaptation on height in the UK population was found to be

**Table 1 | WS-MR: effects of height and BMI on 23 phenotypes**

| Outcome (units) | IVW estimate of effect of s.d. increase in height on outcome (95% CI) | | Diff P | IVW estimate of effect of s.d. increase in BMI on outcome (95% CI) | | Diff P |
|---|---|---|---|---|---|---|
| | Population | Within-sibship | | Population | Within-sibship | |
| Age at first birth (years) | 0.27 (0.16, 0.39) | 0.08 (−0.12, 0.29) | 0.052 | −0.79 (−1.04, −0.54) | −0.25 (−0.63, 0.13) | 0.0023 |
| Alcohol consumption (units) | 0.03 (−0.03, 0.09) | 0.03 (−0.07, 0.14) | 0.87 | −0.15 (−0.28, −0.02) | −0.19 (−0.39, 0.02) | 0.71 |
| Cigarettes per day | 0.23 (0.01, 0.46) | 0.29 (−0.12, 0.69) | 0.78 | 0.74 (0.24, 1.23) | 0.56 (−0.21, 1.33) | 0.66 |
| CRP (s.d.) | −0.03 (−0.05, −0.01) | −0.00 (−0.04, 0.03) | 0.078 | 0.28 (0.24, 0.33) | 0.25 (0.18, 0.32) | 0.30 |
| Number of children | −0.02 (−0.04, 0.01) | −0.00 (−0.05, 0.04) | 0.52 | 0.04 (−0.01, 0.10) | 0.07 (−0.01, 0.15) | 0.48 |
| Cognitive ability (s.d.) | 0.07 (0.04, 0.10) | 0.05 (0.00, 0.10) | 0.43 | −0.20 (−0.27, −0.13) | −0.08 (−0.18, 0.01) | 0.020 |
| Depressive symptoms (s.d.) | −0.02 (−0.04, 0.00) | −0.02 (−0.06, 0.02) | 0.94 | 0.04 (−0.01, 0.09) | −0.01 (−0.09, 0.07) | 0.18 |
| Educational attainment (s.d.) | 0.06 (0.04, 0.07) | 0.02 (−0.01, 0.04) | 0.0012 | −0.19 (−0.22, −0.16) | −0.05 (−0.09, −0.01) | <0.001 |
| Ever smoking (risk difference) | −0.01 (−0.01, 0.00) | 0.01 (−0.01, 0.02) | 0.058 | 0.07 (0.05, 0.08) | 0.04 (0.02, 0.07) | 0.065 |
| FEV1 (s.d.) | −0.02 (−0.04, 0.00) | −0.03 (−0.07, 0.01) | 0.67 | −0.17 (−0.22, −0.12) | −0.17 (−0.25, −0.09) | 1.00 |
| FEV1FVC (s.d.) | 0.02 (−0.00, 0.04) | 0.02 (−0.02, 0.06) | 0.96 | −0.02 (−0.06, 0.03) | −0.02 (−0.09, 0.05) | 0.87 |
| HbA1c (s.d.) | −0.00 (−0.02, 0.02) | 0.02 (−0.02, 0.06) | 0.21 | 0.15 (0.11, 0.20) | 0.14 (0.07, 0.22) | 0.77 |
| HDL-cholesterol (s.d.) | −0.01 (−0.03, 0.01) | −0.02 (−0.05, 0.00) | 0.31 | −0.32 (−0.36, −0.29) | −0.33 (−0.38, −0.28) | 0.79 |
| Low-density lipoprotein-cholesterol (s.d.) | −0.05 (−0.06, −0.03) | −0.03 (−0.06, −0.00) | 0.31 | 0.02 (−0.02, 0.06) | 0.02 (−0.03, 0.08) | 0.86 |
| Age at menarche (years) | 0.09 (0.04, 0.13) | 0.07 (−0.00, 0.14) | 0.63 | −0.62 (−0.71, −0.52) | −0.62 (−0.76, −0.49) | 0.93 |
| Age at menopause (years) | −0.17 (−0.37, 0.02) | −0.15 (−0.51, 0.20) | 0.89 | −0.49 (−0.93, −0.05) | −0.35 (−1.02, 0.31) | 0.72 |
| Neuroticism (s.d.) | −0.02 (−0.03, 0.00) | 0.01 (−0.02, 0.04) | 0.14 | 0.00 (−0.04, 0.04) | −0.03 (−0.09, 0.03) | 0.28 |
| Physical activity (risk difference) | −0.00 (−0.01, 0.01) | −0.01 (−0.03, 0.00) | 0.12 | −0.04 (−0.05, −0.02) | −0.03 (−0.06, 0.00) | 0.63 |
| SBP (mmHg) | −0.77 (−1.04, −0.50) | −0.64 (−1.11, −0.17) | 0.56 | 3.17 (2.57, 3.78) | 3.21 (2.33, 4.10) | 0.93 |
| Triglycerides (s.d.) | −0.02 (−0.03, −0.00) | 0.01 (−0.02, 0.04) | 0.051 | 0.27 (0.23, 0.31) | 0.27 (0.21, 0.33) | 0.96 |
| Waist-to-hip ratio adjusted for BMI (WHR×100) | 0.00 (0.00, 0.00) | 0.00 (0.00, 0.00) | 0.26 | 0.01 (0.01, 0.01) | 0.01 (0.01, 0.01) | 0.29 |
| Wellbeing (s.d.) | 0.01 (−0.01, 0.03) | −0.01 (−0.04, 0.03) | 0.39 | −0.05 (−0.09, −0.00) | −0.05 (−0.12, 0.01) | 0.85 |
| eGFR | −0.67 (−0.92, −0.43) | −0.86 (−1.28, −0.45) | 0.36 | −0.10 (−0.64, 0.44) | 0.32 (−0.47, 1.11) | 0.22 |

Table 1 contains population MR and WS-MR estimates of height and BMI on 23 phenotypes. Units are presented in terms of a standard deviation increase in height or BMI. Difference (Diff) P values refer to evidence of differences between population and within-sibship estimates which were derived using a difference-of-two-means test with standard errors derived using leave-one-out jackknifing.

biased by population stratification in the Genetic Investigation of ANthropometric Traits (GIANT) consortium[26–28]. Previous evidence for adaptation on height using siblings in UK Biobank was suggestive of some adaptation, but statistically inconclusive[26]. Here, using within-sibship GWAS estimates from a larger (~4-fold) sample of siblings, we found strong evidence of polygenic adaptation on increased height and some evidence of adaptation on number of children and HDL-cholesterol. We anticipate that future studies on human evolution will benefit from using large within-family datasets such as our resource.

Within-family GWAS are limited by both available family data and statistical inefficiency (homozygosity within families). To help address this issue, future population-based biobanks could recruit the partners, siblings and offspring of study participants. In contrast, conventional population GWAS designs sampling unrelated individuals are likely to be the optimal approach to maximize statistical power for discovery GWAS for genetic

associations. Indeed, we found that many genotype–phenotype associations from population GWAS models were also observed in within-sibship GWASs, albeit sometimes attenuated towards zero. A notable limitation of within-sibship models is that they do not control for indirect genetic effects of siblings, that is, effects of sibling genotypes on the shared environment. Sibling effects have been estimated to be modest compared with parental effects[6,48] but could have impacted our GWAS estimates. Another limitation is that while assortative mating is unlikely to affect within-sibship GWAS estimates, it can bias within-sibship estimates of heritability downwards[49] and so may have affected our LDSC SNP heritability and genetic correlation estimates. However, the within-sibship shrinkage in GWAS estimates and LDSC heritability estimates were largely consistent, suggesting any such bias is unlikely to have impacted our conclusions. Our findings are also limited to adult phenotypes. Within-family GWAS (for example, using parent–offspring trios) could use data from children to

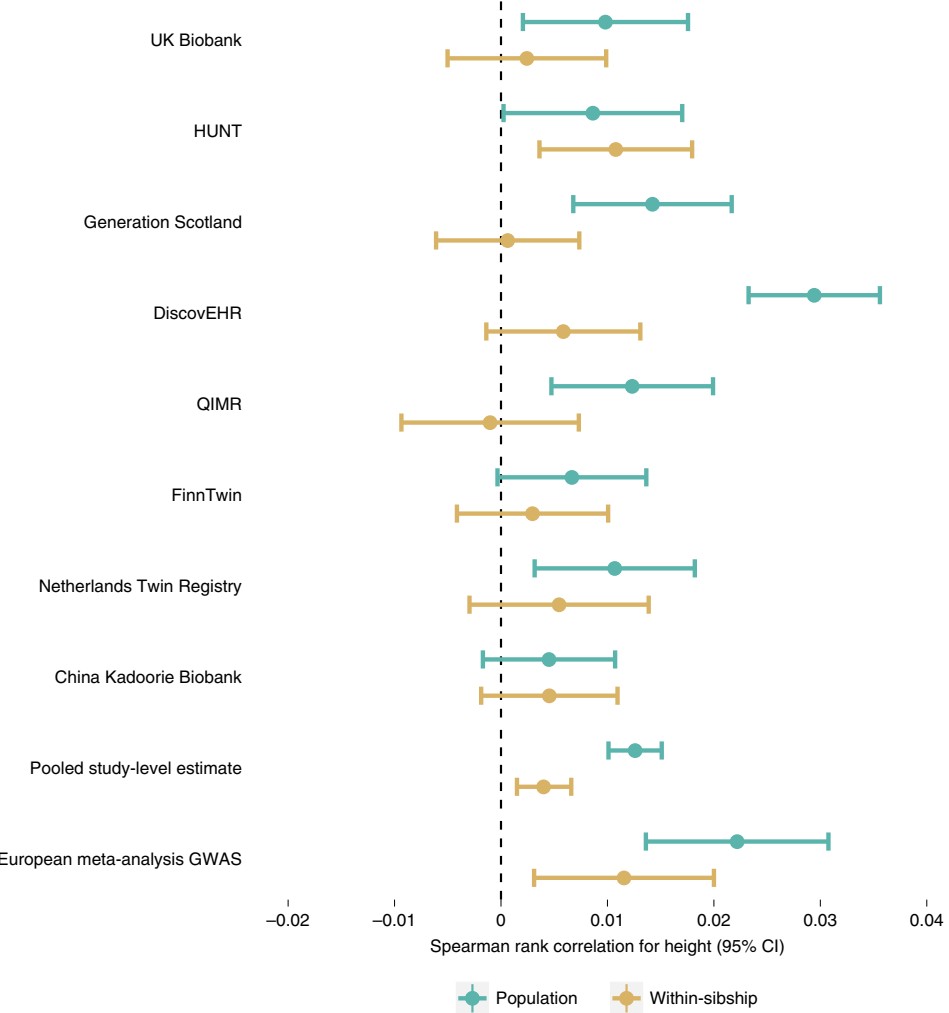

**Fig. 7 | Spearman rank correlation estimates and corresponding 95% CIs between tSDS (SDS aligned with height-increasing alleles) and absolute height *Z* scores.** Positive correlations indicate evidence of historical positive selection on height-increasing alleles. The pooled estimate is a meta-analysis of the correlation estimates from the individual studies shown above while the European meta-analysis estimate is the correlation estimate using the meta-analysis GWAS data. The number of individuals in the meta-analysis estimate was *n* = 149,174 with the sample sizes for the displayed individual studies ranging from *n* = 40,068 in UK Biobank to 4,708 in the Netherlands Twin Register. Further information on available height data in each phenotype is contained in Supplementary Table 1. QIMR, Queensland Institute of Medical Research.

evaluate if childhood phenotypes are more strongly affected by indirect genetic effects.

## Online content

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

**K. Paige Harden** [45], **W. David Hill** [46,47], **Amanda Hughes** [1,2], **Shona M. Kerr** [19], **Yongkang Kim** [26],
**Hyeokmoon Kweon** [48], **Antti Latvala** [10,49], **Deborah A. Lawlor** [1,2,50], **Liming Li** [51], **Kuang Lin** [36],
**Per Magnus** [52], **Patrik K. E. Magnusson** [15], **Travis T. Mallard** [45], **Pekka Martikainen** [53,54,55],

Melinda C. Mills[56], Pål Rasmus Njølstad[57,58], John D. Overton[30], Nancy L. Pedersen[15], David J. Porteous[34], Jeffrey Reid[30], Karri Silventoinen[53], Melissa C. Southey[18,59,60], Camilla Stoltenberg[43,61], Elliot M. Tucker-Drob[45], Margaret J. Wright[62], Social Science Genetic Association Consortium*, Within Family Consortium, John K. Hewitt[25,26], Matthew C. Keller[25,26], Michael C. Stallings[25,26], James J. Lee[23], Kaare Christensen[21,22,63], Sharon L. R. Kardia[20], Patricia A. Peyser[20], Jennifer A. Smith[20,64], James F. Wilson[19,35], John L. Hopper[16], Sara Hägg[15], Tim D. Spector[13], Jean-Baptiste Pingault[12,65], Robert Plomin[12], Alexandra Havdahl[11,29,38], Meike Bartels[3], Nicholas G. Martin[44], Sven Oskarsson[6], Anne E. Justice[5], Iona Y. Millwood[36,37], Kristian Hveem[4,66], Øyvind Naess[42,43], Cristen J. Willer[4,67,68], Bjørn Olav Åsvold[4,66,69], Philipp D. Koellinger[48,70], Jaakko Kaprio[10], Sarah E. Medland[7,9,71], Robin G. Walters[36,37], Daniel J. Benjamin[72,73,74], Patrick Turley[75,76], David M. Evans[1,77,78], George Davey Smith[1,2], Caroline Hayward[19], Ben Brumpton[1,4,66,154]✉, Gibran Hemani[1,2,154]✉ and Neil M. Davies[1,2,4,154]✉

[1]Medical Research Council Integrative Epidemiology Unit at the University of Bristol, Bristol, UK. [2]Population Health Sciences, Bristol Medical School, University of Bristol, Bristol, UK. [3]Department of Biological Psychology, Netherlands Twin Register, Vrije Universiteit, Amsterdam, the Netherlands. [4]K.G. Jebsen Center for Genetic Epidemiology, Department of Public Health and Nursing, NTNU, Norwegian University of Science and Technology, Trondheim, Norway. [5]Department of Population Health Sciences, Geisinger Health, Danville, PA, USA. [6]Department of Government, Uppsala University, Uppsala, Sweden. [7]Psychiatric Genetics, QIMR Berghofer Medical Research Institute, Brisbane, Australia. [8]School of Biomedical Sciences, Queensland University of Technology, Brisbane, Australia. [9]Faculty of Medicine, University of Queensland, Brisbane, Australia. [10]Institute for Molecular Medicine FIMM, University of Helsinki, Helsinki, Finland. [11]PROMENTA Research Center, Department of Psychology, University of Oslo, Oslo, Norway. [12]Social Genetic & Developmental Psychiatry Centre, Institute of Psychiatry, Psychology & Neuroscience, King's College London, London, UK. [13]Department of Twin Research and Genetic Epidemiology, King's College London, London, UK. [14]NIHR Biomedical Research Centre at Guy's and St Thomas' Foundation Trust, London, UK. [15]Department of Medical Epidemiology and Biostatistics, Karolinska Institutet, Stockholm, Sweden. [16]Centre for Epidemiology and Biostatistics, Melbourne School of Population and Global Health, The University of Melbourne, Parkville, Victoria, Australia. [17]Centre for Cancer Genetic Epidemiology, Department of Public Health and Primary Care, University of Cambridge, Cambridge, UK. [18]Precision Medicine, School of Clinical Sciences at Monash Health, Monash University, Clayton, Victoria, Australia. [19]MRC Human Genetics Unit, Institute of Genetics and Cancer, University of Edinburgh, Western General Hospital, Edinburgh, UK. [20]Department of Epidemiology, School of Public Health, University of Michigan, Ann Arbor, MI, USA. [21]The Danish Twin Registry, Department of Public Health, University of Southern Denmark, Odense, Denmark. [22]Department of Clinical Genetics, Odense University Hospital, Odense, Denmark. [23]Department of Psychology, University of Minnesota, Minneapolis, MN, USA. [24]Department of Psychology, University of California, Riverside, Riverside, CA, USA. [25]Department of Psychology & Neuroscience, University of Colorado at Boulder, Boulder, CO, USA. [26]Institute for Behavioral Genetics, University of Colorado at Boulder, Boulder, CO, USA. [27]NORMENT Centre, University of Oslo, Oslo, Norway. [28]Division of Mental Health and Addiction, Oslo University Hospital, Oslo, Norway. [29]Department of Mental Disorders, Norwegian Institute of Public Health, Oslo, Norway. [30]Regeneron Genetics Center, Tarrytown, NY, USA. [31]BioMarin Pharmaceutical Inc., Novato, CA, USA. [32]Biomedical and Translational Informatics, Geisinger Health, Danville, PA, USA. [33]Amsterdam Public Health (APH) and Amsterdam Reproduction and Development (AR&D), Amsterdam, the Netherlands. [34]Centre for Genomic and Experimental Medicine, Institute of Genetics & Cancer, University of Edinburgh, Western General Hospital, Edinburgh, UK. [35]Centre for Global Health, Usher Institute, University of Edinburgh, Edinburgh, UK. [36]Nuffield Department of Population Health, University of Oxford, Oxford, UK. [37]MRC Population Health Research Unit, University of Oxford, Oxford, UK. [38]Nic Waals Institute, Lovisenberg Diaconal Hospital, Oslo, Norway. [39]Department of Public Health, Aarhus University, Aarhus, Denmark. [40]Department of Ecology & Evolutionary Biology, University of Colorado at Boulder, Boulder, CO, USA. [41]Amsterdam Public Health Research Institute, Amsterdam UMC, Amsterdam, the Netherlands. [42]Institute of Health and Society, University of Oslo, Oslo, Norway. [43]Norwegian Institute of Public Health, Oslo, Norway. [44]Department of Genetics and Computational Biology, QIMR Berghofer Medical Research Institute, Brisbane, Queensland, Australia. [45]Department of Psychology and Population Research Center, University of Texas at Austin, Austin, TX, USA. [46]Lothian Birth Cohorts Group, Department of Psychology, University of Edinburgh, Edinburgh, UK. [47]Department of Psychology, University of Edinburgh, Edinburgh, UK. [48]Department of Economics, School of Business and Economics, Vrije Universiteit Amsterdam, Amsterdam, the Netherlands. [49]Institute of Criminology and Legal Policy, Faculty of Social Sciences, University of Helsinki, Helsinki, Finland. [50]Bristol NIHR Biomedical Research Centre, Bristol, UK. [51]Department of Epidemiology and Biostatistics, School of Public Health, Peking University Health Science Center, Beijing, China. [52]Centre for Fertility and Health, Norwegian Institute of Public Health, Skøyen, Oslo, Norway. [53]Population Research Unit, Faculty of Social Sciences, University of Helsinki, Helsinki, Finland. [54]The Max Planck Institute for Demographic Research, Rostock, Germany. [55]Department of Public Health Sciences, Stockholm University, Stockholm, Sweden. [56]Leverhulme Centre for Demographic Science, University of Oxford, Oxford, UK. [57]Department of Clinical Science, University of Bergen, Bergen, Norway. [58]Children and Youth Clinic, Haukeland University Hospital, Bergen, Norway. [59]Department of Clinical Pathology, Melbourne Medical School, The University of Melbourne, Melbourne, Victoria, Australia. [60]Cancer Epidemiology Division, Cancer Council Victoria, Melbourne, Victoria, Australia. [61]Department of Global Public Health and Primary Care, University of Bergen, Bergen, Norway. [62]Queensland Brain Institute, The University of Queensland, Brisbane, Queensland, Australia. [63]Department of Clinical Biochemistry and Pharmacology, Odense University Hospital, Odense, Denmark. [64]Survey Research Center, Institute for Social Research, University of Michigan, Ann Arbor, MI, USA. [65]Department of Clinical, Educational and Health Psychology, University College London, London, UK. [66]HUNT Research Center, Department of Public Health and Nursing, NTNU, Norwegian University of Science and Technology, Levanger, Norway. [67]Department of Internal Medicine: Cardiology, University of Michigan, Ann Arbor, MI, USA. [68]Department of Computational Medicine and Bioinformatics, University of Michigan, Ann Arbor, MI, USA. [69]Department of Endocrinology, Clinic of Medicine, St. Olavs Hospital, Trondheim University Hospital, Trondheim, Norway. [70]La Follette School of Public Affairs, University of Wisconsin-Madison, Madison, WI, USA. [71]School of Psychology, University of Queensland, Brisbane, Queensland, Australia. [72]UCLA Anderson School of Management, Los Angeles, CA, USA. [73]Human Genetics Department, UCLA David Geffen School of Medicine, Gonda (Goldschmied) Neuroscience and Genetics Research Center, Los Angeles, CA, USA. [74]National Bureau of Economic Research, Cambridge, MA, USA. [75]Center for Economic and Social Research, University of Southern California, Los

Angeles, CA, USA. [76]Department of Economics, University of Southern California, Los Angeles, CA, USA. [77]University of Queensland Diamantina Institute, University of Queensland, Brisbane, Queensland, Australia. [78]Institute for Molecular Bioscience, University of Queensland, Brisbane, Queensland, Australia. [154]These authors contributed equally: Ben Brumpton, Gibran Hemani, Neil M. Davies. *Lists of authors and their affiliations appear at the end of the paper. ✉e-mail: laurence.howe@bristol.ac.uk; ben.brumpton@ntnu.no; g.hemani@bristol.ac.uk; neil.davies@bristol.ac.uk

## Social Science Genetic Association Consortium

**Hyeokmoon Kweon**[79]**, Philipp D. Koellinger**[79,80]**, Daniel J. Benjamin**[81,82,83] **and Patrick Turley**[84,85]

[79]Department of Economics, School of Business and Economics, Vrije Universiteit Amsterdam, Amsterdam, the Netherlands. [80]La Follette School of Public Affairs, University of Wisconsin-Madison, Madison, WI, USA. [81]UCLA Anderson School of Management, Los Angeles, CA, USA. [82]Human Genetics Department, UCLA David Geffen School of Medicine, Gonda (Goldschmied) Neuroscience and Genetics Research Center, Los Angeles, CA, USA. [83]National Bureau of Economic Research, Cambridge, MA, USA. [84]Center for Economic and Social Research, University of Southern California, Los Angeles, Los Angeles, CA, USA. [85]Department of Economics, University of Southern California, Los Angeles, Los Angeles, CA, USA.

## Within Family Consortium

**Laurence J. Howe**[86,87]**, Michel G. Nivard**[88]**, Tim T. Morris**[86,87]**, Ailin F. Hansen**[89]**, Humaira Rasheed**[86,89]**, Yoonsu Cho**[86,87]**, Geetha Chittoor**[90]**, Rafael Ahlskog**[91]**, Penelope A. Lind**[92,93,94]**, Teemu Palviainen**[95]**, Matthijs D. van der Zee**[88]**, Rosa Cheesman**[96,97]**, Massimo Mangino**[98,99]**, Yunzhang Wang**[100]**, Shuai Li**[101,102,103]**, Lucija Klaric**[104]**, Scott M. Ratliff**[105]**, Lawrence F. Bielak**[105]**, Marianne Nygaard**[106,107]**, Alexandros Giannelis**[108]**, Emily A. Willoughby**[108]**, Chandra A. Reynolds**[109]**, Jared V. Balbona**[110,111]**, Ole A. Andreassen**[112,113]**, Helga Ask**[114]**, Dorret I. Boomsma**[88,115]**, Archie Campbell**[116]**, Harry Campbell**[117]**, Zhengming Chen**[118,119]**, Paraskevi Christofidou**[98]**, Elizabeth Corfield**[114,120]**, Christina C. Dahm**[121]**, Deepika R. Dokuru**[110,111]**, Luke M. Evans**[111,122]**, Eco J. C. de Geus**[88,123]**, Sudheer Giddaluru**[124,125]**, Scott D. Gordon**[126]**, K. Paige Harden**[127]**, W. David Hill**[128,129]**, Amanda Hughes**[86,87]**, Shona M. Kerr**[104]**, Yongkang Kim**[111]**, Antti Latvala**[95,130]**, Deborah A. Lawlor**[86,87,131]**, Liming Li**[132]**, Kuang Lin**[118]**, Per Magnus**[133]**, Patrik K. E. Magnusson**[100]**, Travis T. Mallard**[127]**, Pekka Martikainen**[134,135,136]**, Melinda C. Mills**[137]**, Pål Rasmus Njølstad**[138,139]**, Nancy L. Pedersen**[100]**, David J. Porteous**[116]**, Karri Silventoinen**[134]**, Melissa C. Southey**[103,140,141]**, Camilla Stoltenberg**[125,142]**, Elliot M. Tucker-Drob**[127]**, Margaret J. Wright**[143]**, John K. Hewitt**[110,111]**, Matthew C. Keller**[110,111]**, Michael C. Stallings**[110,111]**, James J. Lee**[108]**, Kaare Christensen**[106,107,144]**, Sharon L. R. Kardia**[105]**, Patricia A. Peyser**[105]**, Jennifer A. Smith**[105,145]**, James F. Wilson**[104,117]**, John L. Hopper**[101]**, Sara Hägg**[100]**, Tim D. Spector**[98]**, Jean-Baptiste Pingault**[97,146]**, Robert Plomin**[97]**, Alexandra Havdahl**[96,114,120]**, Meike Bartels**[88]**, Nicholas G. Martin**[126]**, Sven Oskarsson**[91]**, Anne E. Justice**[90]**, Iona Y. Millwood**[118,119]**, Kristian Hveem**[89,147]**, Øyvind Naess**[124,125]**, Cristen J. Willer**[89,148,149]**, Bjørn Olav Åsvold**[89,147,150]**, Jaakko Kaprio**[95]**, Sarah E. Medland**[92,94,151]**, Robin G. Walters**[118,119]**, David M. Evans**[86,152,153]**, George Davey Smith**[86,87]**, Caroline Hayward**[104]**, Ben Brumpton**[86,89,147]**, Gibran Hemani**[86,87] **and Neil M. Davies**[86,87,89]

[86]Medical Research Council Integrative Epidemiology Unit at the University of Bristol, Bristol, UK. [87]Population Health Sciences, Bristol Medical School, University of Bristol, Bristol, UK. [88]Department of Biological Psychology, Netherlands Twin Register, Vrije Universiteit, Amsterdam, the Netherlands. [89]K.G. Jebsen Center for Genetic Epidemiology, Department of Public Health and Nursing, NTNU, Norwegian University of Science and Technology, Trondheim, Norway. [90]Department of Population Health Sciences, Geisinger Health, Danville, PA, USA. [91]Department of Government, Uppsala University, Uppsala, Sweden. [92]Psychiatric Genetics, QIMR Berghofer Medical Research Institute, Brisbane, Queensland, Australia. [93]School of Biomedical Sciences, Queensland University of Technology, Brisbane, Queensland, Australia. [94]Faculty of Medicine, University of Queensland, Brisbane, Queensland, Australia. [95]Institute for Molecular Medicine FIMM, University of Helsinki, Helsinki, Finland. [96]PROMENTA Research Center, Department of Psychology, University of Oslo, Oslo, Norway. [97]Social Genetic & Developmental Psychiatry Centre, Institute of Psychiatry, Psychology & Neuroscience, King's College London, London, UK. [98]Department of Twin Research and Genetic Epidemiology, King's College London, London, UK. [99]NIHR Biomedical Research Centre at Guy's and St Thomas' Foundation Trust, London, UK. [100]Department of Medical Epidemiology and Biostatistics, Karolinska Institutet, Stockholm, Sweden. [101]Centre for Epidemiology and Biostatistics, Melbourne School of Population and Global Health, The University of Melbourne, Parkville, Victoria, Australia. [102]Centre for Cancer Genetic Epidemiology, Department of Public Health and Primary Care, University of Cambridge, Cambridge, UK. [103]Precision Medicine, School of Clinical Sciences at Monash Health, Monash University, Clayton, Victoria, Australia. [104]MRC Human Genetics Unit, Institute of Genetics and Cancer, University of Edinburgh, Western General Hospital, Edinburgh, UK. [105]Department of Epidemiology, School of Public Health, University of Michigan, Ann Arbor, MI, USA. [106]The Danish Twin Registry, Department of Public Health, University of Southern Denmark, Odense, Denmark. [107]Department of

Clinical Genetics, Odense University Hospital, Odense, Denmark. [108]Department of Psychology, University of Minnesota, Minneapolis, MN, USA. [109]Department of Psychology, University of California, Riverside, Riverside, CA, USA. [110]Department of Psychology & Neuroscience, University of Colorado at Boulder, Boulder, CO, USA. [111]Institute for Behavioral Genetics, University of Colorado at Boulder, Boulder, CO, USA. [112]NORMENT Centre, University of Oslo, Oslo, Norway. [113]Division of Mental Health and Addiction, Oslo University Hospital, Oslo, Norway. [114]Department of Mental Disorders, Norwegian Institute of Public Health, Oslo, Norway. [115]Amsterdam Public Health (APH) and Amsterdam Reproduction and Development (AR&D), Amsterdam, the Netherlands. [116]Centre for Genomic and Experimental Medicine, Institute of Genetics & Cancer, University of Edinburgh, Western General Hospital, Edinburgh, UK. [117]Centre for Global Health, Usher Institute, University of Edinburgh, Edinburgh, UK. [118]Nuffield Department of Population Health, University of Oxford, Oxford, UK. [119]MRC Population Health Research Unit, University of Oxford, Oxford, UK. [120]Nic Waals Institute, Lovisenberg Diaconal Hospital, Oslo, Norway. [121]Department of Public Health, Aarhus University, Aarhus, Denmark. [122]Department of Ecology & Evolutionary Biology, University of Colorado at Boulder, Boulder, CO, USA. [123]Amsterdam Public Health Research Institute, Amsterdam UMC, Amsterdam, the Netherlands. [124]Institute of Health and Society, University of Oslo, Oslo, Norway. [125]Norwegian Institute of Public Health, Oslo, Norway. [126]Department of Genetics and Computational Biology, QIMR Berghofer Medical Research Institute, Brisbane, Queensland, Australia. [127]Department of Psychology and Population Research Center, University of Texas at Austin, Austin, TX, USA. [128]Lothian Birth Cohorts Group, Department of Psychology, University of Edinburgh, Edinburgh, UK. [129]Department of Psychology, University of Edinburgh, Edinburgh, UK. [130]Institute of Criminology and Legal Policy, Faculty of Social Sciences, University of Helsinki, Helsinki, Finland. [131]Bristol NIHR Biomedical Research Centre, Bristol, UK. [132]Department of Epidemiology and Biostatistics, School of Public Health, Peking University Health Science Center, Beijing, China. [133]Centre for Fertility and Health, Norwegian Institute of Public Health, Skøyen, Oslo, Norway. [134]Population Research Unit, Faculty of Social Sciences, University of Helsinki, Helsinki, Finland. [135]The Max Planck Institute for Demographic Research, Rostock, Germany. [136]Department of Public Health Sciences, Stockholm University, Stockholm, Sweden. [137]Leverhulme Centre for Demographic Science, University of Oxford, Oxford, UK. [138]Department of Clinical Science, University of Bergen, Bergen, Norway. [139]Children and Youth Clinic, Haukeland University Hospital, Bergen, Norway. [140]Department of Clinical Pathology, Melbourne Medical School, The University of Melbourne, Melbourne, Victoria, Australia. [141]Cancer Epidemiology Division, Cancer Council Victoria, Melbourne, Victoria, Australia. [142]Department of Global Public Health and Primary Care, University of Bergen, Bergen, Norway. [143]Queensland Brain Institute, The University of Queensland, Brisbane, Queensland, Australia. [144]Department of Clinical Biochemistry and Pharmacology, Odense University Hospital, Odense, Denmark. [145]Survey Research Center, Institute for Social Research, University of Michigan, Ann Arbor, Michigan, USA. [146]Department of Clinical, Educational and Health Psychology, University College London, London, UK. [147]HUNT Research Center, Department of Public Health and Nursing, NTNU, Norwegian University of Science and Technology, Levanger, Norway. [148]Department of Internal Medicine: Cardiology, University of Michigan, Ann Arbor, MI, USA. [149]Department of Computational Medicine and Bioinformatics, University of Michigan, Ann Arbor, MI, USA. [150]Department of Endocrinology, Clinic of Medicine, St. Olavs Hospital, Trondheim University Hospital, Trondheim, Norway. [151]School of Psychology, University of Queensland, Brisbane, Queensland, Australia. [152]University of Queensland Diamantina Institute, University of Queensland, Brisbane, Queensland, Australia. [153]Institute for Molecular Bioscience, University of Queensland, Brisbane, Queensland, Australia.

## Methods

**Study participants.** Nineteen cohorts contributed data to the overall study (Supplementary Table 1). These cohorts were selected on the basis of having at least 500 genotyped siblings (an individual with 1 or more siblings in the study sample) with at least 1 of the 25 phenotypes that were analyzed in the study. Phenotypes were selected based on available data and to include a range of different phenotypes. Detailed information on genotype data, quality control and imputation processes are provided in the Cohort Descriptions in the Supplementary Materials. Individual cohorts defined each phenotype based on suggested definitions from an analysis plan (see the Phenotype Definitions in the Supplementary Materials).

**GWAS analyses.** GWAS analyses were performed uniformly across individual studies using automated scripts and a preregistered analysis plan (https://github.com/LaurenceHowe/SiblingGWAS). Scripts checked strand alignment, imputation scores and allele frequencies for the genetic data as well as missingness for covariates and phenotypes. Scripts also summarized covariates and phenotypes and set phenotypes to missing for sibships if only one individual in the sibship had nonmissing phenotype data. To harmonize variants for meta-analysis, genetic variants were renamed in a format including information on chromosome, base pair and polymorphism type (SNP or INDEL: insertion or deletion). The automated pipeline restricted analyses to common genetic variants (minor allele frequency (MAF) > 0.01) and removed poorly imputed variants (INFO: information score < 0.3). Analyses were restricted to include individuals in a sibship, that is, a group of two or more full siblings in the study. Monozygotic twins were included if they had an additional sibling in the study.

GWAS analyses involved fitting both population and within-sibship models to the same samples. The population model is synonymous with a conventional principal component adjusted model, and was fit using linear regression in R (v.3.5.1). The within-sibship model is an extension of the population model including the mean sibship genotype (the mean genotype of siblings in each sibship) as a covariate to account for family structure, with each individual's genotype centered around the mean sibship genotype[7,14]. Age, sex and up to 20 principal components (10 principal components were included in smaller studies at the discretion of study co-authors) were included as covariates in both models. The pipeline used imputed 'best guess' genotype calls rather than dosage data.

For individual $j$ in sibship $i$ with $ni > 2$ siblings:

Population model:

$$\text{Phenotype}_{ij} \sim G_{ij} + \text{Sex}_{ij} + \text{Age}_{ij} + \text{PC1}_{ij} + \text{PC20}_{ij}$$

Within-sibship model:

$$\text{Phenotype}_{ij} \sim G_{ij}^C + G_i^F + \text{Sex}_{ij} + \text{Age}_{ij} + \text{PC1}_{ij} + \text{PC20}_{ij}$$

where $G_i^F = \frac{\sum_j^n G_{ij}}{n}$ and $G_{ij}^C = G_{ij} - G_i^F$

$G_{ij}$, genotype of sibling $j$ in sibship $i$; $G_i^F$, mean family genotype for sibship $i$ over $n$ siblings; $G_{ij}^C$, genotype of sibling $j$ in sibship $i$ centered around $G_i^F$; PC, principal component.

Standard errors from both estimators were clustered over families at the sibling level to account for nonrandom clustering of siblings within families. Note that this clustering accounts for sibling relationships but does not account for further relatedness present in each sample. For example, a sibling pair could be related to another sibling pair (that is, two pairs of siblings who are first-cousins). We performed simulations, described below, confirming that such relatedness can lead to underestimating standard errors in the population model and has no effect on the standard errors of the within-sibship model.

GWAS models were performed in individual studies, harmonized and then meta-analyzed for each phenotype using a fixed-effects model in METAL[50] with population and within-sibship data meta-analyzed separately. We performed meta-analyses using only samples of European ancestry. We used data from 13,856 individuals from the China Kadoorie Biobank separately in downstream analyses. Information on sample sizes for individual phenotypes is contained in Supplementary Table 2. Information on further quality control performed before meta-analysis is detailed in the Supplementary Methods.

**Meta-analysis.** Phenotypes were harmonized between studies using phenotypic summary data on means and standard deviations. GWAS of study-specific phenotypes that did not conform to analysis plan definitions (for example, binary instead of continuous) were excluded from meta-analyses. GWAS presented in different continuous units (for example, not standardized) were transformed before meta-analysis by dividing association estimates and standard errors by the standard deviation of the phenotype as measured in the cohort. Meta-analyses for 25 phenotypes were performed using a fixed-effects model in METAL[50].

**Within-sibship and population-based GWAS comparison.** *Overview.* We hypothesized that the within-sibship estimates would differ compared with population-based estimates due to the exclusion of effects from demographic and familial pathways. In general, these effects have been shown to inflate (rather than shrink) population-based estimates, so we estimated within-sibship shrinkage

(the % difference from population to within-sibship estimates). To estimate this shrinkage, we required estimates of the associations with a phenotype from each within-sibship and population-based analysis that was not affected by winner's curse. Hence, we adopted a strategy where we used an independent reference dataset to select the variants associated with a phenotype. Using the meta-analysis results to obtain association estimates for these variants, we generated summary-based weighted scores of those association estimates in the within-sibship and population-based analyses and estimated the ratio of those scores. We used the UK Biobank dataset excluding sibling data as the independent reference dataset.

*GWAS in independent reference discovery dataset.* We performed GWAS in an independent sample of UK Biobank (excluding siblings) for each phenotype using a linear mixed model as implemented in BOLT-LMM[51]. We started with a sample of 463,006 individuals of 'European' ancestry derived using in-house $k$-means cluster analysis performed using the first four principal components provided by UK Biobank with standard exclusions also removed[52]. To remove sample overlap, we then excluded the sibling sample ($N = 40,276$), resulting in a final sample of 422,730 individuals. To model population structure in the sample, we used 143,006 directly genotyped SNPs, obtained after filtering on MAF > 0.01; genotyping rate > 0.015; Hardy–Weinberg equilibrium $P < 0.0001$; and LD pruning to an $r^2$ threshold of 0.1 using PLINK v.2.0 (ref. [53]). Age and sex were included in the model as covariates.

All 25 phenotypes (conforming to our phenotype definition) were available in UK Biobank data except for a continuous measure of depressive symptoms. For depressive symptoms, we performed a GWAS of binary depression which was excluded from the meta-analysis (see definition in Supplementary Materials). Using the BOLT-LMM UK Biobank GWAS summary data, we performed strict LD clumping in PLINK v.2.0 (ref. [53]) ($r^2 < 0.001$, physical distance threshold = 10,000 kb) using the 1000 Genomes Phase 3 EUR reference panel[54] to generate independent variants associated with each phenotype at genome-wide significance ($P < 5 \times 10^{-8}$) and at a more liberal threshold ($P < 1 \times 10^{-5}$).

*Summary-based weighted scores.* For a particular phenotype the sets of independent variants obtained from the independent UK Biobank GWAS were used to generate a summary-based weighted score using an inverse variance weighting (IVW) approach[55,56]:

$$S = \frac{\sum_k^M \frac{w_k \beta_k}{\sigma_k^2}}{\sum_k^M \frac{w_k^2}{\sigma_k^2}}$$

with standard error

$$\sigma_S = \sqrt{\frac{1}{\sum_k^M \frac{w_k^2}{\sigma_k^2}}}$$

Here, the score $S$ represents the weighted average of the association estimates of the $M$ variants on a phenotype, where $\beta$ and $\sigma$ represent the beta coefficients and standard errors from the within-sibship (W) or population-based (P) meta-analysis results. The discovery association estimates from the UK Biobank GWAS were used as weights ($w$). The set of $M$ variants were determined using either the genome-wide significance (G) or the more liberal threshold (L). Hence, depending on which model is used to determine the association estimates and which set of SNPs are used, four scores can be calculated for each phenotype—$S_{P,G}$, $S_{P,L}$, $S_{W,G}$ and $S_{W,L}$.

These sets of scores were obtained for each of the 25 phenotypes with weights for binary depression used as a substitute for depressive symptoms because a suitable measure was unavailable in UK Biobank. The scores were strongly associated with the set of phenotypes in the meta-analysis data based on determining $P$ values from their $Z$ scores. The $S_{W,L}$ scores were nominally associated at $P < 0.05$ for 24 of 25 (exception: number of children) of the phenotypes, with the $S_{P,L}$ scores associated with all 25 phenotypes at this threshold (Supplementary Table 9).

*Estimating shrinkage from population to within-sibship estimates.* We used the within-sibship and population-based scores to calculate the average shrinkage ($\delta$, that is, proportion decrease) of genetic variant–phenotype associations

$$\delta = 1 - \frac{S_{W,}}{S_{P,}}$$

The standard errors of $\delta$ could be estimated using the delta method as below using the standard errors of the scores and the covariance between the scores $\text{Cov}(S_w, S_P)$:

$$\sigma_\delta \sim \left(\frac{S_{W,}}{S_{P,}}\right) \sqrt{\left(\frac{\sigma_{S_{W,}}^2}{S_{W,}^2} + \frac{\sigma_{S_{P,}}^2}{S_{P,}^2}\right) - \frac{2\text{Cov}(S_w, S_{P,})}{S_{W,} S_{P,}}}$$

However, we do not have an estimate of this covariance term because the two GWAS were fit in separate regression models. We therefore used the jackknife to estimate $\sigma_\delta$. For a score of $M$ variants, we removed each variant in turn and repeated IVW and shrinkage analyses as above, extracting the shrinkage point estimate in each of the $M$ iterations. We then calculated $\sigma_\delta$ as follows:

$$\sigma_\delta = \sqrt{\frac{M-1}{M} \sum_k^M (\sigma_{\delta,k} - \mu)^2}$$

where

$$\mu = \frac{\sum_k^M \sigma_{\delta,k}}{M}$$

As a sensitivity analysis, we investigated the effects of positive covariance between the population and within-sibship models on the shrinkage standard errors using individual-level participant data from UK Biobank. Analyzing shrinkage on height, we used seemingly unrelated regression to estimate the covariance term between the population and within-sibship estimators. We found that standard errors for shrinkage estimates decreased by around 15% when the covariance was modeled (Supplementary Table 10). Seemingly unrelated regression standard errors were consistent with the jackknife approach standard errors.

As the primary analysis, we reported shrinkage results using the liberal threshold ($P < 1 \times 10^{-5}$), with results using the genome-wide threshold ($P < 5 \times 10^{-8}$) reported as a sensitivity analysis. In the main text, we report the shrinkage estimates that reach nominal significance ($P < 0.05$). We presented shrinkage estimates in terms of % (multiplying by 100).

As a sensitivity analysis, we also presented study-level shrinkage estimates for height and educational attainment and tested for heterogeneity. These phenotypes were chosen because of previous evidence for shrinkage on these phenotypes and available data.

**Heterogeneity of shrinkage across variants within a phenotype.** We used results of the within-sibship and population-based meta-analyses to estimate whether shrinkage estimates were consistent across all variants within a phenotype, using an estimate of heterogeneity. As above, we only evaluated heterogeneity for height and educational attainment because of previous evidence and available data. For each variant we estimated the Wald ratio of the shrinkage estimate

$$s_k = \frac{\beta_{P,k}}{\beta_{W,k}}$$

The heterogeneity estimate was obtained as

$$Q = \sum_k^M w_k^2 (s_k - S)^2$$

where

$$w_k = \sqrt{\frac{S^2}{\sigma_{W,k}^2 + S^2 \sigma_S^2}}$$

**Applying LDSC to within-sibship data.** LDSC is a widely used method that can be applied to GWAS summary data to estimate heritability and genetic correlation[20,23]. The LDSC ratio, a function of the LDSC intercept unrelated to statistical power, is a measure of the proportion of association signal that is due to confounding. In this work, we apply LDSC to estimate SNP heritability and genetic correlation using the population and within-sibship GWAS data, so we investigated the LDSC intercept/ratio estimates from these data. Further detail is contained in the Supplementary Methods.

LDSC confounding estimates varied across the 25 phenotypes in the within-sibship model. Confounding estimates were modest for height (10%; 95% CI 6%, 14%) and BMI (9%; 2%, 16%), while the estimate for educational attainment was imprecise (35%; 12%, 57%). Across all phenotypes in the within-sibship data, the median confounding estimate was 21% (Q1–Q3: 10%, 28%), but stronger conclusions are limited by imprecise estimates (Supplementary Table 11 and Extended Data Fig. 8). The LDSC confounding estimates were higher using the population GWAS data (median 42%: Q1–Q3, 35%, 56%) than both the within-sibship model and previous studies (Supplementary Table 12). For example, the population model LDSC ratio estimates were higher for height (23%; 21%, 26%), BMI (22%; 19%, 25%) and educational attainment (41%; 37%, 45%).

The observed nonzero confounding in the within-sibship model was unexpected because of the intuition that the within-sibship GWAS models are unlikely to be confounded. The LDSC ratios in the population GWAS were also higher than previous studies. We followed up these findings by evaluating the effects of LD score mismatch and cryptic relatedness on the LDSC ratios.

**Evaluation of LD score mismatch.** A large proportion of samples in the meta-analysis were from UK-based studies such as UK Biobank and Generation Scotland, for which the LD scores, generated using 1000 Genomes project (phase 3) European samples (CEU, TSC, FIN, GBR), have been shown to fit reasonably well[20]. However, a large number of samples were from Scandinavian populations (HUNT study, FinnTwin), where LD mismatch leading to elevated LDSC intercept/ratios has been previously discussed[20]. We investigated this possibility using empirical and simulated data.

We investigated variation in LDSC ratios across populations by comparing ratios for height across well-powered individual studies ($N > 5,000$): UK Biobank, HUNT, the China Kadoorie Biobank (using default East Asian LD scores), Generation Scotland, DiscovEHR, Queensland Institute of Medical Research (QIMR) study and FinnTwin. We found some evidence of heterogeneity between studies: ratio estimates were higher in Scandinavian studies compared with UK-based studies (Extended Data Fig. 9). We also calculated within-sibship ratio estimates for BMI, SBP and educational attainment using UK Biobank summary data. UK Biobank estimates were largely consistent with zero confounding although confidence intervals were wide (Supplementary Table 13).

We also performed simulations to evaluate potential mismatch between the Norwegian HUNT study and the default LD scores, which were generated using 1000 Genomes data, finding evidence of LD score mismatch between the 1000 Genomes LD scores and HUNT. The simulation setup and results are detailed in the Supplementary Methods.

The combined findings from the empirical and simulated analyses suggest that LD score mismatch with the 1000 Genomes LD scores in the Norwegian HUNT study and other studies likely contributed to inflated LDSC ratios in both population and within-sibship GWAS models.

**Cryptic relatedness.** One source of inflation in GWAS associations is cryptic relatedness: nonindependence between close relatives in the study sample results which leads to inflated precision. In sibling GWAS models we clustered standard errors over sibships, but this clustering does not account for nonindependence between related sibships, for example, uncle/mother and two offspring. Inflated signal relating to cryptic relatedness may result in confounded signal, which is detected by the LD score intercept/ratio. In conventional population GWAS, either close relatives are removed or a mixed model is used to account for close relatives. We performed empirical and simulated analyses detailed in the Supplementary Methods to investigate the effect of cryptic relatedness on the population and within-sibship models.

The results suggest that the standard errors in the within-sibship model are not underestimated because of cryptic relatedness relating to common environmental effects shared between relatives. Thus, cryptic relatedness likely inflated LDSC ratios in the population models but not in the within-sibling data.

**Within-sibship SNP heritability estimates.** LDSC was used to generate SNP heritability estimates for 25 phenotypes using the LDSC harmonized (see above) meta-analysis summary data. The summary data were harmonized using the LDSC munge_sumstats.py function, and we used the precomputed European LD scores from 1000 Genomes Phase 3.

LDSC requires a sample size parameter $N$ to estimate SNP heritability. For this parameter, we used the effective sample size for each meta-analysis phenotype, equivalent to the number of independent observations. This was estimated as follows using GWAS standard errors, minor allele frequencies and the phenotype standard deviations (after adjusting for covariates).

$$\text{Effective } N = \frac{1}{\text{s.e.}^2} \frac{\text{s.d.\_Resid}^2}{2 \times \text{MAF} \times (1 - \text{MAF})}$$

s.e., GWAS model standard error; MAF, MAF of the variant; s.d._Resid, standard deviation of the regression residual.

Effective sample size was estimated for each individual study GWAS and each model (for example, UK Biobank population GWAS of height). To reduce noise from low-frequency variants, we restricted to variants with MAF between 0.1 and 0.4 (from 1000 Genomes EUR). At the meta-analysis stage, the effective sample size for each variant was calculated as the sum of sample sizes of studies in which the variant was present. Simulations evaluating the use of effective sample sizes are detailed in the Supplementary Methods.

In empirical analyses, we decided to focus on the differences between the population model ($h_{\text{Pop}}^2$) and within-sibship model ($h_{\text{WS}}^2$) SNP heritability estimates. If we assume that biases affect the estimates equally then the difference between the two estimates will be unbiased. We estimated the difference between the heritability estimates ($h_{\text{Diff}}^2$) using a difference-of-two-means test[57] as below.

$$h_{\text{Diff}}^2 = h_{\text{Pop}}^2 - h_{\text{WS}}^2$$

$$\text{s.e.}\left(h_{\text{Diff}}^2\right) \sim \sqrt{\text{s.e.}(h_{\text{Pop}}^2)^2 + \text{s.e.}(h_{\text{WS}}^2)^2 - 2\text{Cov}(h_{\text{Pop}}^2, h_{\text{WS}}^2)}$$

To estimate $\text{Cov}(h_{\text{Pop}}^2, h_{\text{WS}}^2)$, we computed the cross-GWAS LDSC intercept between the population and within-sibship GWAS data (for the same phenotype)

which is an estimate of $\mathrm{Cor}(h_{\mathrm{Pop}}^2, h_{\mathrm{WS}}^2)$. The estimates of this term were ~0.40 across phenotypes. We then calculated the covariance term as follows:

$$\mathrm{Cov}(h_{\mathrm{Pop}}^2, h_{\mathrm{WS}}^2) = \mathrm{Cor}(h_{\mathrm{Pop}}^2, h_{\mathrm{WS}}^2) \times \mathrm{s.e.}(h_{\mathrm{Pop}}^2) \times \mathrm{s.e.}(h_{\mathrm{WS}}^2)$$

We used the difference $Z$ score (that is, $\frac{h_{\mathrm{Diff}}^2}{\mathrm{s.e.}(h_{\mathrm{Diff}}^2)}$) to generate a $P$ value for the difference between $h_{\mathrm{Pop}}^2$ and $h_{\mathrm{WS}}^2$. In the text, we report differences reaching nominal significance (difference $P < 0.05$).

We calculated the expected effect of shrinkage on LDSC SNP heritability estimates. LDSC heritability estimates ($h^2$) are derived from the formulation below[20]:

$$\chi^2 \sim \frac{Nh^2 l_j}{M} + Na + 1$$

where $\chi^2$ is the square of the GWAS $Z$ score, $N$ is the sample size, $M$ is number of variants such that $\frac{h^2}{M}$ is the average heritability for each variant, $l_j$ is the LD score of variant $j$ and $a$ is the effect of confounding biases.

Uniform shrinkage across the genome would lead to GWAS $Z$ scores being multiplied by a factor $(1-k)$, where $k$ is the shrinkage coefficient, and $\chi^2$ statistics being multiplied by $(1-k)^2$. As above, we have used effective sample size to account for differences in $N$ between the population and within-sibship models. Therefore, assuming all other coefficients remain consistent, the expectation of $h_{\mathrm{WS}}^2$ can be written as a function of $k$ and $h_{\mathrm{Pop}}^2$.

$$h_{\mathrm{Pop}}^2 = y$$

$$h_{\mathrm{WS}}^2 = (1-k)^2 y$$

To evaluate the sensitivity of our results to assumptions of heritability models, we also estimated SNP heritability using SumHer[21], which allows the use of different heritability models with regard to how local LD and allele frequencies affect the heritability contributions of individual SNPs. In SumHer analyses, we followed the same procedure as above for LDSC using effective sample sizes and estimating SNP heritability for all 25 phenotypes. We used the LDAK-Thin model with the precomputed tagging file over the BLD-LDAK model because of the limited power of our datasets (the BLD-LDAK model includes additional parameters so generates less precise estimates).

**Within-sibship $r_g$ with educational attainment.** We used LDSC to estimate $r_g$ between educational attainment and other phenotypes using both population and within-sibship data. LDSC requires nonzero heritability to generate meaningful $r_g$ estimates, so we restricted analyses to the 22 phenotypes with SNP heritability point estimates greater than zero in both population and within-sibship models (that is, omitted physical activity and ratio of forced expiratory volume (FEV1)/forced vital capacity (FEV1FVC)). We estimated only pairwise genetic correlations between educational attainment and all other phenotypes because of previous evidence that educational attainment is influenced by demographic and indirect genetic effects and, given the limited statistical power, to reduce the multiple testing burden. Estimates failed to converge for genetic correlation analyses involving age at first birth and age at menopause, so these phenotypes were not analyzed here. We estimated the difference between the population ($r_{g,\mathrm{Pop}}$) and within-sibship ($r_{g,\mathrm{WS}}$) estimates ($r_{g,\mathrm{Diff}}$) using a difference-of-two-means test[57].

$$r_{g,\mathrm{Diff}} = r_{g,\mathrm{Pop}} - r_{g,\mathrm{WS}}$$

We used the jackknife to estimate the standard error of the difference, s.e.$(r_{g,\mathrm{Diff}})$. After restricting to ~1.2 million Hapmap 3 variants present in the 1000 Genomes LD scores, we ordered variants by chromosome and base pair and separated variants into 100 blocks. We removed each block in turn and computed $r_{g,\mathrm{Diff}}$ using LDSC 100 times. We then calculated s.e.$(r_{g,\mathrm{Diff}})$ across the 100 iterations as follows:

$$\mathrm{s.e.}(r_{g,\mathrm{Diff}}) = \sqrt{\frac{99}{100} \sum_1^{100} (r_{g,\mathrm{Diff}\,k} - \mu)^2}$$

where

$$\mu = \frac{\sum_1^{100} r_{g,\mathrm{Diff},k}}{100}$$

$r_{g,\mathrm{Diff},k}$ is the $r_g$ estimate in the $k$th iteration and $\mu$ is the mean $r_g$ estimate across all 100 iterations.

We used the difference $Z$ score (that is, $\frac{r_{g,\mathrm{Diff}}}{\mathrm{s.e.}(r_{g,\mathrm{Diff}})}$) to generate a $P$ value for heterogeneity between $r_{g,\mathrm{Pop}}$ and $r_{g,\mathrm{WS}}$. In the text, we report differences reaching nominal significance (heterogeneity $P < 0.05$).

**WS-MR: effects of height and BMI.** We performed MR analyses using the within-sibship meta-analysis GWAS data to estimate the effect of two exposures (height and BMI) on 23 outcome phenotypes. For the exposure instruments, we

used 803 and 418 independent genetic variants for height and BMI, respectively. These variants were identified by LD clumping in PLINK ($r^2 < 0.001$, physical distance threshold $= 10,000$ kb, $P < 5 \times 10^{-8}$) using the 1000 Genomes Phase 3 EUR reference panel[54]. We then performed an MR-IVW analysis using the within-sibship meta-analysis data to estimate the effect of the exposure on the outcome as

$$\beta_{\mathrm{MR}} = \sum \frac{\beta_{\mathrm{Exp}} * \beta_{\mathrm{Out}}}{(\sigma_{\mathrm{Out}})^2} \Big/ \sum \frac{(\beta_{\mathrm{Exp}})^2}{(\sigma_{\mathrm{Out}})^2}$$

where $\beta_{\mathrm{Exp}}$ is the association estimate from exposure GWAS, $\beta_{\mathrm{Out}}$ is the association estimate from outcome GWAS and $\sigma_{\mathrm{Out}}$ is the standard error from outcome GWAS.

We also performed MR analyses using the population meta-analysis GWAS data for comparison. We estimated differences between population MR and WS-MR estimates using the difference-of-two-means test[57]:

$$\beta_{\mathrm{MR,Diff}} = \beta_{\mathrm{MR,Pop}} - \beta_{\mathrm{MR,WS}}$$

We used the jackknife to estimate the standard error of the difference, s.e.$(\beta_{\mathrm{MR,Diff}})$. With $n$ genetic instruments, we removed each variant from the analysis in turn and then computed $\beta_{\mathrm{MR,Diff}}$, storing the estimate from the $n$ iterations. We then calculated s.e.$(\beta_{\mathrm{MR,Diff}})$ as follows:

$$\mathrm{s.e.}(\beta_{\mathrm{MR,Diff}}) = \sqrt{\frac{n-1}{n} \sum_1^n (\beta_{\mathrm{MR,Diff},k} - \mu)^2}$$

where

$$\mu = \frac{\sum_1^n \beta_{\mathrm{MR,Diff},k}}{n}$$

$n$ is the number of genetic variants used as instruments, $\beta_{\mathrm{MR,Diff},k}$ is the $\beta_{\mathrm{MR,Diff}}$ estimate in the $k$th iteration and $\mu$ is the mean $\beta_{\mathrm{MR,Diff}}$ estimate across all $n$ iterations.

We used the difference $Z$ score (that is, $\frac{\beta_{\mathrm{MR,Diff}}}{\mathrm{s.e.}(\beta_{\mathrm{MR,Diff}})}$) to generate a $P$ value for heterogeneity between $\beta_{\mathrm{MR,Pop}}$ and $\beta_{\mathrm{MR,WS}}$. In the text, we report differences reaching nominal significance (heterogeneity $P < 0.05$).

**Polygenic adaptation.** Polygenic adaptation was estimated using similar methods to a previous publication[28]. Precomputed SDS were downloaded for UK10K data from https://web.stanford.edu/group/pritchardlab/. Genomic regions under strong recent selection (*MHC* chr6: 25,892,529–33,436,144; lactase chr2: 134,608,646–138,608,646) were removed and SDS were normalized within each 1% allele frequency bin.

SDS were merged with GWAS meta-analysis data for 25 phenotypes. Variants with low effective sample sizes (<50% of maximum) were removed for each phenotype. SDS were transformed to tSDS such that the reference allele was the phenotype-increasing allele.

Spearman's rank test was used to estimate the correlation between tSDS and the absolute value of GWAS $Z$ scores from the population and within-sibship models. Standard errors were estimated using the jackknife. The genome was ordered by chromosome and base pair and divided into 100 blocks. Correlations were estimated 100 times with each $k$th block removed in turn. The standard error of the correlation estimate, s.e.(Cor), was calculated as follows:

$$\mathrm{s.e.}(\mathrm{Cor}) = \sqrt{\frac{99}{100} \sum_1^{100} (\mathrm{Cor}_k - \mu)^2}$$

where

$$\mu = \frac{\sum_1^{100} \mathrm{Cor}_k}{100}$$

$\mathrm{Cor}_k$ is Spearman's rank correlation estimate in the $k$th iteration and $\mu$ is the mean correlation estimate across the 100 iterations.

Given previous concerns[26,27], we performed several sensitivity analyses for the height analysis detailed in the Supplementary Methods.

**Reporting Summary.** Further information on research design is available in the Nature Research Reporting Summary linked to this article.

## Data availability

European meta-analysis summary statistics for both the within-sibship and population GWAS models are publicly available on OpenGWAS (https://gwas.mrcieu.ac.uk/). The relevant GWAS IDs in OpenGWAS are ieu-b-4813 to ieu-b-4860 (for example, within-sibship GWAS estimates for height are in https://gwas.mrcieu.ac.uk/datasets/ieu-b-4813/). A description of the available summary data will be on the consortium website (https://www.withinfamilyconsortium.com/home/). UK Biobank individual-level participant data are available via enquiry

to access@ukbiobank.ac.uk. Researchers associated with Norwegian research institutes can apply for the use of HUNT data and samples with approval by the Regional Committee for Medical and Health Research Ethics. Researchers from other countries may apply if collaborating with a Norwegian Principal Investigator. Information for data access can be found at https://www.ntnu.edu/hunt/data. Generation Scotland data access can be applied for via enquiry to access@generationscotland.org. Please see https://www.ed.ac.uk/generation-scotland/for-researchers/access. Researchers interested in China Kadoorie Biobank data access should contact ckbaccess@ndph.ox.ac.uk. Please see https://www.ckbiobank.org/site/Data+Access. Researchers interested in TEDS data can complete a data request form at https://www.teds.ac.uk/researchers/teds-data-access-policy. Researchers interested in TwinsUK data can fill in a proposal form at https://twinsuk.ac.uk/resources-for-researchers/access-our-data/. Researchers interested in data from ORCADES and Viking1 can contact accessQTL@ed.ac.uk. GENOA data are available via application to dbGaP https://ega-archive.org/studies/phs000379. Researchers interested in Swedish Twin Registry data can find instructions at https://ki.se/en/research/swedish-twin-registry-for-researchers. Researchers interested in Danish Twin Registry data can contact tvilling@health.sdu.dk.

## Code availability

Code for running GWAS analyses is available on GitHub (https://github.com/LaurenceHowe/SiblingGWAS)[58]. Code for downstream analyses is available (https://github.com/LaurenceHowe/SiblingGWASPost)[59].

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

## Acknowledgements

L.J.H., T.T.M., Y.C., D.A.L., G.D.S., G.H. and N.M.D. work in a unit that receives support from the University of Bristol and the UK MRC (grant nos. MC_UU_00011/1 & 6). N.M.D. is supported by a Norwegian Research Council Grant (no. 295989). G.H. is supported by the Wellcome Trust and Royal Society (grant no. 208806/Z/17/Z). B.M.B., B.O.Å., H.R., A.F.H. and K.H. work in a research unit funded by Stiftelsen Kristian Gerhard Jebsen, the Liaison Committee for education, research and innovation in Central Norway and the Joint Research Committee between St. Olavs Hospital and the Faculty of Medicine and Health Sciences, NTNU. Funding information for other co-authors is contained in the Supplementary information. We thank H. Mostafavi and J. Pritchard for helpful suggestions and guidance relating to the polygenic adaptation analyses.

## Author contributions

L.J.H., M.G.N., T.T.M., Y.C., J.B.P, J.F.W, J.L.H., S.L., M.C.S., D.A.L., N.G.M., A.H., K.H., C.J.W., B.O.Å., P.D.K., J.K., S.E.M., D.J.B., P.T., D.M.E., G.D.S., C.H., B.M.B., G.H. and N.M.D. were closely involved in conceptualizing and designing the study. J.K., K.P.H., E.M.T.D., S.M.K., H.C., J.F.W., E.J.C.D., R.P., J.A.S., P.A.P., S.L.R.K., S.L., J.L.H., M.C.S., K.C., N.M.D., S.E.M., N.G.M., B.M.B., R.G.W., I.Y.M., K.L., K.H., C.J.W., C.R.B., A.E.J., D.P., C.H. and A.C. were involved in data and funding acquisition. L.J.H. developed the GitHub GWAS pipeline with support from G.H. and N.M.D. and programming code from G.H. and P.T. (via SSGAC). C.H. kindly beta tested the GWAS pipeline and suggested improvements. Other analysts (listed below) also made major contributions to the GWAS pipeline. L.J.H., S.G., A.F.H., H.R., C.H., Y.C., G.C., R.A., P.A.L., T.P., M.D.V.Z., R.C., M.M., Y.W., S.L., L.K., S.M.R., L.F.B., C.A.R., M.N., J.V.B., A.G. and E.A.W. performed GWAS analyses in individual cohorts with the support and guidance of N.M.D., G.H., B.M.B., R.G.W., I.Y.M., K.L., S.O., A.E.J., S.E.M., J.K., M.G.N., M.B., J.B.P., S.H., J.L.H., J.F.W., J.A.S., P.A.P., S.L.R.K., K.C., M.C.K. and J.J.L. L.J.H. performed meta-analyses and all downstream analyses with the meta-analysis data. L.J.H. drafted the first version of the manuscript. M.G.N., T.T.M., B.O.Å., P.D.K., J.K., S.E.M., R.G.W., D.J.B., P.T., D.M.E., G.D.S., C.H., B.M.B., G.H. and N.M.D. played a key role in interpreting the results, planning additional analyses and revising the manuscript. All authors contributed to and critically reviewed the manuscript.

## Competing interests

O.A.A. is a consultant to HealthLytix in a capacity unrelated to this work. All other authors declare no competing interests.

## Additional information

**Extended data** is available for this paper at https://doi.org/10.1038/s41588-022-01062-7.

**Correspondence and requests for materials** should be addressed to Laurence J. Howe, Ben Brumpton, Gibran Hemani or Neil M. Davies.

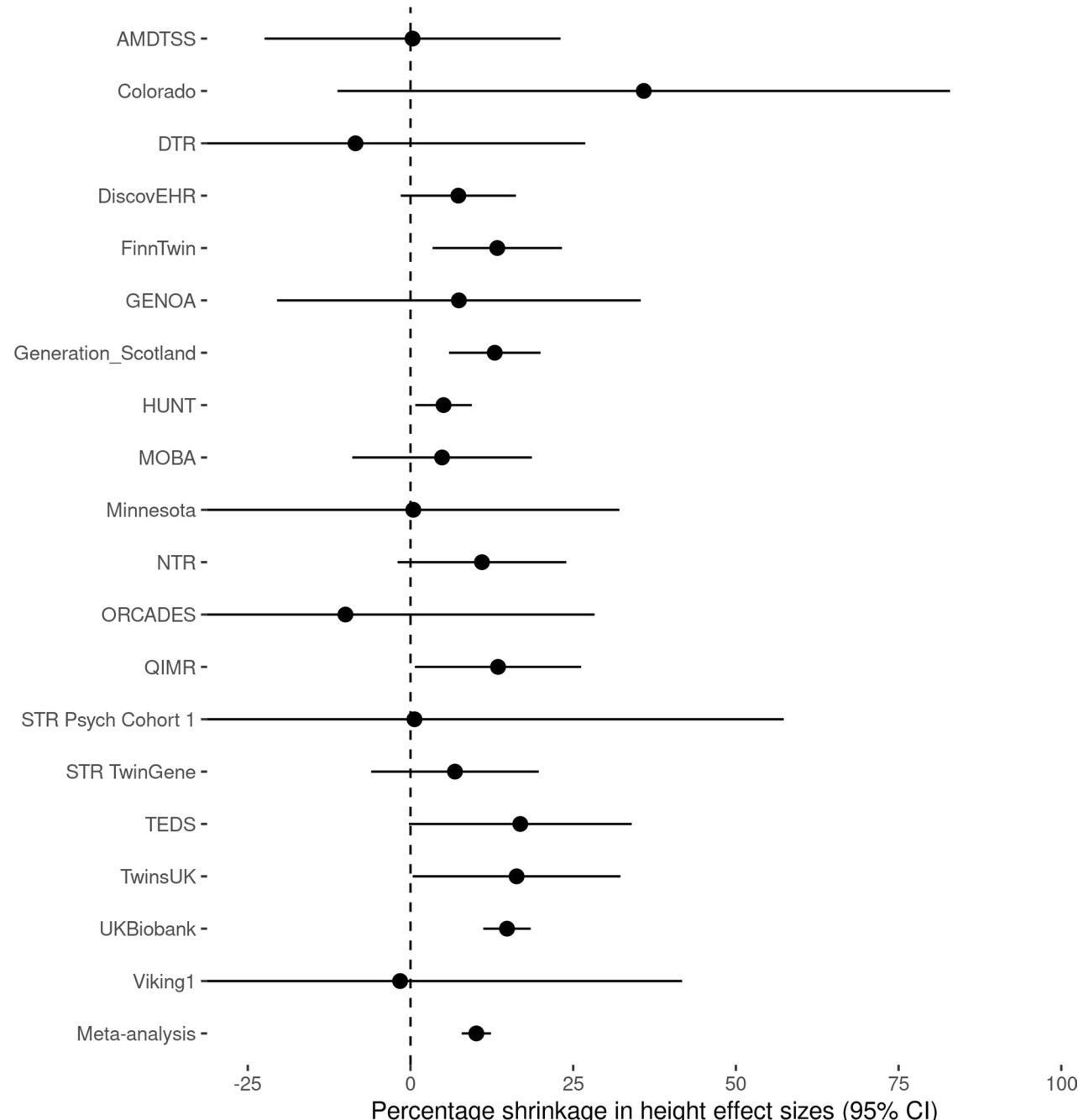

**Extended Data Fig. 1 | Within-sibship shrinkage for height across European ancestry cohorts.** AMDTSS = Australian Mammographic Density Twin Study, DTR = Danish Twins Registry, NTR = Netherlands Twin Registry, QIMR = QIMR Berghofer Medical Research Institute (QIMR), TEDS = Twins Early Development Study. Extended Data Figure 1 shows estimates of within-sibship shrinkage and 95% confidence intervals for height variants in all of the cohorts contributing to the European meta-analysis as well as the meta-analysis GWAS. Shrinkage is defined as the % decrease in association between the relevant weighted score and phenotype when comparing the population estimate to the within-sibship estimate. Shrinkage was computed as the ratio of two weighted score association estimates with standard errors derived using leave-one-out jackknifing. These estimates used the weighted score for each phenotype at the more liberal threshold (P < 1×10⁻⁵). The total number of individuals in the meta-analysis was n = 149,174 with individual study sample sizes ranging from n = 601 for the Colorado based CADD study to n = 40,068 for UK Biobank. Further information on samples with height data in each cohort are contained in Supplementary Table 1.

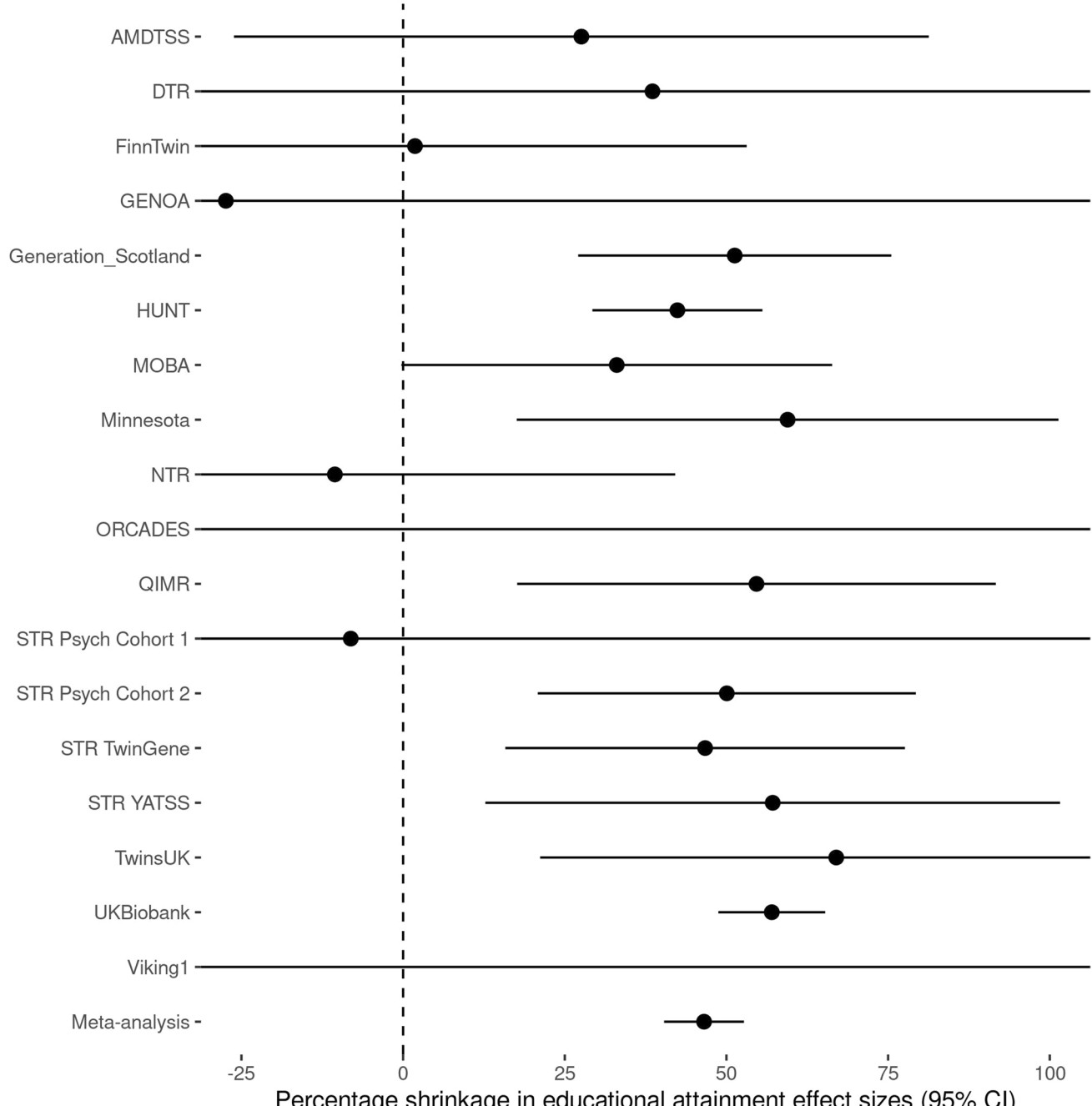

**Extended Data Fig. 2 | Within-sibship shrinkage for educational attainment across European ancestry cohorts.** AMDTSS = Australian Mammographic Density Twin Study, DTR = Danish Twins Registry, NTR = Netherlands Twin Registry, QIMR = QIMR Berghofer Medical Research Institute (QIMR). Extended Data Figure 2 shows estimates of within-sibship shrinkage and 95% confidence intervals for educational attainment variants in all of the cohorts contributing to the European meta-analysis as well as the meta-analysis GWAS. Shrinkage is defined as the % decrease in association between the relevant weighted score and phenotype when comparing the population estimate to the within-sibship estimate. Shrinkage was computed as the ratio of two weighted score association estimates with standard errors derived using leave-one-out jackknifing. These estimates used the weighted score for each phenotype at the more liberal threshold (P < 1×10$^{-5}$). The total number of individuals in the meta-analysis was n = 128,777 with individual study sample sizes ranging from n = 742 for STR Psych Cohort 1 to n = 39,531 for UK Biobank. Further information on samples with educational attainment data in each cohort are contained in Supplementary Table 1.

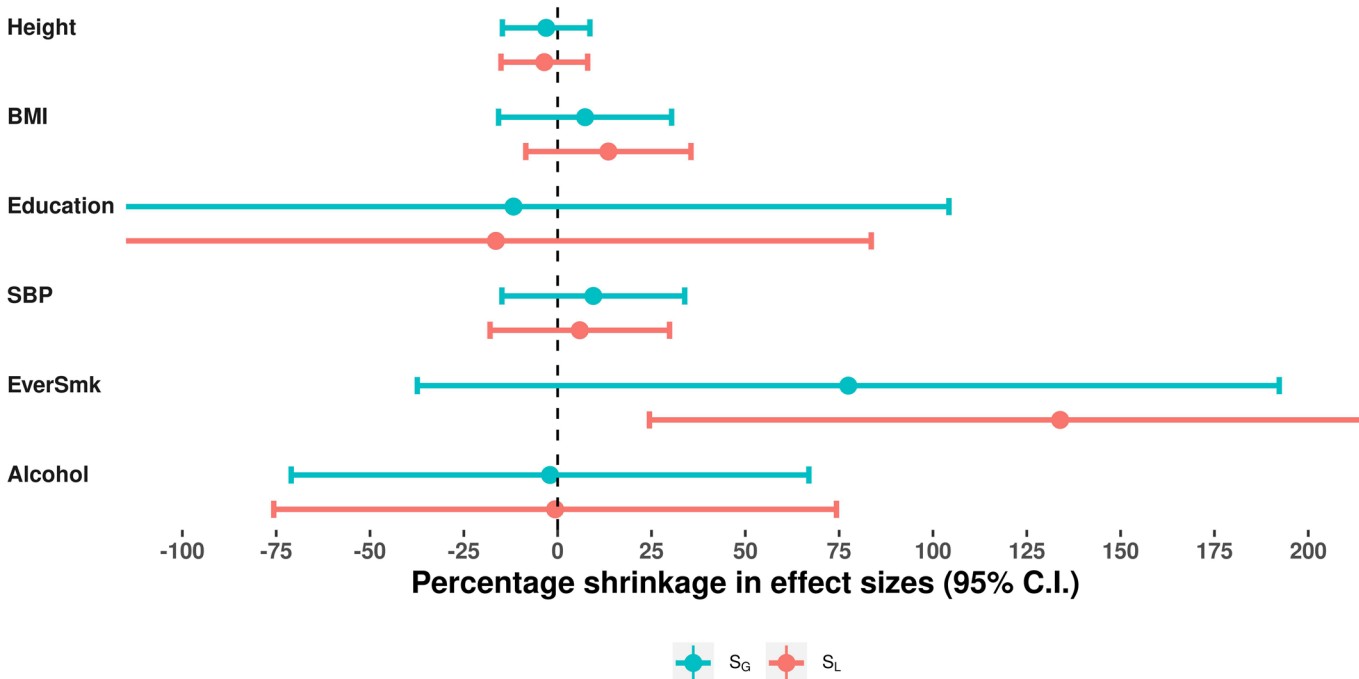

**Extended Data Fig. 3 | Within-sibship shrinkage estimates from China Kadoorie Biobank.** $S_G$ = score including variants with P < 5×10$^{-8}$, $S_L$ = score including variants with P < 1×10$^{-5}$, BMI = body mass index, SBP = systolic blood pressure, EverSmk = ever smoking. Extended Data Figure 3 contains within-sibship shrinkage estimates and 95% confidence intervals for height, BMI, educational attainment, systolic blood pressure and ever-smoking genetic variants in China Kadoorie Biobank. Shrinkage is defined as the % decrease in association between the relevant weighted score and phenotype when comparing the population estimate to the within-sibship estimate. Shrinkage was computed as the ratio of two weighted score association estimates with standard errors derived using leave-one-out jackknifing. The figure includes genetic variants from the genome-wide significant (blue) and liberal (red) thresholds. Note that the genetic variants tested were identified in UK Biobank, but any ancestral differences will likely equally affect both the population and within-sibship estimates, meaning that the shrinkage estimate are unlikely to be biased by ancestral differences. Data was available from n = 13,856 individuals for each of the 6 phenotypes.

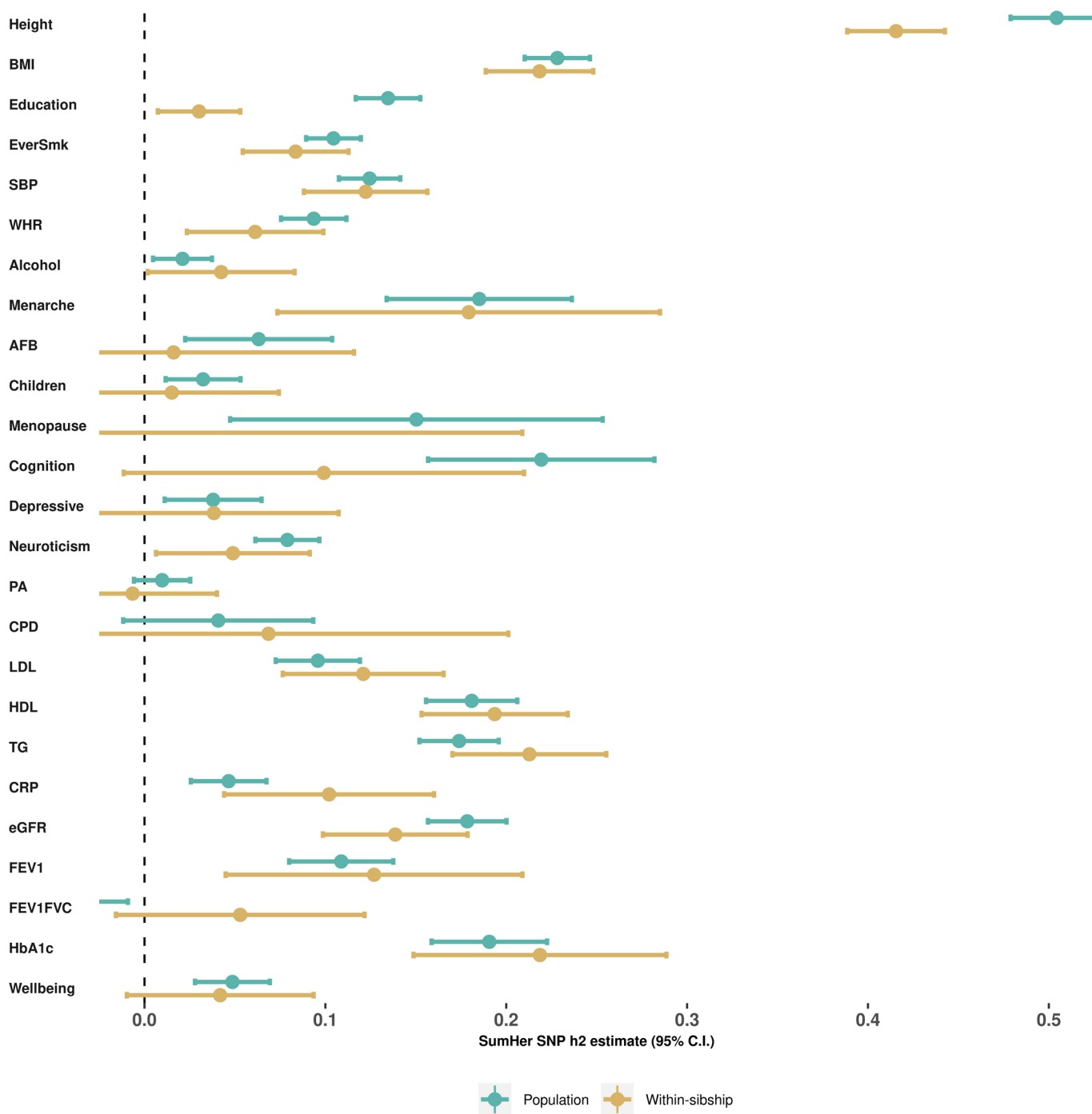

**Extended Data Fig. 4 | SumHer SNP heritability estimates.** BMI = body mass index, Education = educational attainment, EverSmk = ever smoking, SBP = systolic blood pressure, WHR = waist-hip ratio, Alcohol = weekly alcohol consumption, Menarche = age at menarche, AFB = age at first birth, Children = number of biological children, Menopause = age at menopause, Cognition = cognitive ability, Depressive = depressive symptoms, PA = physical activity, CPD = cigarettes per day, LDL = LDL cholesterol, HDL = HDL cholesterol, TG = triglycerides, CRP = C-reactive protein, eGFR = estimated glomerular filtration rate, FEV1 = forced expiratory volume, FEV1FVC = ratio of FEV1/forced vital capacity, HbA1c = Haemoglobin A1C. Extended Data Figure 4 displays SumHer SNP $h^2$ (LDAK-thin model) estimates and corresponding 95% confidence intervals for 25 phenotypes using population and within-sibship meta-analysis data. The number of individuals contributing to each phenotype ranged from n = 149,174 for height to n = 13,375 for age at menopause. Further information on the sample sizes of each phenotype are contained in Supplementary Table 2.

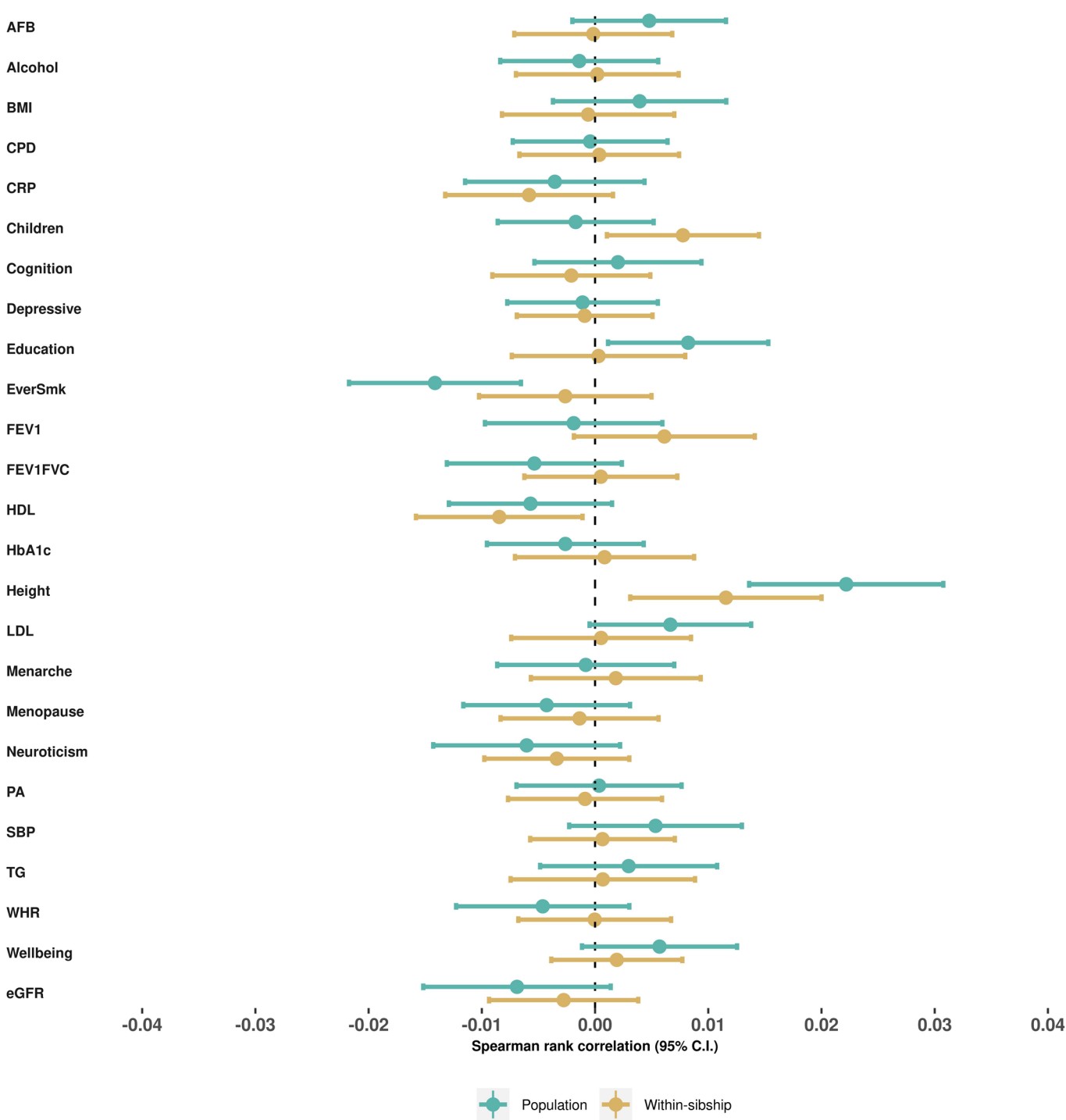

**Extended Data Fig. 5 | Evidence of polygenic adaption using SDS for 25 phenotypes.** BMI = body mass index, EverSmk = ever smoking, SBP = systolic blood pressure, WHR = waist-hip ratio, AFB = age at first birth, PA = physical activity, CPD = cigarettes per day, TG = triglycerides, CRP = C-reactive protein, eGFR = estimated glomerular filtration rate, FEV1 = forced expiratory volume, FEV1FVC = ratio of FEV1/forced vital capacity, HbA1c = Haemoglobin A1C. Extended Data Figure 5 displays spearman rank correlation estimates and corresponding 95% confidence intervals between tSDS (SDS aligned with phenotype increasing alleles) and absolute phenotype Z scores for 25 phenotypes. The phenotype Z scores were taken from both the meta-analysis of population (blue) and within-sibship (red) estimates. Positive correlations indicate evidence of historical positive selection on phenotype increasing alleles. The number of individuals contributing to each phenotype ranged from n = 149,174 for height to n = 13,375 for age at menopause. Further information on the sample sizes of each phenotype are contained in Supplementary Table 2.

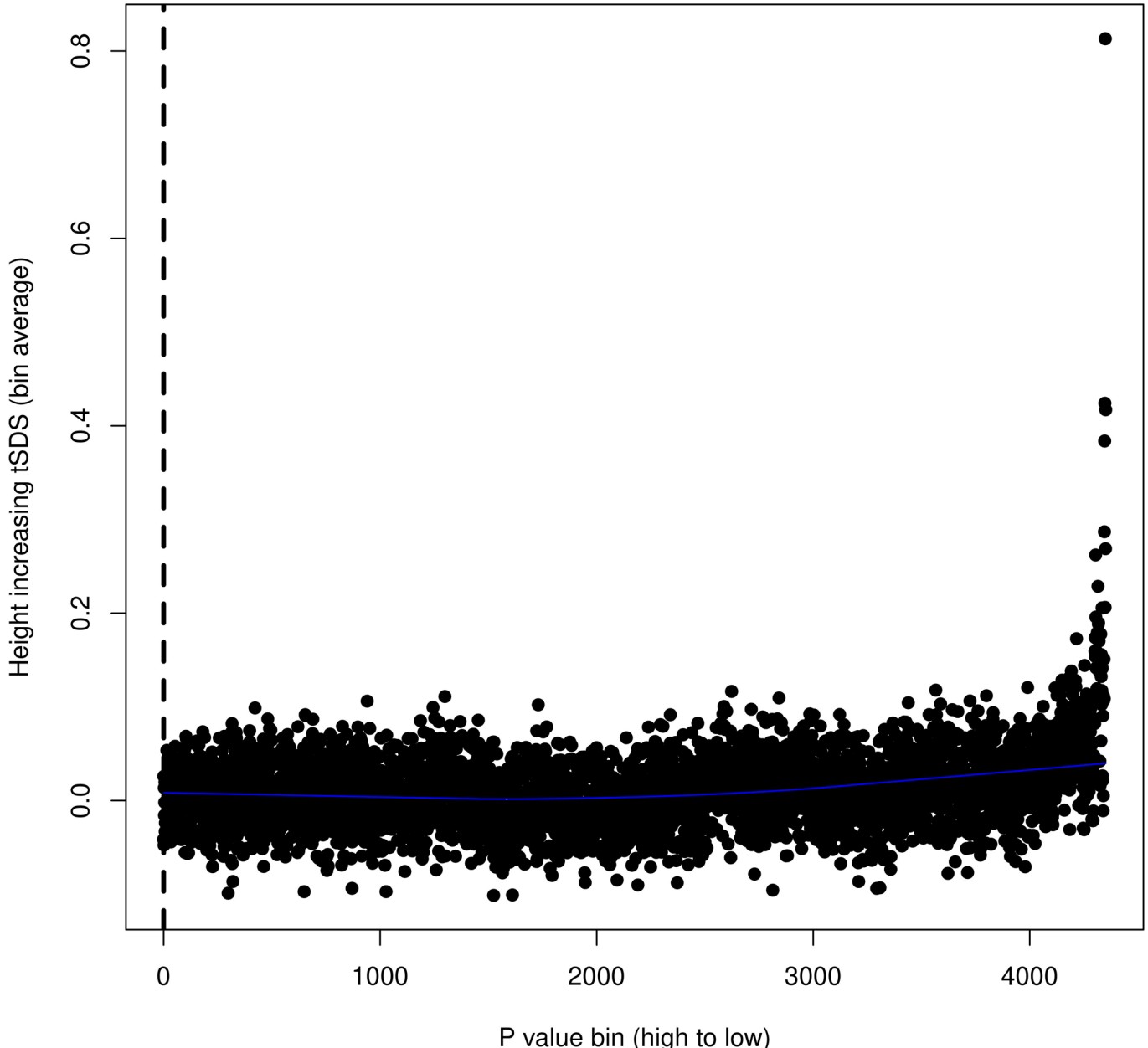

**Extended Data Fig. 6 | Scatter plot of SDS p-value bins against mean tSDS (of the bin) using the within-sibship height meta-analysis GWAS data.** In Extended Data Figure 6 each data point is the mean tSDS (SDS alligned with height increasing allele) in a set of 1000 genetic variants. Genetic variants were ordered by height P-value (from within-sibship meta-analysis GWAS data) and divided into bins. The plot illustrates evidence of a correlation between decreasing height P-value and higher mean tSDS suggestive of polygenic adaptation on height increasing alleles. The within-sibship European GWAS meta-analysis data (n = 149,174 individuals) were used for this analysis.

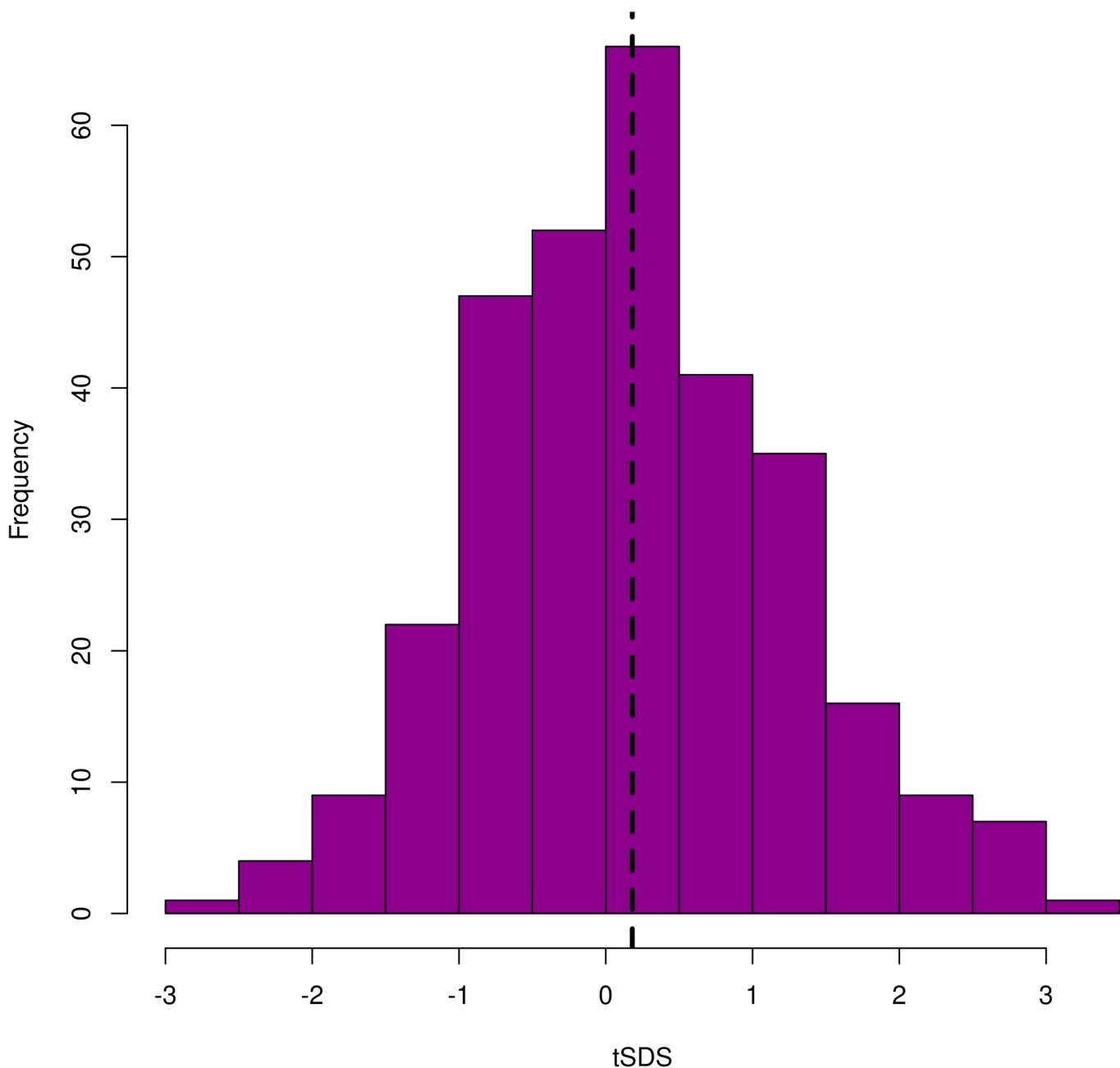

**Extended Data Fig. 7 | Histogram of tSDS for independent variants associated with height in the within-sibship meta-analysis data (P < 1×10⁻⁵).**
Extended Data Figure 7 Extended Data Figure 7 is a histogram of the distribution of tSDS (SDS aligned with height increasing alleles) amongst 310 putative independent height loci identified from the within-sibship meta-analysis data (P < 1×10⁻⁵). The plot indicates that the mean tSDS of these loci is higher than 0, consistent with polygenic adaptation on height increasing alleles. The within-sibship European GWAS meta-analysis data (n = 149,174 individuals) were used for this analysis.

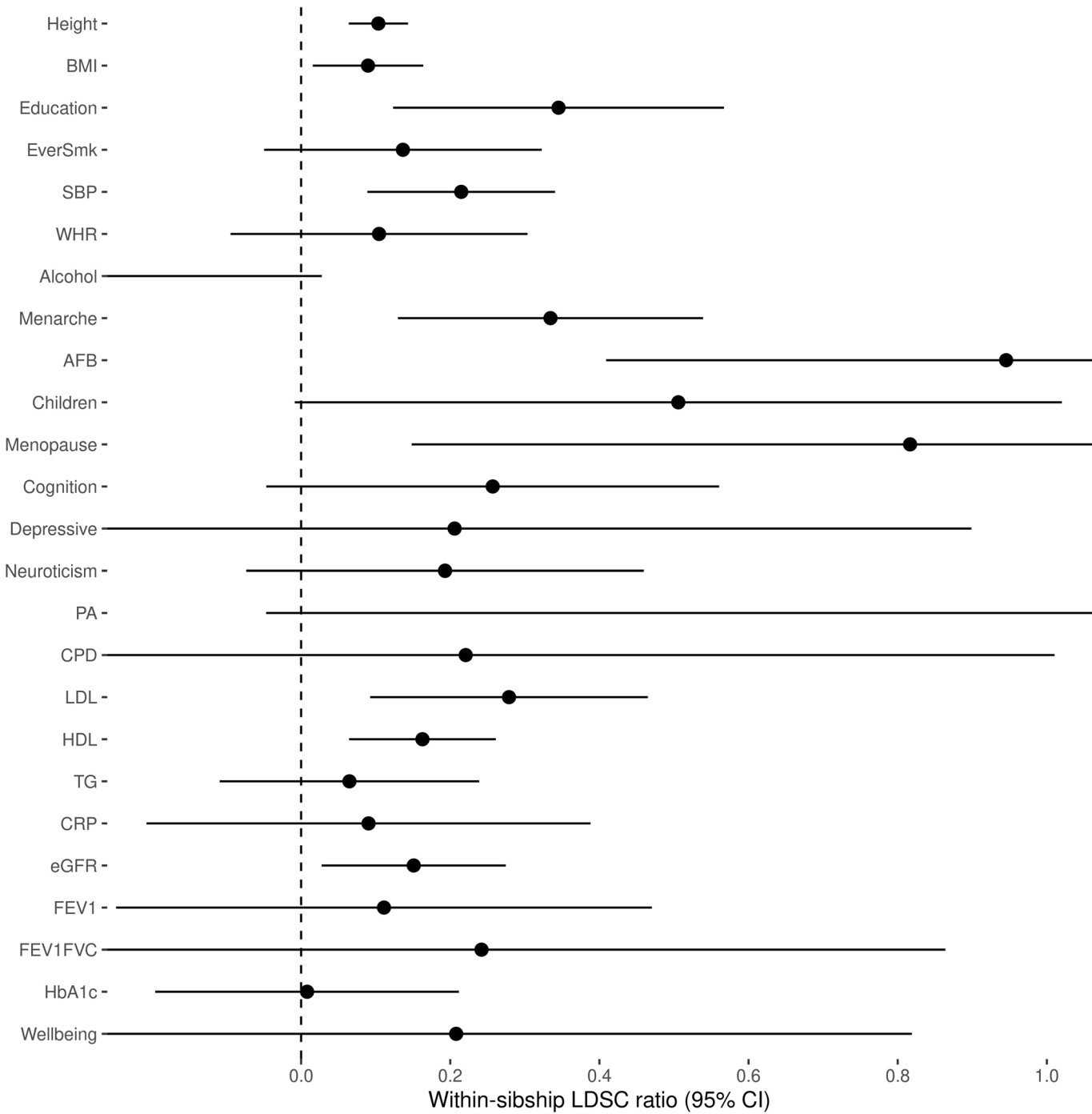

**Extended Data Fig. 8 | LDSC estimates of confounding across 25 phenotypes using within-sibship data.** BMI = body mass index, EverSmk = ever smoking, SBP = systolic blood pressure, WHR = waist-hip ratio, AFB = age at first birth, PA = physical activity, CPD = cigarettes per day, TG = triglycerides, CRP = C-reactive protein, eGFR = estimated glomerular filtration rate, FEV1 = forced expiratory volume, FEV1FVC = ratio of FEV1/forced vital capacity, HbA1c = Haemoglobin A1C. Extended Data Figure 8 shows LDSC ratio estimates and corresponding 95% confidence intervals 25 phenotypes using the within-sibship meta-analysis data. The LDSC ratio is a measure of the % of the polygenic signal attributable to confounding in a GWAS dataset. The number of individuals contributing to each phenotype ranged from n = 149,174 for height to n = 13,375 for age at menopause. Further information on the sample sizes of each phenotype are contained in Supplementary Table 2.

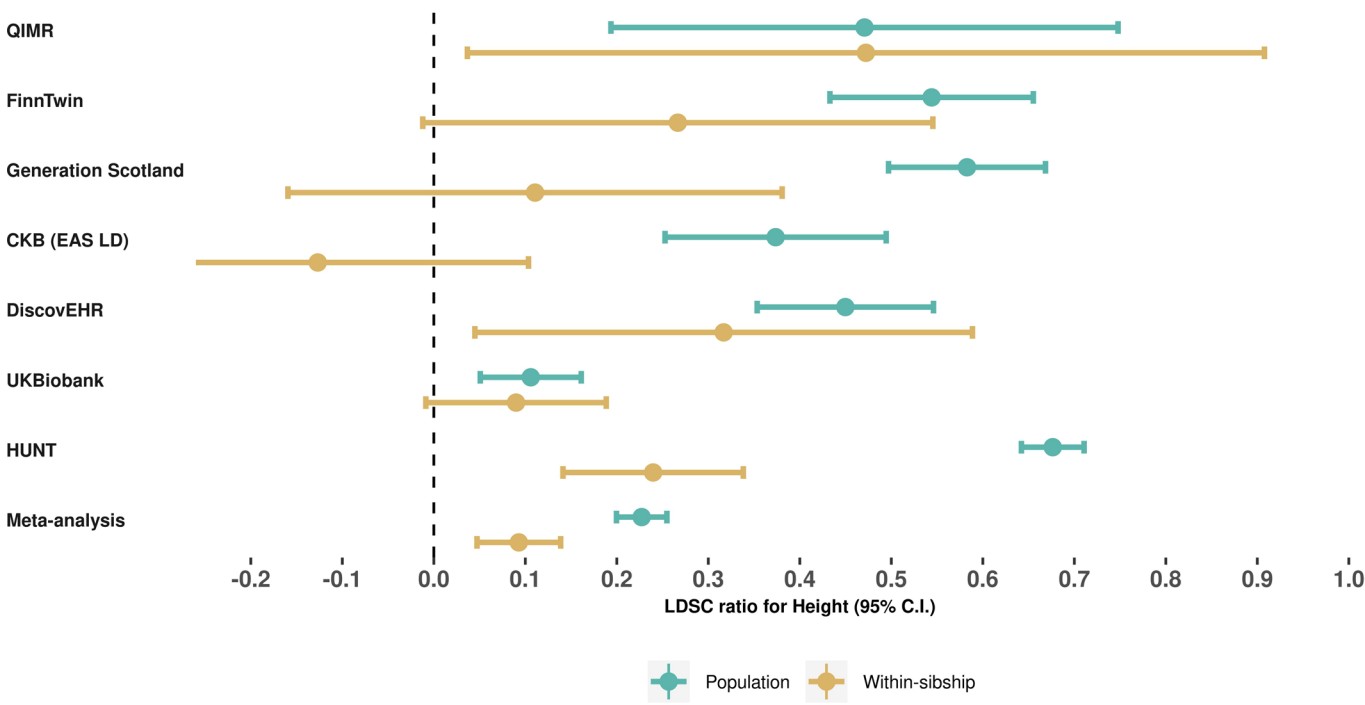

**Extended Data Fig. 9 | LDSC ratios from height GWAS.** Extended Data Figure 9 shows LDSC ratio estimates and corresponding 95% confidence intervals for height GWAS from the summary data of 7 individual studies and the meta-analysis of European studies. The LDSC ratio is a measure of the % of the polygenic signal attributable to confounding in a GWAS dataset. The number of individuals in the meta-analysis estimate was n = 149,174 with the sample sizes for the displayed individual studies ranging from n = 40,068 in UK Biobank to 8,810 in the Finnish Twin Cohort. Further information on available height data in each phenotype are contained in Supplementary Table 1.

# Reporting Summary

Nature Research wishes to improve the reproducibility of the work that we publish. This form provides structure for consistency and transparency in reporting. For further information on Nature Research policies, see our Editorial Policies and the Editorial Policy Checklist.

## Statistics

For all statistical analyses, confirm that the following items are present in the figure legend, table legend, main text, or Methods section.

| n/a | Confirmed | |
|---|---|---|
| ☐ | ☒ | The exact sample size (*n*) for each experimental group/condition, given as a discrete number and unit of measurement |
| ☐ | ☒ | A statement on whether measurements were taken from distinct samples or whether the same sample was measured repeatedly |
| ☐ | ☒ | The statistical test(s) used AND whether they are one- or two-sided *Only common tests should be described solely by name; describe more complex techniques in the Methods section.* |
| ☐ | ☒ | A description of all covariates tested |
| ☐ | ☒ | A description of any assumptions or corrections, such as tests of normality and adjustment for multiple comparisons |
| ☐ | ☒ | A full description of the statistical parameters including central tendency (e.g. means) or other basic estimates (e.g. regression coefficient) AND variation (e.g. standard deviation) or associated estimates of uncertainty (e.g. confidence intervals) |
| ☐ | ☒ | For null hypothesis testing, the test statistic (e.g. *F*, *t*, *r*) with confidence intervals, effect sizes, degrees of freedom and *P* value noted *Give P values as exact values whenever suitable.* |
| ☒ | ☐ | For Bayesian analysis, information on the choice of priors and Markov chain Monte Carlo settings |
| ☐ | ☒ | For hierarchical and complex designs, identification of the appropriate level for tests and full reporting of outcomes |
| ☐ | ☒ | Estimates of effect sizes (e.g. Cohen's *d*, Pearson's *r*), indicating how they were calculated |

*Our web collection on statistics for biologists contains articles on many of the points above.*

## Software and code

Policy information about availability of computer code

| Data collection | No software was used for data collection. |
|---|---|
| Data analysis | The open-source software, tools, and packages used for data analysis in this study, as well as the version of each program, were ImageJ (v2.1.0), R (v3.5.3 and v3.6.1), FASTQC (v0.11.9), HISAT2 (v2.1.0), featureCounts (v2.0.1), Seurat R package (v3.0.1), Harmony R package (v0.1), caret R package (v6.0-90), Rtsne R package (v0.15), PAMES R package (v2.6.2), CONICSmat R package (v1.0), DeepTools (v3.1.2), survival R package (v3.2-13), survAUC R package (v1.0-5), rms R package (v6.2-0), rpart R package (v4.1.16), DynNom R package (v5.0.1), DESeq2 (Bioconductor v3.10), SeSAMe (Bioconductor v3.10), minfi (Bioconductor v3.10), karyoplotR (Bioconductor v3.10), ConsensusClusterPlus (Bioconductor v3.10), and DiffBind (Bioconductor v3.10). No software was used for data collection. A methylation profile multi-class support vector machine (SVM) classifier was generated using the caret R package, and was deposited in the github repository abrarc/meningioma-svm (DOI:10.5281/enodo.6353877). In brief, a linear kernel SVM was constructed using training data comprising 75% of randomly selected samples from the discovery cohort (n=150) with 10-fold cross validation. 2,000 probes from each pre-processing pipeline were used as variables. The remaining 25% of samples from the discover cohort (n=50) were used to test the model, which performed with 97.9% accuracy when classifying samples into 3 SeSAMe groups (95% CI 89.2-99.9%, p<2.2x10-16). SVM classifiers for 3, 4, 5, or 6 minfi groups were generated using the same approach and performed with 91.8% (95% CI 80.4%-97.7%, p=4.69x10-9), 91.8% (95% CI 80.4%-97.7%, p=9.58x10-16), 93.8% (95% CI 82.8%-98.7%, p=2.98x10-16), and 93.6% (95% CI 82.5%-98.7%, p<2.2x10-16) accuracy, respectively. SVM classification and K-means consensus clustering of the validation cohort was performed with the same parameters as for the discovery cohort using the same probes in the validation cohort that were identified from the discovery cohort. |

For manuscripts utilizing custom algorithms or software that are central to the research but not yet described in published literature, software must be made available to editors and reviewers. We strongly encourage code deposition in a community repository (e.g. GitHub). See the Nature Research guidelines for submitting code & software for further information.

## Data

Policy information about availability of data

All manuscripts must include a data availability statement. This statement should provide the following information, where applicable:

- Accession codes, unique identifiers, or web links for publicly available datasets
- A list of figures that have associated raw data
- A description of any restrictions on data availability

DNA methylation (n=565), RNA sequencing (n=185), and single-cell RNA sequencing data (n=8 meningioma samples, n=2 dura samples) of new samples reported in this manuscript have been deposited in the NCBI Gene Expression Omnibus under the accession GSE183656 (https://www.ncbi.nlm.nih.gov/geo/query/acc.cgi?acc=GSE183656). Additional RNA sequencing data from previously reported meningiomas (n=15) from the discovery cohort are available under the accession GSE101638 (https://www.ncbi.nlm.nih.gov/geo/query/acc.cgi?acc=GSE101638). Whole exome sequencing, ChIP sequencing, and additional DNA methylation profiling data incorporated into this study were derived from previously reported and deposited meningiomas in GSE101638 (https://www.ncbi.nlm.nih.gov/geo/query/acc.cgi?acc=GSE101638), GSE139652 (https://www.ncbi.xyz/geo/query/acc.cgi?acc=GSE139652), and . The publicly available GRCh38 (hg38, https://www.ncbi.nlm.nih.gov/assembly/GCF_000001405.39/) and CRCh37.p13 datasets (hg19, https://www.ncbi.nlm.nih.gov/assembly/GCF_000001405.25/) were used in this study.

# Field-specific reporting

Please select the one below that is the best fit for your research. If you are not sure, read the appropriate sections before making your selection.

☒ Life sciences  ☐ Behavioural & social sciences  ☐ Ecological, evolutionary & environmental sciences

For a reference copy of the document with all sections, see nature.com/documents/nr-reporting-summary-flat.pdf

# Life sciences study design

All studies must disclose on these points even when the disclosure is negative.

| | |
|---|---|
| Sample size | No statistical methods were used to predetermine clinical sample sizes, but our discovery and validation cohort sizes are similar or larger to those reported in previous publications. All experiments were performed with independent biological replicates (2-3 biological replicates for molecular or cell biology experiments, and 7+ biological replicates for animal experiments). In our experience and in the experience from previous publications, these samples sizes provide sufficient resolution to resolve biologically-meaningful differences between conditions tested using molecular, cellular, or animal techniques. To validate this approach, all experiments were repeated, and statistics were derived from biological replicates (rather than technical replicates). Biological replicates are indicated in each panel or figure legend. Data distribution was assumed to be normal, but this was not formally tested. |
| Data exclusions | No clinical, molecular, cellular, or animal data points were excluded from the analyses. |
| Replication | All experiments were performed with at least 3 biologic replicates. All attempts at replication were successful. |
| Randomization | This was a retrospective non-randomized study of human tumor samples with no intervention. All samples were interrogated equally. Thus, controlling for covariants among clinical samples is not relevant. Cells, organoids, and animals were randomized across experimental conditions, but pre-treatment tumor sizes and other potentially-confounding covariates were controlled across conditions before experimentation. |
| Blinding | Investigators were blinded to conditions during clinical data collection and analysis of mechanistic or functional studies. Bioinformatic analyses were performed blind to clinical features, outcomes, or molecular characteristics. |

# Reporting for specific materials, systems and methods

We require information from authors about some types of materials, experimental systems and methods used in many studies. Here, indicate whether each material, system or method listed is relevant to your study. If you are not sure if a list item applies to your research, read the appropriate section before selecting a response.

### Materials & experimental systems

| n/a | Involved in the study |
|---|---|
| ☐ | ☒ Antibodies |
| ☐ | ☒ Eukaryotic cell lines |
| ☒ | ☐ Palaeontology and archaeology |
| ☐ | ☒ Animals and other organisms |
| ☐ | ☒ Human research participants |
| ☒ | ☐ Clinical data |
| ☒ | ☐ Dual use research of concern |

### Methods

| n/a | Involved in the study |
|---|---|
| ☒ | ☐ ChIP-seq |
| ☒ | ☐ Flow cytometry |
| ☒ | ☐ MRI-based neuroimaging |

# Antibodies

| | |
|---|---|
| Antibodies used | -Merlin (#ab88957, clone AF1G4, Abcam, 1:2000)<br>-GAPDH (#MA515738, clone GA1R, Thermo Fischer Scientific, 1:2000)<br>-Caspase-7 (#9492, Cell Signaling, 1:500)<br>-IRF8 (#5628S, clone D20D8, Cell Signaling, 1:500)<br>-Tubulin (#T5168, clone B-5-1-2, Sigma, 1:5000)<br>-HH3 (#702023, clone 17H2L9, Thermo Fischer Scientific, 1:1000)<br>-FLAG (#F1804, clone F1804, Sigma, 1:1000)<br>-ARHGAP35 (#2860, clone C59F7, Cell Signaling, 1:1000)<br>-FOXM1 (#sc-376471, clone G-5, Santa Cruz, 1:500)<br>-pRB-S780 (#8180P, clone D59B7, Cell Signaling, 1:1000)<br>-pRB-S807/811 (#8516P, clone D20B12, Cell Signaling, 1:1000)<br>-Anti-mouse HRP-conjugated secondary antibody (#7076, Cell Signaling, 1:2000)<br>-Anti-rabbit HRP-conjugated secondary antibody (#7074, Cell Signaling, 1:2000)<br>-LYVE-1 (#ab14917, Abcam, 1:1000)<br>-PROX-1 (#AF2727, R&D Systems, 1:1000)<br>-Anti-rabbit Alexa Fluor secondary antibody (#A21206, Thermo Fischer Scientific, 1:1000)<br>-Anti-goat Alexa Fluor secondary antibody (#A21469, Thermo Fischer Scientific 1:1000)<br>-FOXM1 (#ab207298, clone EPR17379, Abcam, 1:600)<br>-Ki-67 (#790-4286, clone 30-9, Ventana, 1:6)<br>-cleaved Caspase-3 (#9664, clone 5A1E, Cell Signaling, 1:2000)<br>-CD3 (#A0452, Agilent Technologies, 1:200) |
| Validation | -Merlin: Knockout validated for human immunoblots.<br>-GAPDH: Knockout validated for human immunoblots.<br>-Caspase-7: Validated +/- apoptosis induction for human immunoblots.<br>-IRF8: Validated +/- IFN stimulation for human immunoblots. An unknown background band is detected at 80 kDa in some cell lines.<br>-Tubulin: Validated for human immunoblots using recombinant expressed antibodies, genetic strategies, independent antibody verification, RNA sequencing, functional assays, expression/overexpression, and immunocapture followed by mass spectrometry.<br>-HH3: Validated for human immunoblots using subcellular fractionation.<br>-FLAG: Validated for human immunoblots and immunoprecipitation using affinity purification and competition assays.<br>-ARHGAP35: Validated for human immunoblots using immunoprecipitation<br>-FOXM1: Validated for human immunoblots using overexpression<br>-pRB-S780: Validated for human immunoblots using phosphorylated or nonphosphorylated recombinant truncated Rb with or without RB blocking peptides.<br>-pRB-S807/811 : Validated for human immunoblots using phosphorylated or nonphosphorylated recombinant truncated Rb with or without RB blocking peptides.<br>-Anti-mouse HRP-conjugated secondary antibody: Validated for human immunoblots using affinity purification and competition assays.<br>-Anti-rabbit HRP-conjugated secondary antibody: Validated for human immunoblots using affinity purification and competition assays.<br>-LYVE-1: Knockout validated for human imunofluorescence.<br>-PROX-1: Validated for human immunofluorescence using subcellular localization.<br>-Anti-rabbit Alexa Fluor secondary antibody: Validated for human immunofluorescence using affinity purification.<br>-Anti-goat Alexa Fluor secondary antibody: Validated for human immunofluorescence using affinity purification.<br>-FOXM1: Validated for human immunofluorescence using recombinant expressed antibodies.<br>-Ki-67: Validated for human immunohistochemistry and immunofluorescence using proliferating versus non-proliferating tissues.<br>-cleaved Caspase-3: Validated +/- apoptosis induction for human immunohistochemistry<br>-CD3: Validated for human immunohistochemistry using T-cell versus B-cell lines. |

# Eukaryotic cell lines

Policy information about cell lines

| | |
|---|---|
| Cell line source(s) | HEK293T cells were obtained from ATCC. CH-157MN, IOMM-Lee, DI-98, DI-134, MSC1, and M10G primary meningioma cell lines were obtained from collaborators or derived from patient tumor samples and described in previous studies, as referenced in the Methods section. |
| Authentication | Meningioma cell lines were authenticated using DNA methylation profiling and CNV analyses to confirm concordance to tumors of origin, most recently in 2021. Non-meningioma cell lines purchased from reputable commercial suppliers (HEK293T cells from ATCC) were not authenticated. |
| Mycoplasma contamination | All cell lines tested negative for mycoplasma. |
| Commonly misidentified lines<br>(See ICLAC register) | No commonly misidentified cell lines were used in this study. |

## Animals and other organisms

Policy information about <u>studies involving animals</u>; <u>ARRIVE guidelines</u> recommended for reporting animal research

| | |
|---|---|
| Laboratory animals | 5-6 week old female NU/NU mice purchased from Harlan Sprague Dawley for this study. All animal care and experimental procedures were in accordance with federal policies and guidelines governing the use of animals and were approved by the University of California San Francisco's (UCSF) Institutional Animal Care and Use Committee (IACUC). The IACUC is in full compliance with the 8th edition of The Guide for the Care and use of Laboratory Animals. UCSF has an AAALAC accredited animal care and use program. Mice were housed in solid-bottomed cages containing autoclaved paper chips in individually ventilated cages. Animals had continuous access to irradiated food and water purified by reverse osmosis and UV lighting. The housing room was maintained at 68 to 74º Fahrenheit with 30-70 % relative humidity. All cages were maintained in a SPF barrier facility from which dirty bedding sentinel mice were tested quarterly. All sentinels were found to be seronegative for mouse hepatitis virus, pneumonia virus of mice, mouse parvovirus, minute virus of mice, epizootic diarrhea of infant mice, Theiler's murine encephalomyelitis virus, ectromelia and were free of ectoparasites and endoparasites. Mice were observed daily by animal care staff for any clinical abnormalities. |
| Wild animals | Study did not involve wild animals. |
| Field-collected samples | Study did not involve samples collected in the field. |
| Ethics oversight | Study was approved by the UCSF Institutional Animal Care and Use Committee (AN174769). |

Note that full information on the approval of the study protocol must also be provided in the manuscript.

## Human research participants

Policy information about <u>studies involving human research participants</u>

| | |
|---|---|
| Population characteristics | Patients undergoing resection of meningioma at UCSF or HKU of all ages, genders, past and current diagnosis and treatment categories were included. Covariates are summarized in Supplementary table 1, and are recapitulated here:<br><br>Patients: 565<br>Median age: 58 years<br>Median follow-up: 5.6 years<br>Male:Female (ratio): 193:372 (1:1.93)<br>Recurrences: 161<br>Extent of resection<br> Gross total: 394 (70%)<br> Near total: 171 (30%)<br>WHO grade<br> 1 : 388 (69%)<br> 2 (atypical): 142 (25%)<br> 3 (anaplastic): 35 (6%) |
| Recruitment | As part of routine clinical practice at UCSF and HKU institutions, all patients undergoing craniotomy for tumor resection sign a waiver of informed consent to contribute de-identified data to research projects. Thus, there was no self-selection bias or other biases that may influence or impact our results. Meningioma samples for the discovery cohort were selected from the UCSF Brain Tumor Center Biorepository and Pathology Core in 2017, with an emphasis on high-grade meningiomas and low-grade meningiomas with long clinical follow-up. All WHO grade 2 and grade 3 meningiomas with available frozen samples were included. For WHO grade 1 meningiomas, frozen samples in the tissue bank were cross-referenced for clinical follow-up data from a retrospective institutional meningioma clinical outcomes database, and all cases with available frozen tissue and clinical follow-up greater than 10 years (n=40) were included. To achieve a discovery cohort of 200 cases, additional WHO grade 1 meningiomas with available frozen tissue and the longest possible clinical follow-up (albeit less than 10 years, n=47) were included. The electronic medical record was reviewed for all patients in late 2018, and paper charts were reviewed in early 2019 for patients treated prior to the advent of the electronic medical record. All available clinical pathology material was reviewed for diagnostic accuracy by a board-certified neuropathologist (D.A.S.). WHO grading was performed using contemporary criteria outlined in the WHO classification of tumors of the central nervous system. Cases for which other tumors remained in the differential diagnosis (such as schwannoma or solitary fibrous tumor/hemangiopericytoma) were excluded. The validation cohort was comprised of 365 consecutive meningiomas from patients who were treated at The University of Hong Kong (HKU) from 2000 to 2019 that had frozen tissue suitable for DNA methylation profiling. The medical record was reviewed for all patients in late 2019. For the discovery and validation cohorts, meningioma recurrence was defined as new radiographic tumor on magnetic resonance imaging after gross total resection, or enlargement/progression/ growth of residual tumor on magnetic resonance imaging after subtotal resection. All magnetic resonance imaging studies in the discovery cohort were reviewed for accuracy and meningioma location by a board-certified radiologist with a Certificate of Added Qualification in Neuroradiology (J.E.V-M.) (Supplementary note). Nomograms integrating clinical and molecular features influencing meningioma outcomes were developed to guide clinical translation of meningioma DNA methylation groups (Supplementary note). |
| Ethics oversight | This study complied with all relevant ethical regulations and was approved by the UCSF Institutional Review Board (IRB #13-12587, #17-22324, #17-23196, and #18-24633), and by the HKU Institutional Review Board (UW 07-273 and UW 21-112). Meningiomas and de-identified clinical information were transferred from HKU to UCSF in 2019 for analysis under protection of a Material Transfer Agreement that was certified by both institutions. |

Note that full information on the approval of the study protocol must also be provided in the manuscript.

