## [Peer Review File · Nature Genetics]

Peer Review Information

Manuscript Title: Within-sibship genome-wide association analyses decrease bias in estimates of direct genetic effects

Corresponding author name(s): Dr Laurence Howe

Reviewer Comments & Decisions:

Decision Letter, initial version:
--

13th Apr 2021

Dear Dr Howe,

Your Letter, "Within-sibship GWAS improve estimates of direct genetic effects" has now been seen by 2 referees. You will see from their comments below that while they find your work of interest, some important points are raised. We are interested in the possibility of publishing your study in Nature Genetics, but would like to consider your response to these concerns in the form of a revised manuscript before we make a final decision on publication.

As you will see from these comments, both referees have identified aspects of the analyses and the presentation that need to be improved. We therefore invite you to revise your manuscript taking into account all reviewer comments. Please highlight all changes in the manuscript text file. At this stage we will need you to upload a copy of the manuscript in MS Word .docx or similar editable format.

*2) If you have not done so already please begin to revise your manuscript so that it conforms to our

Letter format instructions, available

[here](http://www.nature.com/ng/authors/article_types/index.html).

*3) Include a revised version of any required Reporting Summary:

[REDACTED]

We hope to receive your revised manuscript within eight to ten weeks. If you cannot send it within this time, please let us know.

Sincerely,

Wei

Wei Li, PhD
Senior Editor
Nature Genetics
One New York Plaza, 47th Fl.
New York, NY 10004, USA

www.nature.com/ng

Reviewers' Comments:

Reviewer #1:

Remarks to the Author:

This is an important paper that compares estimates of genetic association from a "conventional" GWAS to estimates derived from within-family GWAS. The comparison is performed for over 20 phenotypes, using harmonized procedures that hold constant things such as variable coding, quality control and the estimation sample. There are many scientific questions that we'd be able to answer more convincingly if we knew more about how family-based GWAS parameters compare to conventional GWAS parameters. To date, the main hurdle has been sample size. The authors of this study have put a lot of effort into gathering large samples for many phenotypes. For that reason alone, the paper represents a major advance. Provided the authors make good on their promise to make the summary statistics public, I absolutely think a revised version of the paper should be published.

There is much to like about the paper. The harmonized analysis protocol rules out many common concerns about "artificial" sources of differences between conventional GWAS parameters and within-family parameters. The samples are large, and the basic estimation framework is credible and transparent.

My main criticism is that the exposition needs more work. I also think some of the findings are oversold. I don't have any concerns that couldn't be addressed with a bit more work. I am convinced that this is a paper with solid fundamentals - I just wish the authors had put as much effort into producing a well-crafted manuscript that they put into the underlying analyses (which are excellent and very comprehensive).

Specific Comments (in no particular order)

1. The title does not relay what I consider to be the paper's main contribution - I would cite prior work showing deflation of within-family coefficients for many traits (e.g. by Kong, Young and many others) more prominently and try to say more about how the new results compare quantitatively to the previous ones. It was already known from earlier work that family-based associations tend to be appreciably smaller for most (all?) of the phenotypes highlighted in the abstract's sentence about overestimation. I think the contribution here is the systematic comparison of "population" and family-based estimates of genetic associations for 25 separate phenotypes in samples that, by today's standards, are impressively large. The larger sample enables more informative SNP-level analyses and more comprehensive cross-phenotype comparisons.

2. As the authors are well aware, the claims in the opening sentence of the abstract ("Estimates ... represent a combination of ...") are not universal truths. The claims are true under the assumptions of a specific model of how differences between conventional and within-family GWAS parameters arise. I could write down a different model where the differences arise for entirely different reasons. I

personally believe that the model the authors are using to interpret the evidence is very reasonable and captures the most important forces. But I think it'd be useful to use more guarded language in several places where you are making claims that are only valid under the assumptions of your model ("Viewed through the lens of our model, these results mean that XXXX" rather than "These results demonstrate...") and acknowledge alternative interpretations. Similarly, the MS sometimes reads as if it is obvious that within-family associations are unbiased estimates of direct effects. (If we define a direct effect as the within-family association, then this is trivially true!) Other times, the authors make it quite clear that they are thinking of a direct effect as something conceptually distinct from the within-family association (e.g. the example of genetic nurture effects from sibs in the conclusion). I think the authors are informally thinking of a direct effect as as something akin to Fisher's average effect of a gene substitution (but unfortunately, they never make it clear exactly what they have in mind).

I am not asking for anything more than a careful rewrite that is a bit more precise and nuanced. The main thing is to make sure you explicitly lay out any unstated premises behind key conclusions. I would also recommend going through the entire manuscript and making sure that every time terms such as bias or biased are invoked, there is no ambiguity about what the estimand is. If it is not clear what the estimand is, it doesn't make sense to talk about bias.

3. It is very interesting to compare genetic correlations estimated from the two sets of summary statistics. Ultimately, my reading of the evidence is that the paper only makes limited progress in this area. The main problem appears to be that the sample sizes are still not large enough to ensure reasonable power to detect plausible differences between within-family and conventional r_g s. In reality, the authors had $n=25$ phenotypes, so they could have tried to estimate $n(n-1)/2 = 300$ pairwise correlations. As far as I can tell, the authors did not even think it worthwhile to analyze the majority of them. And they may well be right that most of the 300 potential tests would not have been informative. That is fine, but could the authors please acknowledge this clearly and explicitly?!

In practice, they only estimated 22 correlations, limiting their analyses to the pairwise genetic correlation between education and 22 "phenotypes with sufficient heritability". Sufficient heritability is not clearly defined but it would appear a phenotype has sufficient heritability if both estimates of h^2 (based either on conventional or family-based summary statistics) are positive (surely the condition given in the text is not quite right, since the ratio is positive also if both point estimates are negative). The main text claims that the null that the two r_g s are the same can be rejected in three out of the 22 cases (and the language in the abstract implies the difference is substantial). The three are:

Education-Height -> r_g went from 0.16 to -0.02 ($p = 0.004$)
 Education-BMI -> r_g went from -0.32 to 0.01 ($p = 0.000069$)
 Education-CRP -> r_g went from -0.47 to 0.04 ($p = 0.012$)

As far as I can tell, the p -values reported in the text are all nominal, and only the BMI correlation remains significant after a Bonferroni adjustment. If so, this should be clarified. All within-family estimates of r_g s have substantial standard errors and I think it is a bit strong to assert that the genetic correlations "disappear". The evidence is certainly consistent with the hypothesis that the within-family r_g s are in some small neighborhood around zero, whereas conventional ones are not. But the evidence is also consistent with r_g s outside the same small neighborhood around zero. We just have no way of knowing until the samples get even bigger. That is not the authors' fault. All I ask is that they don't overstate how much we learn from these data alone (or explain to me why my

reading of the evidence is too skeptical).

A final remark about the r_{gs} : on my first reading of the paper, I struggled to reconcile the empirical findings of uniform (or near-uniform) SNP-level shrinkage with claims that conventional and within-family r_{gs} sometimes differ substantially. Presumably, the r_{gs} would be identical if shrinkage were exactly uniform? I eventually concluded that the main problem was that the authors overstate how strong the evidence for substantial differences in r_{gs} actually is.

In principle, there could be other explanations.

A first possibility is that there is substantial shrinkage heterogeneity for BMI, but not education. Is that plausible? (I suspect not, but wonder if the authors can look into it more systematically.)

A second possibility is that, even for educational attainment, the tests of heterogeneity are not very well-powered. If so, it is possible that the actual amount of heterogeneity is large enough to shift true r_{gs} from, say, approximately 0.15 to approximately 0. Yet, the tests of heterogeneity fail to reject. That would explain the apparent contradiction. If so, the authors should not characterize the evidence of heterogeneity as "minimal".

4. The writing could be a lot more precise in a number of places. Let me just give some examples:

i. It is hard for readers to gauge the sample sizes in the various analyses. For some reason, the authors insist on giving the number of siblings included in their largest GWAS. It'd be much more informative to give a range, a mean and the median. And then try to provide more information about the sample size underlying specific analyses whose results are referenced in the text.

ii. "The common environment terms from classical twin studies..." - the argument made here is a little vague. You could have a high c^2 without any indirect effects.

iii. "We demonstrate..."

iv. Characterizing within-family estimates as more "accurate" is strange since we usually define accuracy as something depends both on bias and variance. The within-family estimates may be less biased (even though that is not guaranteed) but they will also have worse precision. So it is not at all clear that their accuracy is better.

v. Whenever you can replace a qualitative claim by a quantitative claim, consider doing so.

5. Could the authors please invest a couple of hours into preparing more professional-looking figures and tables?

6. The results for height in the polygenic adaptation analyses are very interesting. Some may not find the evidence quite strong enough to settle definitively a debate that has now been raging for several years. But I'd say the results are very close to dispositive. It looks like the original claims of adaptation were qualitatively correct but that the estimates originally reported were contaminated by stratification bias (as alleged by skeptics).

Reviewer #2:

Remarks to the Author:

Within-sibship GWAS improve estimates of direct genetic effects

The authors have addressed a long-standing problem in GWAS by achieving a sibship sample size large enough to generate association estimates that have reduced bias from demography and indirect genetic effects. They find, somewhat expectantly, that the most affected traits among the 25 studied are generally those with more social determinants. Thus, the importance of the paper lies in the fact that they have empirically demonstrated the magnitude of these potential biases and outlined which kind of traits we should worry about when considering these biases within the context of GWAS and MR.

What is more impressive about the paper is the magnitude of bias from demography and indirect genetic effects for some traits. For example, the within-sibship SNP heritability for educational attainment was 0.04, compared to 0.14 for the population estimate. Of equal importance to the field is that heritability estimates for many traits don't change importantly. These findings have important implications for understanding the social determinants of health.

I am quite enthusiastic about this manuscript, given that it addresses an important set of questions for the field, but do feel that there are some limitations.

Major

-The authors should clearly state what model they are using for the standard population GWAS model and what software was used in the Results. It seems that the authors did this in R using a linear regression. However, this does not seem to be a standard GWAS software package that the field would be familiar with. Why wouldn't they compare their results with SAIGE, for example, which accounts properly for relatedness?

-Using a software package that can better account for distant relatedness could strongly influence their results in my experience.

-The authors have used LDSR to estimate heritability. This assumes that each SNP in the genome is expected to contribute equally to heritability. As has been shown by Balding, this tends to underestimate SNP heritability. The authors should reconsider estimating h^2 using SumHer (Nature Genetics, 2019).

-A main message of the paper in my reading is how little MR estimates are influenced by demography and indirect effects. It appears that there are relevant biases introduced only when the outcome phenotype is strongly influenced by these sources of biases. However, when only the exposure is influenced by these biases, there is little change in the MR estimate. This message should be emphasized to the reader in the abstract

-Interestingly when inspecting Figure 2, it becomes apparent that the shrinkage estimates tend to exclude the null when the p-value threshold is relaxed to E^{-5} for many phenotypes (depressive symptoms and cognition are two examples). This is not apparent when reading the text. This means to me that the bias is more prominent in less strongly associated SNPs. Why is this?

-I disagree with the author's overall conclusions. Should we really use within sibship analyses to provide more accurate estimates of direct genetic effects for most phenotypes? Many GWAS'd phenotypes are unlikely to be affected by these biases in a relevant way, given that most GWAS'd phenotypes are biomedically-relevant traits and diseases. Wouldn't it be more sensible to consider within sibship analyses for traits that are strongly influenced by social/behavioural determinants?

Would the authors suggest using within sibship GWAS for an MR of the effect of serum protein levels on a disease? I don't think they would. I therefore strongly feel their conclusion needs more nuance. -Similarly, the authors conclude that future MR studies could use within-sibship estimates to identify potential bias. Yes, this is true, but is this needed for most traits? I think their data suggest that for many traits, population GWAS for MR studies is sufficient and likely more precise, given the larger sample sizes.

Minor

-It would be helpful for the authors to provide an example of the mean genotype within the sibship. This is not a concept familiar to many readers of Nature Genetics.

-Define "shrinkage" in the results please.

-It is likely my ignorance, but what the authors state that they calculated the mean family genotype for sibship i over n siblings, they must be modeling the dose of a specific allele? Not the actual genotype? Please clarify

Author Rebuttal to Initial comments

2nd July 2021

Wei Li

Senior Editor

Nature Genetics

Dear Dr Li,

I'm writing regarding the manuscript "*Within-sibship GWAS improve estimates of direct genetic effects*". We would like to thank the journal, editors and reviewers for their positive feedback on the manuscript and the opportunity to revise and resubmit.

In response to the reviewer comments, we have made extensive changes to the manuscript, which are detailed below. We have also rerun the meta-analyses to include additional GWAS data from the Swedish Twin Registry, Minnesota Twins and the Norwegian Mother, Father and Child Cohort Study (MoBa). The additional data have increased meta-analysis sample sizes by 5-10% which resulted in minor differences to the reported results. We have added co-authors from these new studies to the manuscript.

We are happy to make further edits to the manuscript to meet the guidelines of the “letter” format. However, the additional details and analyses added to the manuscript in response to reviewer comments have resulted in a higher word count. Could the editor please advise the best course of action with regard to reformatting. For example, what is the recommended compromise between meeting the word count requirements and ensuring that all important points are covered in the manuscript.

We hope that the revised manuscript is suitable for publication in *Nature Genetics*. Thank you for considering our manuscript.

Yours sincerely,

Dr Laurence Howe on behalf of the authors

Reviewers' Comments:

Reviewer #1:

Remarks to the Author:

This is an important paper that compares estimates of genetic association from a "conventional" GWAS to estimates derived from within-family GWAS. The comparison is performed for over 20 phenotypes, using harmonized procedures that hold constant things such as variable coding, quality control and the estimation sample. There are many scientific questions that we'd be able to answer more convincingly if we knew more about how family-based GWAS parameters compare to conventional GWAS parameters. To date, the main hurdle has been sample size. The authors of this study have put a lot of effort into gathering large samples for many phenotypes. For that reason alone, the paper represents a major advance. Provided the authors make good on their promise to make the summary statistics public, I absolutely think a revised version of the paper should be published.

There is much to like about the paper. The harmonized analysis protocol rules out many common concerns about "artificial" sources of differences between conventional GWAS parameters and within-family parameters. The samples are large, and the basic estimation framework is credible and transparent.

My main criticism is that the exposition needs more work. I also think some of the findings are oversold. I don't have any concerns that couldn't be addressed with a bit more work. I am convinced that this is a paper with solid fundamentals - I just wish the authors had put as much effort into producing a well-crafted manuscript that they put into the underlying analyses (which are excellent and very comprehensive).

We thank the reviewer for their positive feedback on the project and their helpful comments. We have responded to individual comments below.

Specific Comments (in no particular order)

1. The title does not relay what I consider to be the paper's main contribution - I would cite prior work showing deflation of within-family coefficients for many traits (e.g. by Kong, Young and many others) more prominently and try to say more about how the new results compare quantitatively to the previous ones. It was already known from earlier work that family-based associations tend to be appreciably smaller for most (all?) of the phenotypes highlighted in the abstract's sentence about overestimation. I think the contribution here is the systematic comparison of "population" and family-based estimates of genetic associations for 25 separate phenotypes in samples that, by today's standards, are impressively large. The larger sample enables more informative SNP-level analyses and more comprehensive cross-phenotype comparisons.

Response: As the reviewer noted, previous research (e.g. by Kong, Young and others) has provided evidence that within-family estimates are smaller for many of the highlighted phenotypes such as height and educational attainment. We therefore agree that perhaps the most important contribution of our work is the analysis of 25 phenotypes in much larger samples than previously studied. A further strength of our study is the inclusion of a wider range of cohorts and populations. Previous research on family-based GWAS estimates for complex traits has largely been based on a handful of studies (e.g. UK Biobank, deCODE, QIMR, STR). We combined many of these previously used datasets and also included large numbers of samples that have not been previously used in family GWAS studies such as the Norwegian HUNT Study and China Kadoorie Biobank. This allowed us to provide evidence that shrinkage on height and educational attainment is generally consistent across European studies (Supplementary Tables 1 and

2) but that there is limited evidence for shrinkage on height in China Kadoorie Biobank (Supplementary Tables 3).

We have made the following changes to clarify the major contributions of our manuscript and discuss our findings with regard to previous studies:

a) We have changed the manuscript title from “Within-sibship GWAS provide improved estimates of direct genetic effects” to “Within-sibship GWAS of 25 phenotypes improve estimates of direct genetic effects” to emphasis the relatively large number of phenotypes analysed.

b) To better illustrate what was previously known regarding within-sibship shrinkage, we have added the following text about the 2018 Educational attainment GWAS to the “Main” section on page 5.

“For example, Lee et al [14] found that UK Biobank within-sibship GWAS estimates for educational attainment variants were around 40% lower than estimates from unrelated individuals indicating effects of demography and indirect genetic effects.”

c) We have made edits to the abstract, “Main” and discussion sections to acknowledge that our estimates of within-sibship are broadly consistent with analyses conducted in previous studies in smaller datasets.

Abstract (page 4):

Before: “We demonstrate that existing GWAS associations for ... overestimate direct effects.”

After: “Consistent with previous studies, within-sibship GWAS estimates were smaller than corresponding population estimates for”

Main (page 5):

Before: “We demonstrate that population GWAS estimates for... partially reflect demography and indirect genetic effects, which affect downstream analyses such as estimates of heritability and genetic correlations.”

After: Our results are broadly consistent with previous studies comparing population and within-sibship GWAS/ PRS estimates in smaller sample sizes [13, 14, 19, 37]. We

found that within-sibship meta-analysis GWAS estimates are smaller than population estimates for 7 phenotypes....

We show that these differences, which are likely to partially reflect demography and indirect genetic effects, can affect downstream analyses such as estimates of heritability, genetic correlations, and Mendelian randomization.

Discussion (page 8):

Before: "Our results demonstrate that GWAS results and downstream analyses of behavioural phenotypes (e.g. educational attainment, smoking behaviour) as well as some biologically proximal phenotypes (e.g. height, BMI) are likely to be affected by demography and indirect genetic effects."

After: "Here we report results from the largest within-sibship GWAS to date which included 25 phenotypes and combined data from 178,076 siblings. Consistent with previous studies [13, 14, 19, 37], we found that GWAS results and downstream analyses of behavioural phenotypes (e.g. educational attainment, smoking behaviour) as well as some biologically proximal phenotypes (e.g. height, BMI) are affected by demography and indirect genetic effects. However, we found that most analyses involving more clinical phenotypes, such as lipids, were not strongly affected."

2. As the authors are well aware, the claims in the opening sentence of the abstract ("Estimates ... represent a combination of ...") are not universal truths. The claims are true under the assumptions of a specific model of how differences between conventional and within-family GWAS parameters arise. I could write down a different model where the differences arise for entirely different reasons. I personally believe that the model the authors are using to interpret the evidence is very reasonable and captures the most important forces. But I think it'd be useful to use more guarded language in several places where you are making claims that are only valid under the assumptions of your model ("Viewed through the lens of our model, these results mean that XXXX" rather than "These results demonstrate...") and acknowledge alternative interpretations. Similarly, the MS sometimes reads as if it is obvious that within-family associations are unbiased estimates of direct effects. (If we define a direct effect as the within-family association, then this is trivially true!) Other times, the authors make it quite clear that they are thinking of a direct effect as something conceptually distinct from the within-family association (e.g. the example of genetic nurture effects from sibs in the conclusion). I

think the authors are informally thinking of a direct effect as something akin to Fisher's average effect of a gene substitution (but unfortunately, they never make it clear exactly what they have in mind).

I am not asking for anything more than a careful rewrite that is a bit more precise and nuanced. The main thing is to make sure you explicitly lay out any unstated premises behind key conclusions. I would also recommend going through the entire manuscript and making sure that every time terms such as bias or biased are invoked, there is no ambiguity about what the estimand is. If it is not clear what the estimand is, it doesn't make sense to talk about bias.

Response:

Absolutely, our interpretation of the results depends on the assumptions of the models we use, and bias must be discussed with respect to the estimand of interest. To improve the manuscript, we have been through and made substantial edits to the language to be more precise and careful. We list specific changes below. In particular, we are more cautious about equating within-sibship/ within-family estimates with direct genetic effects and also endeavour to only refer to “bias” with respect to a specific estimand of interest.

Abstract (page 4):

a) Before: "Estimates ... represent a combination of ..."

After: "Estimates ... capture effects of..."

b) Before: "We demonstrate that existing GWAS associations for ... overestimate direct effects."

After: "Within-sibship GWAS estimates were smaller than corresponding population estimates for ...".

c) Before: "Large-scale family datasets provide new opportunities to quantify direct effects of genetic variation on human traits and diseases."

After: "Large-scale family datasets provide new opportunities to understand the relationships between genetic variation and human traits and diseases."

Main (page 5):

d) Before: “bias downstream analyses using GWAS summary data”

After: “impact downstream analyses using GWAS summary data”

e) Before: “We demonstrate that population GWAS estimates for... partially reflect demography and indirect genetic effects, which affect downstream analyses such as estimates of heritability and genetic correlations.”

After: “Our results are broadly consistent with previous studies comparing population and within-sibship GWAS/ PRS estimates in smaller sample sizes [13, 14, 19, 37]. We found that within-sibship meta-analysis GWAS estimates are smaller than population estimates for 7 phenotypes...We show that these differences, which are likely to partially reflect demography and indirect genetic effects, can affect downstream analyses such as estimates of heritability, genetic correlations, and Mendelian randomization.”

3. It is very interesting to compare genetic correlations estimated from the two sets of summary statistics. Ultimately, my reading of the evidence is that the paper only makes limited progress in this area. The main problem appears to be that the sample sizes are still not large enough to ensure reasonable power to detect plausible differences between within-family and conventional r_{gs} . In reality, the authors had $n=25$ phenotypes, so they could have tried to estimate $n(n-1)/2 = 300$ pairwise correlations. As far as I can tell, the authors did not even think it worthwhile to analyze the majority of them. And they may well be right that most of the 300 potential tests would not have been informative. That is fine, but could the authors please acknowledge this clearly and explicitly?!

Response: We have added the following to the methods section, detailing that we only considered a subset of pairwise genetic correlations involving educational attainment. This is because of priors regarding educational attainment being especially subject to population stratification, assortative mating and indirect genetic effects, and to limit the number of statistical tests.

Page 19 - “We estimated only pairwise genetic correlations between educational attainment and all other phenotypes because of prior evidence that educational attainment is influenced by demography and indirect genetic effects on educational attainment and, given the limited statistical power, to reduce the multiple testing burden.”

In practice, they only estimated 22 correlations, limiting their analyses to the pairwise genetic correlation between education and 22 "phenotypes with sufficient heritability". Sufficient heritability is not clearly defined but it would appear a phenotype has sufficient heritability if both estimates of h^2 (based either on conventional or family-based summary statistics) are positive (surely the condition given in the text is not quite right, since the ratio is positive also if both point estimates are negative). The main text claims that the null that the two r_g s are the same can be rejected in three out of the 22 cases (and the language in the abstract implies the difference is substantial). The three are:

Education-Height -> r_g went from 0.16 to -0.02 ($p = 0.004$)

Education-BMI -> r_g went from -0.32 to 0.01 ($p = 0.000069$)

Education-CRP -> r_g went from -0.47 to 0.04 ($p = 0.012$)

Response: We have clarified in the results that the “sufficient heritability” refers to the h^2 point estimate.

Page 7 - “We used LDSC [23] to estimate cross-phenotype genome-wide genetic correlations (r_g) between educational attainment and 20 phenotypes with sufficient heritability (Population/within-sibship h^2 point estimate > 0) and statistical power.”

As far as I can tell, the p-values reported in the text are all nominal, and only the BMI correlation remains significant after a Bonferroni adjustment. If so, this should be clarified. All within-family estimates of r_g s have substantial standard errors and I think it is a bit strong to assert that the genetic correlations "disappear". The evidence is certainly consistent with the hypothesis that the within-family r_g s are in some small neighborhood around zero, whereas conventional ones are not. But the evidence is also consistent with r_g s outside the same small neighborhood around zero. We just have no way of knowing until the samples get even bigger.

That is not the authors' fault. All I ask is that they don't overstate how much we learn from these data alone (or explain to me why my reading of the evidence is too skeptical).

Response:

The reviewer is correct that we report heterogeneity at nominal significance, which is noted as follows in the methods.

Page 18 - "In the text, we report differences reaching nominal significance (heterogeneity $P < 0.05$)."

We agree that we should be more accurate in our claims so have edited the abstract and results to use more measured language. We have also changed the example in the abstract from height to body mass index as this was the relationship with the strongest evidence of within-sibship attenuation. Furthermore, after rerunning the meta-analysis with additional data, the P-value for the within-sibship attenuation for height is now > 0.05 . This supports the reviewer's point that we should not overinterpret the statistical significance of the genetic correlation analyses.

Abstract (page 4):

Before: "For example, genetic correlations between educational attainment and height largely disappear."

After: "For example, the genetic correlation between educational attainment and BMI attenuated towards zero."

Results (page 7):

Before: "There was strong evidence using population estimates that educational attainment is positively correlated with height ($r_g = 0.16$, 95% C.I [0.10, 0.22]) and negatively correlated with BMI ($r_g = -0.32$, [-0.38, -0.26]) and CRP ($r_g = -0.47$, [-0.69, -0.25]). However, these correlations were negligible when using within-sibship estimates; height ($r_g = -0.02$, [-0.15, 0.10]), BMI ($r_g = 0.01$, [-0.17, 0.19]) and CRP ($r_g = 0.04$, [-0.34, 0.42]) with evidence for differences between population and within-sibship r_g estimates (height difference $P = 4.0 \times 10^{-3}$, BMI difference $P = 6.9 \times 10^{-5}$, CRP difference $P = 0.012$).

These attenuations indicate that genetic correlations between educational attainment and these phenotypes from population estimates are likely to be driven by demography and indirect genetic effects (**Figure 4 / Supplementary Table 6**).“

After: “There was strong evidence using population estimates that educational attainment is negatively correlated with BMI ($r_g = -0.32$, [-0.37, -0.26]), ever smoking ($r_g = -0.41$, [-0.49, -0.34]) and CRP ($r_g = -0.46$, [-0.67, -0.25]). However, these correlations attenuated towards zero when using within-sibship estimates; BMI ($r_g = -0.05$, [-0.22, 0.12]), ever smoking ($r_g = -0.14$, [-0.42, 0.14]) and CRP ($r_g = -0.06$, [-0.43, 0.30]) with some evidence at nominal significance for differences between population and within-sibship r_g estimates (BMI difference $P = 5.3 \times 10^{-4}$, ever smoking difference $P = 0.040$, CRP difference $P = 0.039$). These attenuations indicate that genetic correlations between educational attainment and these phenotypes from population estimates may be inflated by demography and indirect genetic effects (**Figure 4 / Supplementary Table 7**).“

A final remark about the r_{gs} : on my first reading of the paper, I struggled to reconcile the empirical findings of uniform (or near-uniform) SNP-level shrinkage with claims that conventional and within-family r_{gs} sometimes differ substantially. Presumably, the r_{gs} would be identical if shrinkage were exactly uniform? I eventually concluded that the main problem was that the authors overstate how strong the evidence for substantial differences in r_{gs} actually is.

Response: The statement of uniform shrinkage refers only to strongly associated ($P < 1 \times 10^{-5}$) variants, while genetic correlations were calculated using genome-wide data. We agree that the population/within-sibship r_{gs} should be identical if shrinkage was uniform across the genome. In the results we had noted (below quote) that the lack of evidence for shrinkage heterogeneity was specific to the strongest association signals for height and educational attainment.

Page 6 - “This suggests that shrinkage may be largely uniform across these signals for these phenotypes.”

The genetic correlation results (and the Mendelian randomization results of height/BMI on educational attainment) are likely to be indicative of cross-trait shrinkage rather than same-trait shrinkage, i.e. we found little evidence that the association of BMI variants

(e.g. FTO) with BMI shrink in the within-sibship model but the associations of these BMI variants with educational attainment shrink. The BMI variants are unlikely to have been associated strongly enough with educational attainment to be included in the educational attainment shrinkage analyses, so would not have been included in the previously described heterogeneity analyses.

We have made edits to the text to clarify that the shrinkage heterogeneity was on strongly associated variants only.

On page 6:

Before: “In the meta-analysis data, we observed minimal evidence of heterogeneity in shrinkage across individual variants for height and educational attainment, suggesting that shrinkage is largely uniform across the strongest association signals for these phenotypes.”

After: “In the meta-analysis data, we did not observe strong evidence of heterogeneity in shrinkage across variants strongly associated with height and educational attainment. This suggests that shrinkage may be largely uniform across these signals for these phenotypes.”

On pages 8/9:

Before: “We observed minimal evidence of heterogeneity in shrinkage estimates of height and educational attainment genetic variants. This indicates that observed shrinkage is likely to be largely driven by assortative mating or indirect genetic effects since both of these tend to influence associations proportional to the direct effect (whereas population stratification is likely to have larger effects on ancestrally informative markers).”

After: “We found little evidence of heterogeneity in shrinkage estimates of genetic variants strongly associated ($P < 1 \times 10^{-5}$) with height and educational attainment, although power was limited by available samples. The limited detectable heterogeneity could indicate that the observed shrinkage is largely driven by assortative mating or indirect genetic effects. Both of these tend to influence associations proportional to the

direct effect, whereas population stratification is likely to have larger effects on ancestrally informative markers.”

We have also added discussion of the distinction between same-trait and cross-trait shrinkage to the discussion.

Page 9 - “The weak evidence for within-sibship shrinkage in the association between BMI genetic variants and BMI is in contrast to the strong evidence from Mendelian randomization analyses (and genetic correlation analyses) that the association between BMI genetic variants and educational attainment does attenuate. These results indicate cross-trait shrinkage in association estimates for BMI genetic variants even in the absence of same-trait shrinkage.”

In principle, there could be other explanations.

A first possibility is that there is substantial shrinkage heterogeneity for BMI, but not education. Is that plausible? (I suspect not, but wonder if the authors can look into it more systematically.)

Response: This is highly plausible and potentially implied by the genetic correlation results. For example, the association of BMI variants with BMI may not change but the association of educational attainment genetic variants with BMI may shrink.

This could be investigated using an educational attainment polygenic risk score and BMI such as in a Mendelian randomization framework. We considered including this in the current manuscript but decided it was too complicated/nuanced because of the issues of interpreting Mendelian randomization analyses of educational attainment (see “Interpreting Mendelian randomization estimates of the effects of categorical exposures such as disease status and educational attainment” LJ Howe et al Medrxiv 2020). We are currently working on this in a separate manuscript devoted to MR of education. Preliminary results indicate that yes, the association of educational genetic variants with BMI does shrink (by ~50% similar to shrinkage of educational attainment variants on education). Therefore, likely to be genome-wide heterogeneity in shrinkage on BMI as the reviewer suggested, but perhaps not heterogeneity for the top BMI variants.

A second possibility is that, even for educational attainment, the tests of heterogeneity are not very well-powered. If so, it is possible that the actual amount of heterogeneity is large enough to shift true r_g s from, say, approximately 0.15 to approximately 0. Yet, the tests of heterogeneity fail to reject. That would explain the apparent contradiction. If so, the authors should not characterize the evidence of heterogeneity as "minimal".

Response:

We have edited the text as indeed larger sample sizes could be better powered to detect heterogeneity in within-sibship shrinkage across variants.

Discussion (Page 8):

Before: We observed minimal evidence of heterogeneity in shrinkage estimates of height and educational attainment genetic variants.

After: We found little evidence of heterogeneity in shrinkage estimates of height and educational attainment genetic variants, although power was limited by available samples.

4. The writing could be a lot more precise in a number of places. Let me just give some examples:

Response: We have been carefully through the manuscript and endeavoured to improve the precision and clarity of the writing. We have responded to the specific examples the reviewer cited, below.

i. It is hard for readers to gauge the sample sizes in the various analyses. For some reason, the authors insist on giving the number of siblings included in their largest GWAS. It'd be much more informative to give a range, a mean and the median. And then try to provide more information about the sample size underlying specific analyses whose results are referenced in the text.

Response: We have added the following to the results section to provide additional information on the range of GWAS sample sizes across the 25 phenotypes and to clarify that additional information on cohort/phenotype numbers are contained in a supplementary table.

Pages 5/6 - "Sample sizes for individual phenotypes ranged from 13,375 to 163,748 (median: 82,760, mean: 79,794). More information on sample sizes from each cohort and for each phenotype are contained in **Supplementary Table 1.**"

Previously in the results section we had provided information on the sample sizes for the most routinely collected phenotypes such as height, BMI and educational attainment which were the focus of the majority of the analyses (genetic correlation, Mendelian randomization, polygenic adaptation). We have updated this to provide further detail.

Page 6 - "Amongst the phenotypes analysed, the largest available sample sizes in a meta-analysis of European cohorts were for height (N = 149,174), body mass index (BMI) (N = 140,883), educational attainment (N = 128,777), ever smoking (N = 124,791) and systolic blood pressure (SBP) (N = 109,588) (**Supplementary Table 2**), we also report stratified results from non-European samples including 13,856 individuals from China Kadoorie Biobank."

ii. "The common environment terms from classical twin studies..." - the argument made here is a little vague. You could have a high c^2 without any indirect effects.

Response: True, high c does not necessarily indicate indirect genetic effects and so we have edited the text in the discussion accordingly.

Page 9:

Before: "The common environment terms from classical twin studies suggest that there are likely to be indirect genetic effects on... but suggest that the observed shrinkage for height is likely to be a consequence of assortative mating"

After: "Notably, twin studies have indicated effects of the common environment on many of the phenotypes for which we observed shrinkage, such as educational attainment [41], cognitive ability [42] and smoking [43], potentially consistent with

indirect genetic effects of parents. In contrast, twin studies do not find strong evidence for common environmental effects on height, where shrinkage is more likely to be a consequence of assortative mating [10, 43, 44].”

iii. "We demonstrate..."

Response: We have replaced this phrase throughout the manuscript. For example, discussing the findings in terms of evidence: “we strengthen evidence” or “we provide evidence”.

iv. Characterizing within-family estimates as more "accurate" is strange since we usually define accuracy as something depends both on bias and variance. The within-family estimates may be less biased (even though that is not guaranteed) but they will also have worse precision. So it is not at all clear that their accuracy is better.

Response: We have made the following changes to the main and discussion sections to clarify that within-family estimates are less biased rather than more accurate (given the smaller sample sizes of within-family GWAS).

a) Main (page 5):

Before: “Within-family genetic association estimates, such as those obtained from samples of siblings, can provide more accurate estimates of direct genetic effects.”

After: “Within-family genetic association estimates, such as those obtained from samples of siblings, can provide less biased estimates of direct genetic effects because they are unlikely to be affected by demography and indirect genetic effects of parents”

b) Discussion (page 8):

Before: “data from families to provide more accurate estimates of direct genetic effects”

After: “data from families to provide less biased estimates of direct genetic effects”

v. Whenever you can replace a qualitative claim by a quantitative claim, consider doing so.

Response: We have been through the manuscript and made extensive edits to the language in response to the reviewer’s other comments. We hope that these changes will also help to address this comment about qualitative and quantitative claims.

We have aimed to support claims quantitatively in the text using estimates and confidence intervals. However, we have written up our findings in the letter format (which has limited word count) and so have aimed for our writing style to be concise. Therefore, we intend for the tables and figures to be a key aspect of the manuscript for readers to visualise our findings quantitatively. As detailed below in response to Reviewer 1’s 5th comment, we have edited the figures to improve their clarity and presentation.

5. Could the authors please invest a couple of hours into preparing more professional-looking figures and tables?

Response: We have devoted time to improving the presentation of all figures and tables; all figures and tables (including supplementary figures) have been heavily edited. If the manuscript is accepted for publication, we would also be very happy to discuss with the editorial team the presentation of figures and tables.

6. The results for height in the polygenic adaptation analyses are very interesting. Some may not find the evidence quite strong enough to settle definitively a debate that has now been raging for several years. But I'd say the results are very close to dispositive. It looks like the

original claims of adaptation were qualitatively correct but that the estimates originally reported were contaminated by stratification bias (as alleged by skeptics).

Response: Thanks, we agree that our results likely provide some of the strongest evidence to date for adaptation on height. We hope our summary data will be of further use for population / evolutionary genetics analyses.

Reviewer #2:

Remarks to the Author:

Within-sibship GWAS improve estimates of direct genetic effects

The authors have addressed a long-standing problem in GWAS by achieving a sibship sample size large enough to generate association estimates that have reduced bias from demography and indirect genetic effects. They find, somewhat expectantly, that the most affected traits among the 25 studied are generally those with more social determinants. Thus, the importance of the paper lies in the fact that they have empirically demonstrated the magnitude of these potential biases and outlined which kind of traits we should worry about when considering these biases within the context of GWAS and MR.

What is more impressive about the paper is the magnitude of bias from demography and indirect genetic effects for some traits. For example, the within-sibship SNP heritability for educational attainment was 0.04, compared to 0.14 for the population estimate. Of equal importance to the field is that heritability estimates for many traits don't change importantly. These findings have important implications for understanding the social determinants of health.

I am quite enthusiastic about this manuscript, given that it addresses an important set of questions for the field, but do feel that there are some limitations.

We thank the reviewer for their positive feedback on the work. We have responded to the reviewer's concerns below.

Major

-The authors should clearly state what model they are using for the standard population GWAS model and what software was used in the Results. It seems that the authors did this in R using a linear regression. However, this does not seem to be a standard GWAS software package that the field would be familiar with. Why wouldn't they compare their results with SAIGE, for example, which accounts properly for relatedness?

-Using a software package that can better account for distant relatedness could strongly influence their results in my experience.

Response: As the reviewer noted, we used linear regression in R including the first 20 genomic principal components to generate population estimates. This model is broadly equivalent to a PLINK principal component adjusted linear regression model, which readers would be familiar with. The main difference is that siblings were included rather than restricting to unrelated individuals as would be done in a conventional analysis. To account for the non-independence of siblings, we clustered standard errors over sibships. However, we did not account for 2nd/3rd degree relatedness between different sibships (e.g., one pair of siblings are first cousins with another pair of siblings) which resulted in some degree of test statistic inflation in the population model estimates but not the within-sibship estimates.

We considered using linear mixed model approaches (LMMs) such as SAIGE, because these can better account for relatedness. However, because of limitations of LMMs for our study we decided to use the linear regression approach with the clustering of standard errors on sibships.

In response to the reviewer comment, we first outline the practical limitations of LMMs for our analyses. Second, we describe theory and additional analyses using UK Biobank data indicating that 2nd/3rd degree relatedness between different sibships is unlikely to have impacted any of our conclusions. Third, we detail changes we have made to the text to clarify the software and statistical models used for GWAS analyses.

a) Practical limitations of LMMs for within-sibship GWAS

LMMs generally do not work well for smaller sample sizes (e.g. $N < 5000$) as they are optimized for biobank scale analyses of tens or hundreds of thousands of individuals.

For example, from the BOLT-LMM instruction manual:

https://alkesgroup.broadinstitute.org/BOLT-LMM/BOLT-LMM_manual.html

“We recommend BOLT-LMM for analyses of human genetic data sets containing more than 5,000 samples. The algorithms used in BOLT-LMM rely on approximations that hold only at large sample sizes and have been tested only in human data sets.”

Similarly, it is our experience that alternative methods such as FastGWA (which uses a sparse GRM to account for relatedness) will fit a standard linear regression model if the heritability estimate and confidence interval overlaps zero (likely for small GWAS).

Our aim was to generate population and within-sibship estimates from the same samples to enable direct comparisons. Given the scarcity of family data we also wanted to maximise sample sizes by including as many eligible cohorts as possible. Indeed, around half of our cohorts had a maximum sample size less than 5000 and some phenotypes (with missing data) in larger cohorts also had sample sizes less than 5000. To include these studies and maximise sample size, we needed a GWAS approach applicable to sample sizes > 500 (the minimum sample size requirement for a GWAS to be included in the meta-analysis. The linear regression approach we used could be applied to these smaller datasets, but the alternative LMM approach could not.

A further practical limitation of LMM GWAS is that the estimates may not be compatible with LDSC model, which assumes a linear model, because linear mixed model test statistics are also influenced by LD in the formulation. See some discussion of this on the LDSC users forum here. LDSC (h2 and rg) analyses were an important part of our manuscript.

b) Theory and empirical analyses investigating impact on population estimates

Population estimates were used in our study for comparison (in the context of downstream analyses) with within-sibship estimates using the same samples. The most precise SNP-phenotype association estimates come from far larger population based GWAS of unrelated individuals. The within-sibship estimates which are of much more interest to the research community are robust against cryptic relatedness and many forms of demographic and familial biases.

Therefore, the main concern is if cryptic relatedness in population estimates materially affected the inferences of downstream analyses that compared population and within-sibship estimates.

To investigate this, we first explored the extent of 2nd/3rd degree relatedness between different sibships in UK Biobank. We took one sibling from each sibship and then used the precomputed UK Biobank kinships to identify relatedness across different sibships. Two sibships were defined to be related if the kinship between individuals from the different sibships was > 0.088, the lowest value of the reported precomputed UK Biobank kinships. Amongst the 19,588 sibships, we identified 298 instances of pairwise sibship relatedness at this threshold involving 571 unique sibships (around 3% of the sample).

We then performed a sensitivity analysis to evaluate the effects of 2nd/3rd degree relatedness on within-sibship shrinkage in genetic associations. Using individual-level data, we constructed height and educational attainment polygenic risk scores in the UK Biobank siblings (same PRS as described in the text derived in UK Biobank minus the siblings). We then estimated the association between these PRS and the respective phenotype in the population/ within-sibship models and computed the within-sibship shrinkage in the association estimate. The analysis was performed in two samples, the full sample of 19,588 sibships and a sample of 19,017 sibships after excluding all sibships related at 2nd/3rd degree to another sibship. Note that this exclusion is conservative as typically only one individual in a related pair would be excluded.

Perhaps unsurprisingly given that we removed less than 3% of the sample, the within-sibship shrinkage estimates were highly consistent between the full and reduced sample for both height and educational attainment (see table below). These results indicate that the degree of 2nd/3rd degree relatedness between sibships in UK Biobank is unlikely to have affected the within-sibship shrinkage estimates from UK Biobank. UK Biobank is the largest (along with HUNT) contributing study to the meta-analysis suggesting that if the extent of relatedness is similar across studies that the meta-analysis shrinkage estimates are unlikely to be affected.

The extent of 2nd/3rd degree relatedness between sibships may be greater in other studies included in the meta-analysis such as the Norwegian HUNT study but we note

that within-sibship shrinkage estimates for height and educational attainment were consistent across European ancestry cohorts as shown in Supplementary Figures 1/2.

	Shrinkage % before exclusion (SE)	Shrinkage % after exclusion (95% C.I.)
Height	13.2 (1.7)	13.4 (1.8)
Education	56.2 (4.2)	56.0 (4.3)

We next compared genetic correlation estimates using our population model to data to estimates from previous studies which accounted for relatedness (e.g., by removing related individuals). We used the genetic correlation estimates from HAIL, which were estimated from the full UK Biobank sample with related individuals removed (https://ukbb-rg.hail.is/rg_browser/). We found that genetic correlation estimates between educational attainment and height/BMI/ever smoking were similar between HAIL and our population data. Note that the genetic correlation estimates differ slightly quantitatively, especially for smoking, but that the study populations differ between UK Biobank and the within-sibship GWAS which included 19 cohorts.

Phenotype	LDSC genetic correlation with educational attainment: Estimate (95% C.I.)	
	Population (Sibling GWAS)	Population (Hail UK Biobank)
Height	0.16 (0.11, 0.21)	0.23 (0.20, 0.26)
BMI	-0.32 (-0.37, -0.26)	-0.39 (-0.42, -0.35)
Ever Smoking	-0.41 (-0.49, -0.34)	-0.26 (-0.31, -0.21)

Furthermore, as discussed in the introduction, the within-sibship shrinkage results, such as on height and educational attainment, are highly consistent with estimates from previous studies. The population SDS polygenic adaptation estimates for height were also highly consistent with a previous study (<https://elifesciences.org/articles/39725>). The consistency between downstream analyses using population model GWAS data and

previous studies suggests that the 2nd/3rd degree relatedness between sibships is unlikely to have affected our conclusions.

From theory, we would also note that most of the methods used are largely robust against cryptic relatedness or not sensitive to small changes in standard errors that are not correlated with association signal. In LDSC analyses, the intercept will control for cryptic relatedness because inflation from cryptic relatedness should not correlate with LD scores. Cryptic relatedness will lead to inflation in the LD score intercept of the population estimates. In shrinkage, Mendelian randomization and polygenic adaptation analyses, small changes in standard errors which are not strongly correlated with association signal may lead to slightly overestimated precision but are unlikely to affect point estimates as illustrated by the empirical examples above.

c) Changes to the manuscript

To clarify and illustrate how the population estimates are derived, we have made the following changes to the text on pages 5/6 (start of results section).

Before: “For comparison, we also applied a standard population GWAS model that does not account for mean sibship in the same samples.”

After: “For comparison, we also applied a standard population GWAS model; a covariate-adjusted linear regression of the outcome on raw genotype, which does not account for the mean sibship genotype.”

We also added in that this model will adjust for principal components and that all regression models were conducted in R.

Page 6 - “Standard errors were clustered by sibship. Age, sex and principal components were included as covariates in both models.”

“All GWAS analyses were performed separately in individual cohort studies using R (v 3.5.1) and meta-analyses were conducted across these using summary data.”

-The authors have used LDSR to estimate heritability. This assumes that each SNP in the genome is expected to contribute equally to heritability. As has been shown by Balding, this tends to under-estimate SNP heritability. The authors should reconsider estimating h^2 using SumHer (Nature Genetics, 2019).

Response:

SNP heritability estimation methods are sensitive to modelling assumptions. As the reviewer notes, an alternative method (SumHer) which estimates SNP heritability based on the LDK genetic architecture model may provide more accurate SNP heritability estimates than LDSC in empirical data. This method assumes that the contribution of each SNP to heritability depends on allele frequency and local linkage disequilibrium.

The main aim of our manuscript was to compare parameter estimates from downstream analyses using within-sibship and population GWAS data. If the bias in LDSC SNP heritability estimates (e.g. from assuming that each variant contributes equally to heritable) affects the population/within-sibship estimates equally then this difference term will be unbiased.

However, we agree that finding consistent within-sibship SNP heritability attenuations with SumHer would strengthen the evidence by indicating our results are not sensitive to the underlying heritability model. We have therefore run and included SumHer SNP heritability analyses. The SumHer SNP heritability estimates were slightly different (e.g. slightly higher for height) but the SumHer within-sibship attenuations (our main parameter of interest) were highly consistent with the LDSC results. Please see figures on the following page for LDSC and SumHer (LDSC: Figure 3 / SumHer supplemental figure 4) which illustrates the similarities.

On page 7, we added the following to describe the results:

“SNP heritability estimates using SumHer [21] with the LDK-thin model (expected heritability contribution of each SNP is dependent on allele frequencies and local LD) provided consistent evidence for within-sibship attenuations in SNP heritability for

height, educational attainment and cognitive ability (Supplementary Table 6 / Supplementary Figure 4).”

Figure 3 – LDSC SNP h² estimates

Supplementary Figure 4 – SumHer SNP h^2 estimates

-A main message of the paper in my reading is how little MR estimates are influenced by demography and indirect effects. It appears that there are relevant biases introduced only when the outcome phenotype is strongly influenced by these sources of biases.

However, when only the exposure is influenced by these biases, there is little change in the MR estimate. This message should be emphasized to the reader in the abstract.

Response: As the reviewer noted, most of the MR estimates presented in the text were not strongly influenced by demography/IGEs. However, we only estimated effects of height/BMI on 23 outcome phenotypes, so it is currently unclear whether this pattern is

also true for other exposure/ outcome relationships. We would therefore prefer to be cautious in our interpretation of the MR findings and not oversell the results.

If the exposure is influenced by IGEs/demography, then the gene-exposure estimates will shrink. If the gene-outcome estimates shrink by the same amount, then the population/within-sibship MR estimates will be consistent. Depending on the underlying causal model, the MR estimate could be biased if only the exposure is affected. For example, as discussed in previous papers (FP Hartwig et al, Bias in Mendelian randomization due to assortative mating, 2018 *Genetic Epidemiology*; NM Davies et al, Within family Mendelian randomization studies, 2019 *Human Molecular Genetics*), cross-trait assortative mating could induce associations between educational attainment and height genetic variants. If an outcome is influenced by height but not by education, a population MR of education \rightarrow outcome may indicate effects of education just because of the correlations induced by assortative mating.

-Interestingly when inspecting Figure 2, it becomes apparent that the shrinkage estimates tend to exclude the null when the p-value threshold is relaxed to E-5 for many phenotypes (depressive symptoms and cognition are two examples). This is not apparent when reading the text. This means to me that the bias is more prominent in less strongly associated SNPs. Why is this?

Response:

As the reviewer noted, the shrinkage estimates exclude the null for the more liberal threshold ($P < 1 \times 10^{-5}$) but not the genome-wide significance threshold ($P < 5 \times 10^{-8}$) for depressive/cognition. One potential explanation is that IGE/demography have larger effects on certain variants. However, another explanation is that the number of variants in the GWS score is much lower than in the 10^{-5} score for these phenotypes (e.g. < 10 variants for depressive symptoms GWS score). This means that the GWS shrinkage estimates are very imprecise, but the 10^{-5} score has more power to detect shrinkage. The confidence intervals overlap between the 10^{-5} /GWS shrinkage estimates for these phenotypes, so it is difficult to determine if the differences relate to chance or genuine heterogeneity. Future studies with larger datasets will have greater statistical power to test for heterogeneity in shrinkage.

-I disagree with the author's overall conclusions. Should we really use within sibship analyses to provide more accurate estimates of direct genetic effects for most phenotypes? Many GWAS'd phenotypes are unlikely to be affected by these biases in a relevant way, given that most GWAS'd phenotypes are biomedically-relevant traits and diseases. Wouldn't it be more sensible to consider within sibship analyses for traits that are strongly influenced by social/behavioural determinants? Would the authors suggest using within sibship GWAS for an MR of the effect of serum protein levels on a disease? I don't think they would. I therefore strongly feel their conclusion needs more nuance.

Response:

We agree with the reviewer that conventional GWAS are likely to be more useful for biomedical phenotypes that are not strongly impacted by demographic and familial effects. This is precisely the message that we hope the paper expresses. Nevertheless, we have tried to be cautious about concluding when within-sibship GWAS will be useful and when they will not be useful for biomedical phenotypes as despite the large sample we have collated, we do not have enough evidence/data to draw strong conclusions for many phenotypes. For example, it is difficult to control for population stratification of rare variants so within-sibship GWAS (which perfectly control for stratification) may be useful for future studies on rare variants for biomedical phenotypes. Obviously within-family studies would require massive sample sizes for rare variant associations, but this is likely to be feasible in the not-too-distant future. For example, UK Biobank plan to have WGS/WES on the whole cohort which includes many first-degree relatives.

We have made edits to the text to provide nuance to our conclusions.

Discussion (page 8/9):

a) Before: "Future studies should use data from unrelated individuals, to maximise sample size for gene discovery and polygenic prediction, and data from families to provide less biased estimates of direct genetic effects for downstream analyses."

After: "Future studies can use data from unrelated individuals, to maximise sample size for gene discovery and polygenic prediction, and data from families to provide less

biased estimates of direct genetic effects, particularly for phenotypes or analyses sensitive to demography and indirect genetic effects.”

-Similarly, the authors conclude that future MR studies could use within-sibship estimates to identify potential bias. Yes, this is true, but is this needed for most traits? I think their data suggest that for many traits, population GWAS for MR studies is sufficient and likely more precise, given the larger sample sizes.

Response:

We agree with the reviewer that population GWAS estimates are likely to be more useful than within-sibship estimates for many Mendelian randomization analyses because of the increased sample sizes/precision. We see within-sibship Mendelian randomization estimates as a sensitivity analysis (similar to running MR-Egger to test for horizontal pleiotropy) which may be able to detect potential bias from demography/IGEs.

We have edited the discussion to clarify this.

Page 9:

Before: “For future Mendelian randomization studies, within-sibship estimates could elucidate the potential presence of bias”.

After: “In subsequent studies, within-sibship Mendelian randomization could be used as a sensitivity analysis”.

Minor

-It would be helpful for the authors to provide an example of the mean genotype within the sibship. This is not a concept familiar to many readers of Nature Genetics.

Response:

We have added an illustrative example of the mean sibship genotype to the results section to help readers understand the model.

Pages 5/6:

“For example, in a sibling pair where one sibling has 2 risk alleles and the other sibling has 1 risk allele, the mean sibship genotype is 1.5 risk alleles and the individual’s deviations are +0.5 and -0.5 respectively.”

-Define “shrinkage” in the results please.

Response:

We have added a definition of shrinkage into the results section.

Page 6 - “We observed the largest within-sibship shrinkage (% decrease in association estimates from population to within-sibship models) for genetic variants.”

-It is likely my ignorance, but when the authors state that they calculated the mean family genotype for sibship i over n siblings, they must be modeling the dose of a specific allele? Not the actual genotype? Please clarify

Response:

By the mean sibship genotype, we are referring to the mean of the best guess genotype calls across all individuals in a sibship. We have added to the methods section that this pipeline uses best guess genotype calls rather than dosages.

Page 10 - “The pipeline used imputed “best guess” genotype calls rather than dosage data.”

Decision Letter, first revision:

24th Aug 2021

Dear Dr Howe,

Your Letter, "Within-sibship GWAS of 25 phenotypes improve estimates of direct genetic effects" has now been seen by 2 referees. You will see from their comments below that while they find your work of interest, some important points are raised. We are interested in the possibility of publishing your study in Nature Genetics, but would like to consider your response to these concerns in the form of a revised manuscript before we make a final decision on publication.

To guide the scope of the revisions, the editors discuss the referee reports in detail within the team, with a view to identifying key priorities that should be addressed in revision. In this case, we particularly ask that you carefully go over the text to improve the clarity and ensure the writing is as clear and concise as possible, and lighten/remove any overstatements regarding the importance of these findings. We hope that you will find the prioritized set of referee points to be useful when revising your study.

We therefore invite you to revise your manuscript taking into account all reviewer and editor comments. Please highlight all changes in the manuscript text file. At this stage we will need you to upload a copy of the manuscript in MS Word .docx or similar editable format.

*2) If you have not done so already please begin to revise your manuscript so that it conforms to our Letter format instructions, available [here](http://www.nature.com/ng/authors/article_types/index.html). Refer also to any guidelines provided in this letter.

[REDACTED]

We hope to receive your revised manuscript within eight weeks. If you cannot send it within this time, please let us know.

Sincerely,

Wei

Wei Li, PhD
Senior Editor
Nature Genetics
New York, NY 10004, USA
www.nature.com/ng

Reviewers' Comments:

Reviewer #1:

Remarks to the Author:

My basic opinion of the paper has not changed: it has solid fundamentals and I certainly think the findings are important enough to be published in Nat Genet. My previous report suggested some things the authors could do to improve the clarity of the writing and the quality of the exposition. The

revised manuscript contains some improvements along these dimensions, and I appreciate the work the authors have done. That said, I have to confess I had hoped for a more comprehensive overhaul. In my opinion, minor imprecisions in the writing continue to crop up in a number of places, and that is a shame since it likely makes it harder for some readers to appreciate the value of the work (ultimately lowering its impact). Be that as it may... I am confident that even if published in its present form, the work would be very widely read and cited. I commend the analysts who assembled and carefully analyzed all these valuable data for their hard work and their indefatigability.

Reviewer #2:

Remarks to the Author:

I appreciate the effort that has gone into prior questions, which have largely been addressed.

The authors state several times in their response to reviews that they wish not to oversell their results. However, one could argue that they have done just that.

Specifically, they analyze 25 phenotypes and have found importantly different SNP-heritability estimates for 7 phenotypes. However, there is arguably a strong over-representation in social and demographically-influenced traits amongst the 25 that they have sampled, compared to the GWAS literature. The vast majority of GWAS and MR literature focuses on more biologically defined traits. The authors consistently tend to inflate the importance of family-based GWAS and MR, when in fact, the majority of the literature is unlikely to be affected, based on the results that they have generated.

I have suggested this before, but could the authors please provide more nuance to their message? Here's some sentences that are oversold:

"Large-scale family datasets provide new opportunities to understand the relationships between genetic variation and human traits and diseases."

This would be more fairly stated as, "Large-scale family datasets provide new opportunities to understand the relationships between genetic variation and some human traits and diseases that are more likely to be influenced by social and demographic factors."

This is a more balanced sentence: "Our study illustrates the importance of collecting GWAS data from families for understanding the effects of inherited genetic variation, particularly for phenotypes sensitive to assortative mating, population stratification and indirect genetic effects." But again, wouldn't it be more balanced to state that the majority of the human genetics literature would not be different if we started all over again and only did family-based studies?

Do the authors feel that it is reasonable to undertake within-sibship MR for most traits: "In subsequent studies, within-sibship Mendelian randomization could be used as a sensitivity analysis [7, 25]." My take-away from their paper is that in exceptional circumstances one might consider doing so, but that for most of the MR literature, I would not expect a different answer if a within-sib MR study was done.

Author Rebuttal, first revision:

22nd September 2021

Wei Li

Senior Editor

Nature Genetics

Dear Dr Li,

I'm writing regarding the manuscript "*Within-sibship GWAS of 25 phenotypes improve estimates of direct genetic effects*". Thank you for the additional comments on the manuscript and the opportunity to revise and resubmit.

We have revised the manuscript to clarify the descriptions. We have sought to maintain a balance between explaining when within-family designs, such as within-sibship models, will be particularly important, and when population and within-family estimates are likely to be comparable.

We hope that the revised manuscript achieves an accurate and balanced explanation of the importance and relevance of our findings without overclaiming. Thank you for considering our revised manuscript.

Yours sincerely,

Dr Laurence Howe on behalf of the authors

Editor Comments:

To guide the scope of the revisions, the editors discuss the referee reports in detail within the team, with a view to identifying key priorities that should be addressed in revision. In this case, we particularly ask that you carefully go over the text to improve the clarity and ensure the writing is as clear and concise as possible, and lighten/remove any overstatements regarding the importance of these findings. We hope that you will find the prioritized set of referee points to be useful when revising your study.

We would like to thank the editors and the reviewers for their supportive and helpful feedback. We have responded to individual reviewer comments below. We have also been through the manuscript to improve the clarity and conciseness of our reporting.

Reviewers' Comments:

Reviewer #1:

Remarks to the Author:

My basic opinion of the paper has not changed: it has solid fundamentals and I certainly think the findings are important enough to be published in Nat Genet. My previous report suggested some things the authors could do to improve the clarity of the writing and the quality of the exposition. The revised manuscript contains some improvements along these dimensions, and I appreciate the work the authors have done. That said, I have to confess I had hoped for a more comprehensive overhaul. In my opinion, minor imprecisions in the writing continue to crop up in a number of places, and that is a shame since it likely makes it harder for some readers to appreciate the value of the work (ultimately lowering its impact). Be that as it may... I am confident that even if published in its present form, the work would be very widely read and cited. I commend the analysts who assembled and carefully analyzed all these valuable data for their hard work and their indefatigability.

We thank the reviewer for their positive feedback on the manuscript. In response to the comments of both reviewers, we have been through the manuscript to improve the writing. We hope that the updated manuscript addresses the concerns of the reviewer

regarding imprecisions in the writing. Specific edits are detailed below in the response to the second reviewer.

We will happily change any phrase or term that remains unclear.

Reviewer #2:

Remarks to the Author:

I appreciate the effort that has gone into prior questions, which have largely been addressed.

We would like to thank the reviewer for their supportive feedback and help in improving the manuscript.

The authors state several times in their response to reviews that they wish not to oversell their results. However, one could argue that they have done just that.

Specifically, they analyze 25 phenotypes and have found importantly different SNP-heritability estimates for 7 phenotypes. However, there is arguably a strong over-representation in social and demographically-influenced traits amongst the 25 that they have sampled, compared to the GWAS literature. The vast majority of GWAS and MR literature focuses on more biologically defined traits. The authors consistently tend to inflate the importance of family-based GWAS and MR, when in fact, the majority of the literature is unlikely to be affected, based on the results that they have generated.

We aimed to analyse all phenotypes that were widely available in our cohorts, leading to the 25 phenotypes that were analysed. The available phenotypes in participating cohorts meant that we did not have sufficient data to analyse many phenotypes reported in the GWAS literature, such as binary disease outcomes. However, we did

endeavour to include as many clinically relevant phenotypes as possible such as lipids, HbA1c and lung function.

We acknowledge that the available 25 phenotypes may not be representative of the wider range of phenotypes in the GWAS literature. We have revised the manuscript to provide more nuance to our conclusions and to clarify where within family data may be particularly useful.

We have responded below to the specific comments and suggestions. We hope that our edits have improved the manuscript. We will happily change any phrase or term that remains unclear.

I have suggested this before, but could the authors please provide more nuance to their message? Here's some sentences that are oversold:

“Large-scale family datasets provide new opportunities to understand the relationships between genetic variation and human traits and diseases.”

This would be more fairly stated as, “Large-scale family datasets provide new opportunities to understand the relationships between genetic variation and some human traits and diseases that are more likely to be influenced by social and demographic factors.”

We have edited this sentence in the abstract to note that within-family designs are likely to be most useful for phenotypes which are affected by demographic and indirect genetic effects.

Abstract (page 4)

Before: “Large-scale family datasets provide new opportunities to understand the relationships between genetic variation and human traits and diseases.”

After: “Large-scale family datasets provide new opportunities to understand the relationships between genetic variation and phenotypes influenced by demographic and indirect genetic effects.”

We have also edited the following sentence in the abstract to be more cautious in reporting the differences in downstream analyses between population and within-sibship model.

Abstract (page 4)

Before: “Estimates of SNP-heritability, genetic correlations and Mendelian randomization based on GWAS results on these phenotypes substantially differed across population and within-sibship models.”

After: “We also observed some differences between population and within-sibship model estimates in downstream SNP-heritability, genetic correlations and Mendelian randomization analyses.”

This is a more balanced sentence: “Our study illustrates the importance of collecting GWAS data from families for understanding the effects of inherited genetic variation, particularly for phenotypes sensitive to assortative mating, population stratification and indirect genetic effects.” But again, wouldn’t it be more balanced to state that the majority of the human genetics literature would not be different if we started all over again and only did family-based studies?

We agree with the reviewer that many findings from previous genetic studies, especially analyses on clinical phenotypes, are unlikely to have been different if family-based studies were used instead of population studies. A key aim of our study was to investigate when demographic and indirect genetic effects, are, and critically, are not, affecting population estimates. However, while our results demonstrate that some of the phenotypes were unlikely to be affected, other phenotypes were affected.

We also agree with the reviewer that many phenotypes in the wider GWAS literature are unlikely to be affected by demographic and indirect genetic effects. However, we have limited evidence from family-based designs about which phenotypes are influenced by demographic and indirect genetic effects – we only investigated 25 traits. The editors and reviewers have also requested that we should be cautious in the claims made from our results. Therefore, we do not wish to draw definitive inferences about the impact of demographic and indirect genetic effects on the broader GWAS literature comprising thousands of phenotypes. For example, we did not have sufficient data to investigate clinical psychiatric phenotypes such as ADHD, autism and schizophrenia which could plausibly be impacted quite strongly by demographic and indirect genetic effects.

We are clear in the manuscript that population based GWAS of unrelated individuals is likely to be the optimal approach for identifying genetic associations and for genomic prediction. Our evidence suggests that family-based designs are likely to be most useful and important for socioeconomic or behavioural phenotypes.

We have edited the cited sentence slightly to downplay the importance of family data for phenotypes which are not strongly affected by demographic and indirect genetic effects.

Main (page 5)

Before: “Our study illustrates the importance of collecting GWAS data from families for understanding the effects of inherited genetic variation, particularly for phenotypes sensitive to assortative mating, population stratification and indirect genetic effects.”

After: “Our study illustrates the importance of collecting genome-wide data from families to understand the effects of inherited genetic variation on phenotypes which are affected by assortative mating, population stratification and indirect genetic effects.”

We also made edits to the discussion to clarify when family-based GWAS are most useful.

Discussion (pages 8/9)

Before: “Consistent with previous studies [13, 14, 19, 39], we found that GWAS results and downstream analyses of behavioural phenotypes (e.g., educational attainment, smoking behaviour) as well as some biologically proximal phenotypes (e.g., height, BMI) are affected by demography and indirect genetic effects. However, we found that most analyses involving more clinical phenotypes, such as lipids, were not strongly affected. Future studies can use data from unrelated individuals to maximise sample size for gene discovery and polygenic prediction, and data from families to provide less biased estimates of direct genetic effects. This may be particularly important for phenotypes or analyses sensitive to demography and indirect genetic effects such as educational attainment.”

After: “Consistent with previous studies [13, 14, 19, 39], we found that GWAS results and downstream analyses of behavioural phenotypes (e.g., educational attainment, smoking behaviour) as well as some anthropometric phenotypes (e.g., height, BMI) are affected by demographic and indirect genetic effects. However, we found that most analyses involving more molecular phenotypes, such as lipids, were not strongly affected. This suggests that the best strategy for gene discovery and polygenic prediction for these phenotypes remains to maximise sample sizes using unrelated individuals. For phenotypes sensitive to demographic and indirect genetic effects such as educational attainment, family-based estimates are likely to be useful because they provide less biased estimates of direct genetic effects.”

Do the authors feel that it is reasonable to undertake within-sibship MR for most traits: “In subsequent studies, within-sibship Mendelian randomization could be used as a sensitivity analysis [7, 25].” My take-away from their paper is that in exceptional circumstances one might consider doing so, but that for most of the MR literature, I would not expect a different answer if a within-sib MR study was done.

We agree that our findings imply that within-sibship MR estimates are likely to be highly consistent with population MR estimates for many analyses. However, we note that we

only evaluated MR estimates of height and BMI on 23 phenotypes and so cannot extrapolate our findings to all possible exposures and outcomes.

Furthermore, MR analyses often include sensitivity analyses to test for potential biases. For example, using MR-Egger to evaluate directional horizontal pleiotropy.

True MR was originally proposed in the context of transmission within families ('Mendelian randomization': can genetic epidemiology contribute to understanding environmental determinants of disease? *Davey Smith, Ebrahim IJE 2003*) as such a design is closer to a randomised controlled trial than a population MR study. If suitable data is available, then including within-sibship MR as a sensitivity analysis is reasonable to investigate if MR estimates may be biased by population stratification, assortative mating, or indirect genetic effects. Using a range of sensitivity analyses in MR studies to demonstrate consistency of effect estimates can strengthen the evidence for (absence of) causal relationships.

We have edited the cited sentence in the discussion to clarify that within-sibship MR would be most useful if concerned about potential biases from demographic and indirect genetic effects.

Discussion (page 9)

Before: "In subsequent studies, within-sibship Mendelian randomization could be used as a sensitivity analysis."

After: "In subsequent studies, within-sibship Mendelian randomization could be used as a sensitivity analysis when including phenotypes likely to be affected by demographic or indirect genetic effects."

Decision Letter, second revision:

Our ref: NG-LE57087R1

9th Nov 2021

Dear Dr. Howe,

Thank you for submitting your revised manuscript "Within-sibship GWAS of 25 phenotypes improve estimates of direct genetic effects" (NG-LE57087R1). It has now been seen by the original referees and their comments are below. The reviewers find that the paper has improved in revision, and therefore we'll be happy in principle to publish it in Nature Genetics, pending minor revisions to comply with our editorial and formatting guidelines.

Sincerely,
Wei

Wei Li, PhD
Senior Editor
Nature Genetics
New York, NY 10004, USA
www.nature.com/ng

Reviewer #2 (Remarks to the Author):

I appreciate that the authors have toned down their conclusions and clarified that only a specific if subset of the MR literature is likely to be Influenced by the biases presented.

Final Decision Letter:

In reply please quote: NG-A57087R2 Howe

25th Mar 2022

Dear Dr. Howe,

I am delighted to say that your manuscript "Within-sibship genome-wide association analyses decrease bias in estimates of direct genetic effects" has been accepted for publication in an upcoming issue of Nature Genetics.

Over the next few weeks, your paper will be copyedited to ensure that it conforms to Nature Genetics

style. Once your paper is typeset, you will receive an email with a link to choose the appropriate publishing options for your paper and our Author Services team will be in touch regarding any additional information that may be required.

Your paper will be published online after we receive your corrections and will appear in print in the next available issue. You can find out your date of online publication by contacting the Nature Press Office (press@nature.com) after sending your e-proof corrections. Now is the time to inform your Public Relations or Press Office about your paper, as they might be interested in promoting its publication. This will allow them time to prepare an accurate and satisfactory press release. Include your manuscript tracking number (NG-A57087R2) and the name of the journal, which they will need when they contact our Press Office.

Please note that *Nature Genetics* is a Transformative Journal (TJ). Authors may publish their research with us through the traditional subscription access route or make their paper immediately open access through payment of an article-processing charge (APC). Authors will not be required to make a final decision about access to their article until it has been accepted. [Find out more about Transformative Journals](https://www.springernature.com/gp/open-research/transformative-journals)

Authors may need to take specific actions to achieve [compliance with funder and institutional open access mandates](https://www.springernature.com/gp/open-research/funding/policy-compliance-faqs). If your research is supported by a funder that requires immediate open access (e.g. according to [Plan S principles](https://www.springernature.com/gp/open-research/plan-s-compliance)) then you should select the gold OA route, and we will direct you to the compliant route where possible. For authors selecting the subscription publication route, the journal's standard licensing

terms will need to be accepted, including <https://www.nature.com/nature-portfolio/editorial-policies/self-archiving-and-license-to-publish>. Those licensing terms will supersede any other terms that the author or any third party may assert apply to any version of the manuscript.

Please note that Nature Research offers an immediate open access option only for papers that were first submitted after 1 January, 2021.

If you have not already done so, we invite you to upload the step-by-step protocols used in this manuscript to the Protocols Exchange, part of our on-line web resource, natureprotocols.com. If you complete the upload by the time you receive your manuscript proofs, we can insert links in your article that lead directly to the protocol details. Your protocol will be made freely available upon publication of your paper. By participating in natureprotocols.com, you are enabling researchers to more readily reproduce or adapt the methodology you use. [Natureprotocols.com](https://natureprotocols.com) is fully searchable, providing your protocols and paper with increased utility and visibility. Please submit your protocol to <https://protocolexchange.researchsquare.com/>. After entering your [nature.com](https://www.nature.com) username and password you will need to enter your manuscript number (NG-A57087R2). Further information can be found at <https://www.nature.com/nature-portfolio/editorial-policies/reporting-standards#protocols>

Sincerely,
Wei

Wei Li, PhD
Senior Editor

Nature Genetics
New York, NY 10004, USA
www.nature.com/ng